# BETTER TOGETHER: LEVERAGING UNPAIRED MULTIMODAL DATA FOR STRONGER UNIMODAL MODELS

**Sharut Gupta**
MIT CSAIL
sharut@mit.edu

**Shobhita Sundaram**
MIT CSAIL
shobhita@mit.edu

**Chenyu Wang**
MIT CSAIL
wangchy@mit.edu

**Stefanie Jegelka**
TU Munich, MIT CSAIL
stefje@mit.edu

**Phillip Isola**
MIT CSAIL
phillipi@mit.edu

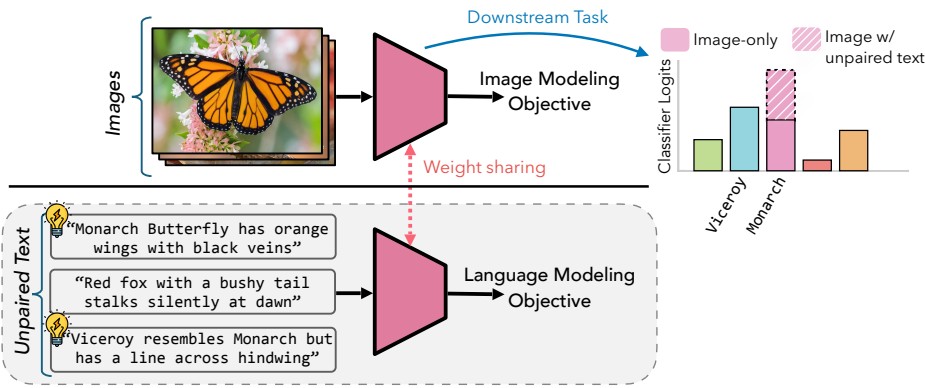

Figure 1: Text provides complementary information beyond images, even when not paired directly; We introduce UNPAIRED MULTIMODAL LEARNER (UML) whereby sharing model weights across modalities (e.g., image and text) extracts synergies and enhances unimodal representations, outperforming methods that rely only on a single modality (such as images above).

## ABSTRACT

Traditional multimodal learners find unified representations for tasks like visual question answering, but rely heavily on large paired datasets. However, an overlooked yet potentially powerful question is: can one leverage auxiliary *unpaired* multimodal data to directly enhance representation learning in a *target* modality? We introduce UML: UNPAIRED MULTIMODAL LEARNER, a modality-agnostic training paradigm in which a single model alternately processes inputs from different modalities while sharing parameters across them. This design exploits the assumption that different modalities are projections of a shared underlying reality, allowing the model to benefit from cross-modal structure without requiring explicit pairs. Theoretically, under linear data-generating assumptions, we show that unpaired auxiliary data can yield representations strictly more informative about the world than unimodal training. Empirically, we show that using unpaired data that share underlying semantic information from auxiliary modalities —such as text, audio, or images—consistently improves downstream performance across diverse unimodal targets such as image and audio. Our project page: https://unpaired-multimodal.github.io/.

# 1 INTRODUCTION

It is often taken for granted that to model a modality well, one must train on data from that modality. For instance, if one wants an accurate image classifier, one trains on images; if one wants a language model, one trains on text. Recent advances in multimodal learning, however, suggest that multiple modalities can benefit one another: in particular, using text captions paired with images yields richer representations, often surpassing their unimodal counterparts in zero-shot transfer, cross-modal retrieval, and downstream classification (Radford et al., 2021; Singh et al., 2022; Mizrahi et al., 2023; Girdhar et al., 2023a; Bachmann et al., 2022; Li et al., 2023; Bachmann et al., 2024; Jia et al., 2021). However, these gains have been realized largely through paired data, where one has access to aligned examples $(x, y) \sim p_{\mathcal{X}, \mathcal{Y}}$, such as an image $x$ and its corresponding caption $y$. Such paired supervision allows aligning modalities into a shared latent space, where cross-modal correlations can be captured and transferred to downstream tasks.

This reliance on paired corpora also poses a bottleneck. Collecting and curating aligned datasets is expensive and domain-limited, whereas unpaired data—independent samples from $p_{\mathcal{X}}$ and $p_{\mathcal{Y}}$—are naturally abundant. For example, vast image collections and vast text corpora exist independently, but without explicit alignment. This raises a more fundamental question:

> *Can unpaired data from a secondary modality $Y$ enhance representations of the target modality $X$, even in the absence of $(x, y)$ correspondences?*

There is growing evidence that the answer may be yes. Recent work posits the existence of a shared statistical model of reality, an empirical parallel to Plato's concept of ideal Forms, suggesting that as deep networks scale, their embeddings across modalities converge toward a unified representation of the underlying world (Huh et al., 2024; Huang et al., 2021). This convergence implies that when paired supervision is available, a model can leverage natural co-occurrences between modalities and thus more accurately capture shared semantics. Critically, however, achieving convergence does not necessarily require explicit pairs; as long as each modality samples from the same underlying ground-truth latent space, it may be sufficient to align their marginal distributions to uncover the common semantic structure (Timilsina et al., 2024; Sturma et al., 2023). Intuitively, even unpaired data from another modality can provide complementary cues to better estimate the underlying reality (Figure 1).

In this work, we formalize this idea through *Unpaired Multimodal Representation Learning*, a framework for improving unimodal representations by leveraging unpaired data across modalities. Theoretically, under linear assumptions, we derive conditions under which incorporating unpaired samples yields strictly more informative representations than unimodal training alone. Strikingly, in some regimes, a single sample from $Y$ provides greater per-sample value than a sample from $X$ itself when the goal is to model $X$. Building on these insights, we introduce **UML: UNPAIRED MULTIMODAL LEARNER**, a shared-network framework that applies the same set of parameters to inputs from different modalities. By nothing more than weight sharing i.e. without surrogate objectives or inferred alignments, the model learns modality-agnostic features and naturally transfers information across modalities (Sutskever, 2023). Importantly, this does not imply that UML or unpaired multimodal training more broadly is immune to the optimization challenges due to modality conflicts documented in prior works for *paired* learning (Huang et al., 2021; Wang et al., 2020; Ma et al., 2025; Du et al., 2021; Xu et al., 2023; Fan et al., 2023; Sun et al., 2021; Zhang et al., 2024; Zhou et al., 2023; Wu et al., 2022); rather, it provides an existence proof of regimes where unpaired multimodal data can help, without guaranteeing such benefits in all multimodal training settings. Empirically, we evaluate UML on diverse image–text tasks in healthcare, and affective computing, as well as 10 standard visual benchmarks. Unpaired data from auxiliary modalities consistently improves unimodal representations across self-supervised and supervised regimes, in both few-shot and full-data regimes, and with diverse encoders including CLIP, DINOv2, OpenLLaMA, and others. These benefits compound when moving from two to three modalities, with audio, vision, and text each adding complementary signals. We further demonstrate effective cross-modal transfer without paired data by initializing vision models with pretrained language-model weights. Finally, we quantify the *exchange rate* between modalities, mapping how many words equate to one image (and vice versa) for optimal performance. These performance gains are clearest when there is shared semantic information between modalities, and do not necessarily apply to every setting (e.g., semantically unrelated modalities).

To summarize, the key contributions of our work are:

- We introduce UML, a modality-agnostic framework that leverages unpaired data to improve unimodal models. We test it across self-supervised and supervised encoders (CLIP, DINOv2, OpenLLaMA, and others), in both few-shot and full-data regimes, showing consistent gains over a range of image-text benchmarks and extensions to audio.

- We theoretically characterize, under linear assumptions, the conditions where unpaired data yield strictly more informative representations than unimodal training. Remarkably, the conversion ratio between modalities can fall below one: in certain regimes, a single sample from $Y$ contributes more to modeling $X$ than an additional sample from $X$ itself.

- We quantify conversion ratios between images and text; i.e., how many words is an image worth for training vision models. We also show that unpaired multimodal data systematically widens inter-class margins and aligns modalities in weights.

## 2 PAIRED MULTIMODAL REPRESENTATION LEARNING

A useful way to conceptualize learning across modalities is to posit a shared ground-truth reality, denoted $\mathcal{Z}^*$, which manifests through multiple projections such as images, text, or audio recordings (Huh et al., 2024; Timilsina et al., 2024; Sturma et al., 2023). The goal of representation learning is to learn an embedding space that captures the structure of $\mathcal{Z}^*$, whether from a single modality or from several jointly. Thus, unimodal representations are inherently limited by what one projection alone can reveal: a single camera view may contain occlusions; an audio recording lacks visual details; textual descriptions may lack layout information.

These limitations motivate multimodal representation learning, which extends the unimodal setting to learn from multiple modalities together. For brevity, consider two modalities with observable data denoted by $\mathcal{X}$ (e.g., images) and $\mathcal{Y}$ (e.g., text) and samples $x \in \mathcal{X}$ and $y \in \mathcal{Y}$. Mathematically, *Multimodal representation learning* seeks encoders $f_{\theta_x} : \mathcal{X} \to \mathcal{Z}$ and $f_{\theta_y} : \mathcal{Y} \to \mathcal{Z}$ mapping each modality into a shared embedding space $\mathcal{Z}$.

Different frameworks optimize for desirable properties of $\mathcal{Z}$. Contrastive methods encourage instances of the same concept (such as an image and its text label) to be close together (Radford et al., 2021; Zhai et al., 2023; Jia et al., 2021). Fusion approaches learn shared layers for multimodal inputs with reconstruction (Singh et al., 2022; Li et al., 2019; Lu et al., 2019; Bachmann et al., 2022) or contrastive objectives (Roy et al., 2025). Generative methods often structure $\mathcal{Z}$ to enable translation between modalities(Li et al., 2022; Rombach et al., 2022). Other methods disentangle common factors from modality-specific factors (Wang et al., 2024; Liang et al., 2023). Further works have also explored the limitations of paired multimodal learning, including paired classification networks, multimodal LLMs, and theoretical multimodal studies (Zhang et al., 2025; Ma et al., 2025; Wang et al., 2020; Javaloy et al., 2022). These works show that in such settings, and without special treatment, modality conflict can give rise to hallucination, optimization, and convergence challenges.

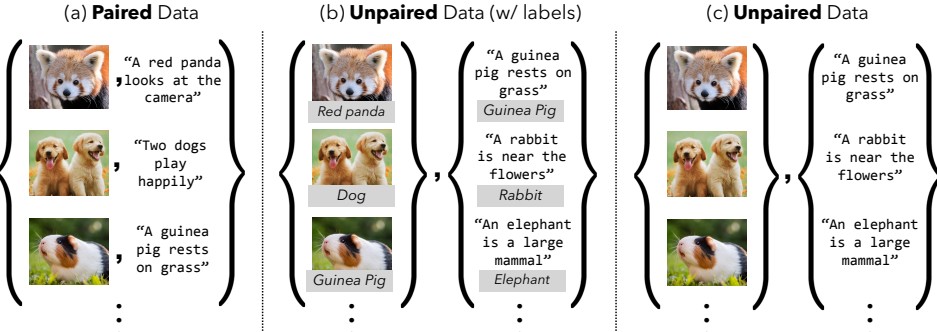

Figure 2: (a) Paired learning uses $(x_i, y_i)$ with known correspondences. We instead study Unpaired learning: (b) with labels, using $(x_i, c_i)$ and $(y_j, \hat{c}_j)$, where $c_i$ and $\hat{c}_j$ denote labels for $x_i$ and $y_j$, but no cross-modal correspondences; and (c) without any labels or correspondences, using $\{x_i\}$ and $\{y_j\}$.

Common to all of these prior work, however, is a requirement for pre-existing knowledge of one-to-one correspondences between the modalities, i.e., samples $(x, y) \sim P_{X,Y}$ (as shown in Figure 2(a)), which are assumed to be aligned views of the same underlying entity in $\mathcal{Z}^*$. We ask whether it is possible to recover this shared structure *without* such correspondences—namely, when only unpaired samples from the marginals $P_X$ and $P_Y$ are available, as shown in Figure 2(c)—and, if so, how this affects downstream performance individually on the primary modality $X$.

## 3 UNPAIRED MULTIMODAL REPRESENTATION LEARNING

Let $\mathcal{D}_{\mathcal{X}} = \{x_i\}_{i=1}^{N_x}$ and $\mathcal{D}_{\mathcal{Y}} = \{y_j\}_{j=1}^{N_y}$ be datasets of $N_x$ and $N_y$ samples respectively, drawn from marginal distributions $P_X$ and $P_Y$ as shown in Figure 2(c). The joint distribution $P_{X,Y}$ is unknown. Critically, in contrast to paired multimodal learning, for any given $x_i \in \mathcal{D}_{\mathcal{X}}$ and $y_j \in \mathcal{D}_{\mathcal{Y}}$, there is no assumption that $x_i$ and $y_i$ are projections of the same underlying entity $z \in \mathcal{Z}$.

Our objective is to learn mappings $f_X : \mathcal{X} \to \mathcal{Z}$ and $f_Y : \mathcal{Y} \to \mathcal{Z}$ from each modality to a common embedding space that captures the shared structures between the unpaired datasets. We refer to this as *Unpaired Multimodal Representation Learning*. In contrast to previous works (Lin et al., 2023; Lee & Yoon, 2025; Girdhar et al., 2022; Chada et al., 2023) (discussed in Appendix A), we leverage unpaired data *without* inferring alignment, incorporating paired data, or assuming pre-aligned embeddings. Intuitively, without known correspondences, even unpaired data can be useful if modalities capture different axes of information pertaining to the underlying $\mathcal{Z}^*$. We formalize this intuition in Section 3.1, within the popularly studied framework of linear models, and introduce a modality-agnostic algorithm UML in Section 3.2 for learning representations from unpaired data. Our analysis is purely information-theoretic and does not characterize optimization or convergence behaviors.

### 3.1 THEORETICAL PERSPECTIVES

We adopt the widely used modeling choice that each modality can be expressed as a linear combination of shared and modality-specific components (Sturma et al., 2023; Timilsina et al., 2024; Huang et al., 2021). This formulation inherently accounts for informational differences across modalities by incorporating modality-specific features (Wang et al., 2024). Prior works (Sturma et al., 2023; Timilsina et al., 2024) have developed sufficient conditions for recovering joint distributions and the shared causal structure under this linear model. Building on this framework, we ask how joint training with unpaired modalities can still enhance representations within a single modality.

**Data Generating Process.** Assume that all factors of variation in reality live in a single $d$-dimensional space, $\mathcal{Z}^*$, i.e., $\theta \in \mathbb{R}^d$, modeled using a linear data-generating pipeline. This parameter can further be decomposed as $\theta \equiv [\theta_c, \theta_x, \theta_y]^\top$ where $\theta_c \in \mathbb{R}^{d_c}, \theta_x \in \mathbb{R}^{d_x}, \theta_y \in \mathbb{R}^{d_y}$ and $d_c + d_x + d_y = d$. Here, $\theta_c$ captures the *common* (shared) parameters that affect both modalities, $\theta_x$ denotes the parameters that only affect modality $X$, and $\theta_y$ denotes the parameters that only affect modality $Y$. We observe two datasets, one from each modality $\{X_i\}_{i=1}^{N_x} \in \mathbb{R}^m$ and $\{Y_j\}_{j=1}^{N_y} \in \mathbb{R}^n$, each reflecting partial measurements of the ground truth latent space $\mathcal{Z}^*$:

$$X_i = A_{c,i}\,\theta_c + A_{x,i}\,\theta_x + \epsilon_{X,i}, \quad \epsilon_{X,i} \sim \mathcal{N}\big(0,\, \sigma_x^2 I_m\big) \tag{1}$$

$$Y_j = B_{c,j}\,\theta_c + B_{y,j}\,\theta_y + \epsilon_{Y,j}, \quad \epsilon_{Y,j} \sim \mathcal{N}\big(0,\, \sigma_y^2 I_n\big). \tag{2}$$

Here, $A_{c,i}, A_{x,i}, B_{c,j}, B_{y,j}$ are known design blocks capturing how each sample probes the latent factors and $\varepsilon_{X,i}, \varepsilon_{Y,j}$ represent the independent measurement noise.

In our linear setting, estimating the true latent state $\theta$—and hence the underlying reality $\mathcal{Z}^*$—is governed by the Fisher information matrix $I(\theta) = -\mathbb{E}\big[\nabla_\theta^2 \ell(\theta)\big]$ (Fisher, 1922; 1925), which measures how sharply the likelihood "curves" around the true $\theta$. Because $X$ and $Y$ samples are independent given $\mathcal{Z}^*$, their curvature contributions add pointwise, resulting in the joint Fisher information being simply the sum of the unimodal blocks. Thus, loosely speaking, any nonzero contribution from the unpaired $Y$-samples strictly increases curvature and thus strictly tightens the variance bound along those directions in $\theta_c$. We formalize these insights in Theorem 1 and Theorem 2. Our results also generalize naturally to more than two modalities since the total contribution of all auxiliary modalities (excluding the primary modality $X$) can be obtained by summing their individual Fisher information matrices. Complete proofs and additional background are provided in Appendix C.

**Theorem 1** (Variance Reduction with Unpaired Multimodal Data). *Let $\hat{\theta}_X, \hat{\theta}_Y$ be the least-squares estimators for $\theta$ using only $\{X_i\}$ and only $\{Y_j\}$ and let $\hat{\theta}_{X,Y}$ be the joint estimator using both unpaired datasets. Then, under the assumption that at least one $B_{c,j}$, where $j \in \{1, 2, ...N_y\}$, has full rank, the common-factor covariance satisfies the strict Loewner ordering i.e. $\mathrm{Var}\big(\hat{\theta}_{X,Y}\big)_{\theta_c, \theta_c} \prec \mathrm{Var}\big(\hat{\theta}_X\big)_{\theta_c, \theta_c}$, or equivalently, the Fisher information on $\theta_c$ strictly increases when combining both modalities, despite not having sample-wise pairing:$(I_X + I_Y)_{\theta_c, \theta_c} \succ (I_X)_{\theta_c, \theta_c}$.*

Theorem 1 says that although there is no sample-wise pairing, simply adding an unpaired modality $Y$, that carries non-degenerate information about every direction in the shared subspace, can only tighten our uncertainty about the shared parameters $\theta_c$. Geometrically, the uncertainty ellipsoid for $\theta_c$ shrinks in every direction where $Y$ contributes any curvature, and never expands elsewhere. In practice however, no single data sample covers every latent axis. In these settings, while global shrinkage (Theorem 1) no longer applies, we still get *directional* gains. We capture this in Theorem 2.

**Theorem 2** (Directional Variance Reduction with Unpaired Multimodal Data). *Let all notation be as in Theorem 1, and let $v \in \mathbb{R}^{d_c} \setminus \{0\}$. If there exists at least one index $j \in \{1, 2, ...N_y\}$ such that $B_{c,j}v \neq 0$, then $v^\top (I_X + I_Y)_{\theta_c, \theta_c} v > v^\top (I_X)_{\theta_c, \theta_c} v$. Further, the variance of the estimator in direction $v$ is strictly reduced if $v \notin range((I_X)_{\theta_c, \theta_c})$.*

Building on Theorem 2, Corollary 1 and Corollary 2 quantify the precise variance contraction factor and analyze special cases where a second modality resolves otherwise ill-posed directions.

Thus far, we have studied incorporating data from an auxiliary modality without considering sample size constraints. A natural next question is how to allocate a fixed budget of $N$ additional samples: is it better to collect them from the same modality $X$, or from complementary modality $Y$? We formally address this in Theorem 3 below.

**Theorem 3** (Data from Auxiliary Modality Can Outperform More of the Same). *Define for any $m$, $I_X^{(m)} = \sum_{i=1}^m A_{c,i}^\top A_{c,i}$ and $I_Y^{(m)} = \sum_{j=1}^m B_{c,j}^\top B_{c,j}$. If $\mathrm{range}\big(I_Y^{(m)}\big) \not\subseteq \mathrm{range}\big(I_X^{(m)}\big)$, then there exists a nonzero $v \in \mathbb{R}^{d_c}$ such that $v^\top I_Y^{(m)} v > v^\top I_X^{(m)} v$.*

Theorem 3 shows that if the second modality covers a "blind spot" of the first, adding additional samples from the first modality does not increase the Fisher information in that direction; however, unpaired samples from the second modality provide strictly positive information along that axis.

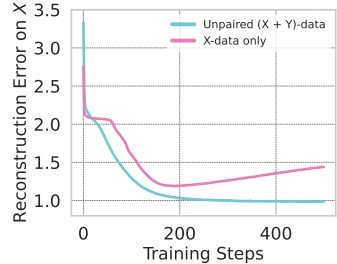

Figure 3: Adding unpaired $Y$ samples boosts $X$ reconstruction more than adding extra $X$ samples.

To empirically validate the above result, we design an illustrative Gaussian experiment (Appendix E.14) in which both modalities are generated as noisy linear projections of the same underlying Gaussian latent variable $\theta_c$. We allocate a fixed budget of samples, splitting evenly between $X$ and $Y$ (details in Appendix E.14). Joint training with UML significantly improves reconstruction fidelity on $X$ (Figure 3) compared to training on $X$ alone. Strikingly, this also reveals that the *effective exchange rate between modalities is below one*, meaning a single additional sample from $Y$ is worth more than an additional sample from $X$, even when the test task is on $X$. In Section 4.4, we extend this idea by quantifying the exchange rate between images and text on real world benchmarks.

## 3.2 UML: UNPAIRED MULTIMODAL LEARNER

The theory developed above establishes that adding an unpaired auxiliary modality strictly increases Fisher information along shared directions, thereby reducing estimator variance. We now translate these insights into a practical framework: UML (UNPAIRED MULTIMODAL LEARNER). The central idea is to share parameters across modalities: since both $X$ and $Y$ are projections of the same underlying reality $\mathcal{Z}^*$, forcing them through shared weights can extract synergies by accumulating training gradients on the same parameters. Under the assumptions in Proposition 1, this accumulation yields exact Fisher-information additivity on the shared parameters, ensuring that even without paired

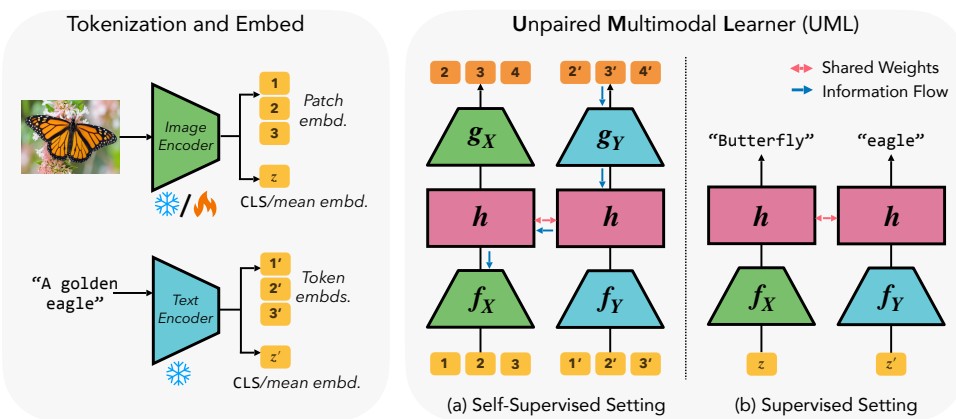

Figure 4: (Left) Inputs from different modalities (e.g., images or text) are tokenized into patch or token embeddings using pretrained encoders or processed features; (Right) UML can be trained under two settings: (a) Self-supervised, where patch/token embeddings are passed through a shared network and modality-specific decoders to perform next-token/patch prediction; (b) Supervised, where mean/CLS embeddings are fed through the shared classifier to predict labels within each modality.

samples, auxiliary modalities tighten estimates of shared structure. While prior work has explored shared weights in partially paired settings (Sturma et al., 2023), for related visual streams such as video and depth (Girdhar et al., 2022), or within already-aligned spaces like CLIP (Lin et al., 2023), we study and show their pronounced impact in the most general case of unpaired modalities with unaligned encoders.

Consider unpaired samples or preprocessed features $x \sim P_X$ and $y \sim P_Y$, each encoded into embeddings $z_X = f_X(x)$ and $z_Y = f_Y(y)$ as shown in Figure 4. These embeddings are passed through a shared network $h$, producing $r_X = h(z_X)$ and $r_Y = h(z_Y)$. We consider two regimes to train these networks: (a) self-supervised learning for fully unpaired data (Figure 2(c)) and (b) supervised learning for unpaired data with labels as shown in Figure 2(b). In the self-supervised setting, each modality has its own decoder ($g_X$ and $g_Y$) trained to reconstruct the input or predict the next token/patch depending on the form of embeddings $z_X$ and $z_Y$ (refer to Figure 4(a)). The training objective (Pseudocode in the appendix: Algorithm 1) is

$$\mathcal{L}_{\text{UML-SSL}} = \mathbb{E}_{x \sim P_X} \ell\big(g_X(r_X),\, x\big) + \mathbb{E}_{y \sim P_Y} \ell\big(g_Y(r_Y),\, y\big),$$

where $\ell$ is the mean squared error for continuous targets or cross-entropy for discrete tokens. In the supervised setting, when labels are available, $c_X$ for $x$ and $c_Y$ for $y$, a shared classifier $c(\cdot)$ is trained on top of $r_X$ and $r_Y$. Since both $c(\cdot)$ and $h(\cdot)$ are shared across modalities, they can be viewed as a single shared head $h$ (refer to Figure 4(b)). The supervised loss (Pseudocode in the appendix: Algorithm 2) is

$$\mathcal{L}_{\text{UML-Sup}} = \mathbb{E}_{(x,c_X)} \ell_{\text{CE}}\big(c(r_X),\, c_X\big) + \mathbb{E}_{(y,c_Y)} \ell_{\text{CE}}\big(c(r_Y),\, c_Y\big).$$

In both scenarios, although supervision is modality-specific, the shared head $h$ receives updates from both modalities. Consequently, gradients from $h$ also flow into $f_X$, effectively transferring information from $f_Y$ and thus $\mathcal{Y}$ even without paired samples. At inference, we discard the auxiliary pathway and use $r_X$ as the representation for the target modality, training a linear probe on top for downstream tasks. UML also naturally extends to more than two modalities by incorporating additional modality-specific encoder/decoder heads or classification layers.

## 4 EXPERIMENTAL RESULTS

In this section, we empirically evaluate UML across diverse benchmarks and configurations, leading to three main takeaways: (i) auxiliary modalities consistently boost target image representations in both self-supervised (Section 4.1) and supervised regimes (Section 4.2), with particularly strong gains in fine-grained and low-shot tasks; (ii) the benefits compound as we move from two to three modalities, with audio, vision, and text each contributing complementary signals (Section 4.2); and

(iii) weight sharing across modalities generalizes beyond co-training, as pretrained language model weights *transfer* effectively to vision (Section 4.3). Finally, we quantify a marginal rate of substitution between modalities, asking how many text samples equate to an image (Section 4.4). While our benchmarks include samples without corresponding data in other modalities (e.g., classes that appear only in images or only in text) and thus touch the missing-modality regime (Dai et al., 2025; Kim & Kim, 2024), they are not constructed as large-scale missing-modality benchmarks formed by simply combining independent unimodal datasets. Our theoretical framework nonetheless subsumes this setting, as it does not require every class or instance to appear in every modality.

## 4.1 UML IN SELF-SUPERVISED SETTING

Table 1: UML (Ours) achieves higher top-1 linear probe accuracy than unimodal baselines on learned representations, averaged over three seeds, across both MultiBench and vision–text benchmarks.

| | Dataset | | | | | | | |
| | MultiBench | | | | | Standard vision-text | | |
| Method | MUSTARD | MIMIC | MOSEI | MOSI | UR-FUNNY | Oxford Pets | UCF101 | DTD |
|---|---|---|---|---|---|---|---|---|
| Unimodal | 59.66 | 55.16 | 70.62 | 56.17 | 56.99 | 85.04 | 79.86 | 78.13 |
| Ours | 63.28 ↑ | 57.10 ↑ | 71.98 ↑ | 58.16 ↑ | 57.34 ↑ | 86.32 ↑ | 80.98 ↑ | 78.49 ↑ |

To study UML in the self-supervised setting, we use the real-world multimodal benchmark from MultiBench (Liang et al., 2021), which includes curated datasets such as text and images, with downstream labels covering a variety of domains, including healthcare, affective computing, and multimedia research and three standard classification benchmarks (Parkhi et al., 2012; Cimpoi et al., 2014; Soomro et al., 2012). We follow the same setting (dataset splitting, encoder architecture, pre-extracted features) as (Liang et al., 2023; 2021).

For training (refer to Figure 4(a)), we use linear encoders ($f_X$ and $f_Y$) to project each modality to a shared embedding space. Projected embeddings are then processed by a shared network ($h$): an autoregressive 5-layer 5-head transformer. Linear decoders ($g_X$ and $g_Y$) then project back to the original modality dimension. We train with a next-token/patch-embedding prediction objective. At inference, we average the transformer output embeddings and use the resulting embedding for linear probing on the downstream classification tasks.

We report test accuracy using embeddings from the primary modality (image), and to ensure fair comparisons, we perform rigorous hyperparameter tuning for all methods and repeat each experiment with three random seeds. For more details, refer to Appendix B.3.1 and Appendix B.4.1. As shown in Table 1, UML significantly outperforms its unimodal counterpart across all benchmarks, particularly on MUSTARD where unique information from the text modality expresses sarcasm, such as ironic tone of voice.

## 4.2 UML IN SUPERVISED SETTING

We evaluate UML on 9 widely-used visual classification benchmarks (Fei-Fei et al., 2004; Parkhi et al., 2012; Krause et al., 2013; Nilsback & Zisserman, 2008; Bossard et al., 2014; Xiao et al., 2010; Cimpoi et al., 2014; Soomro et al., 2012) in three settings: (1) Full fine-tuning: initializing from a pretrained vision backbone and updating all parameters on the target dataset. (2) Few-shot linear probing: freezing the vision backbone and training a linear classifier on $k$ labeled samples per class ($k = 1, 2, 4, 8, 16$). (3) Full-dataset linear probing: freezing the vision backbone and training a linear classifier on the entire training dataset, discussed in Appendix E.1.2. In all cases, we enrich image representations with unpaired text embeddings, using ViT-S/14 DINOv2 as the vision encoder and OpenLLaMA-3B as the frozen text encoder. To construct conceptually related yet unpaired text data, we generate text templates with varying amounts of semantic information about the dataset. For further details and specific prompts, refer to Appendix B.3. To train UML, we initialize the classifier with the average text embedding of each class, giving a strong prior for image–class alignment (refer to Appendix E.1 for ablation).

Table 2: UML (Ours) outperforms unimodal baseline on image classification with ViT-S/14 DINOv2 and OpenLLaMA-3B in both settings: (i) Full finetuning and (ii) Few-shot linear probing ($k = 1, 2, 4$).

| Shot | Method | Stanford Cars | SUN397 | FGVC Aircraft | DTD | UCF101 | Food101 | Oxford Pets | Oxford Flowers | Caltech101 | Average |
|------|--------|---------------|--------|---------------|-----|--------|---------|-------------|----------------|------------|---------|
| *Full-finetuning* | | | | | | | | | | | |
| | Unimodal | 79.45 | 66.20 | 66.99 | 72.16 | 83.18 | 80.65 | 90.67 | 99.18 | 95.45 | 81.54 |
| | Ours | 86.39 ↑ | 66.03 ↓ | 73.44 ↑ | 74.27 ↑ | 84.69 ↑ | 81.97 ↑ | 91.72 ↑ | 99.82 ↑ | 97.60 ↑ | 83.99 ↑ |
| *Few-shot Linear Probing* | | | | | | | | | | | |
| 1 | Unimodal | 13.18 | 34.15 | 14.09 | 36.60 | 46.74 | 35.18 | 63.51 | 89.62 | 76.66 | 45.52 |
| | Ours | 16.49 ↑ | 41.79 ↑ | 15.63 ↑ | 42.04 ↑ | 52.33 ↑ | 42.27 ↑ | 73.59 ↑ | 93.64 ↑ | 84.52 ↑ | 51.36 ↑ |
| 2 | Unimodal | 24.68 | 47.88 | 23.09 | 47.75 | 56.81 | 48.54 | 75.32 | 96.02 | 86.90 | 56.33 |
| | Ours | 28.65 ↑ | 53.15 ↑ | 24.78 ↑ | 53.25 ↑ | 63.86 ↑ | 54.44 ↑ | 81.41 ↑ | 97.63 ↑ | 90.55 ↑ | 60.85 ↑ |
| 4 | Unimodal | 38.76 | 57.51 | 32.10 | 59.69 | 67.75 | 60.79 | 83.89 | 98.59 | 93.48 | 65.84 |
| | Ours | 43.17 ↑ | 60.89 ↑ | 33.86 ↑ | 62.43 ↑ | 71.13 ↑ | 63.88 ↑ | 87.36 ↑ | 99.17 ↑ | 94.96 ↑ | 68.53 ↑ |

**Unpaired Textual Data Improves Visual Classification.** As shown in Table 2, across both full fine-tuning and few-shot linear probing, UML consistently improves over unimodal baselines across all datasets, with the largest gains on fine-grained tasks (e.g., Stanford Cars, FGVC Aircraft) where unpaired text sharpens class boundaries, and in low-shot regimes where textual cues help disambiguate visually similar classes. Additional results on other shots, datasets, model scales, and prompt variants are reported in Appendix E.

We also evaluate the robustness of UML-trained models under test-time distribution shifts. A 16-shot linear probe with DINOv2 is trained on ImageNet and tested across four distribution-shifted target datasets: ImageNet-V2, ImageNet-Sketch, ImageNet-A, and ImageNet-R. UML consistently outperforms the unimodal baseline (Figure 5), showing that language priors yield more transferable features. Additional robustness results are provided in Appendix E.3.

Finally, for all these settings, we also replace the independent vision (DINOv2) and text encoders (OpenLLaMA) with those of CLIP; since CLIP embeddings are already aligned, the gains from UML are even stronger (Appendix E.2.1, Appendix E.2.2, Appendix E.2.3,).

*Additional Ablations.* For all experiments, we keep the text encoder frozen; Unfreezing the text encoder yields slightly weaker gains but still outperforms the unimodal baseline (Appendix E.12). Swapping in different text encoders yields consistent gains (Appendix E.6), while richer, more diverse captions provide especially strong boosts in few-shot settings (Appendix E.7). Further, training with semantically unrelated modalities (Appendix E.9) does not improve over the unimodal counterpart, confirming that gains stem from semantic correlations rather than spurious ones (Appendix E.9). Finally, unpaired multimodal data systematically widens inter-class margins and aligns modalities in weights (Appendix F).

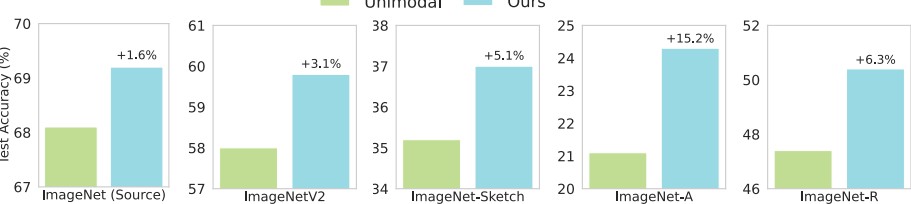

Figure 5: Our approach UML is much more robust than its unimodal counterpart across four test time distribution shifted target test sets. All results are averaged across three random seeds.

**Unpaired Image and Textual Data Improve Audio classification.** We extend UML to an audio–vision–text setting using the ImageNet-ESC benchmark (Lin et al., 2023), which links ImageNet objects and captions with ESC-50 environmental sounds. The benchmark has two

versions: ImageNet-ESC-27 and ImageNet-ESC-19. For audio encoding, we use AudioCLIP with an ES-ResNeXT backbone (Guzhov et al., 2021), while images and text are encoded by DINOv2 and OpenLLaMA-3B encoders. For further details, refer to Appendix B.3.

Unpaired image and text data consistently improve audio classification (Figure 6), with larger gains from CLIP's aligned encoders (Appendix E.13.3). Conversely, both audio and text also enhance image classification, showing that transfer works in both directions (Appendix E.13).

Finally, we study the full three-modality case, where we treat two modalities as auxiliaries and one as the target—for example, using both image and text as auxiliaries for audio classification. As shown in Appendix E.10, performance improves monotonically with each added auxiliary modality, with the strongest gains achieved when all three modalities (audio, vision, and text) are used together.

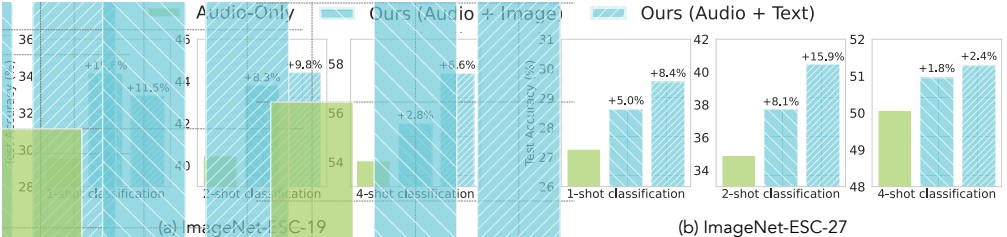

Figure 6: UML (Ours) improves audio classification using unpaired image and text samples on ImageNet-ESC-19 and ImageNet-ESC-27. All results are averaged across three random seeds.

## 4.3 TRANSFER LEARNING

Thus far, we have explored how sharing model weights while co-training with multiple unpaired modalities improves the learned representation. But weight sharing need not be restricted to co-training: if modalities capture the same underlying latents, then pretrained weights from one should also serve as a useful initialization for another. We therefore study if *transferring* knowledge from one modality can enhance performance in another by initializing the transformer layers of a ViT (Dosovitskiy et al., 2020) with pretrained BERT (Devlin et al., 2019) weights and evaluating on ImageNet (details in Appendix B.4.4). Patch embeddings are learned from scratch through a linear layer, augmented by a CLS token and positional embeddings. As shown in Figure 7, initializing with BERT boosts performance for both frozen and unfrozen backbones. Our results indicate that the semantic knowledge of language models provides a strong initialization for vision compared to training from scratch.

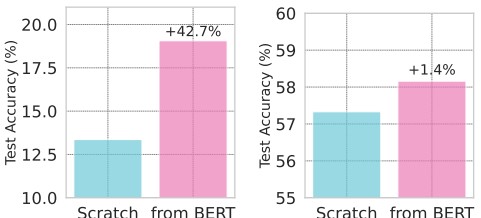

Figure 7: Image classifier trained from BERT initialization outperforms training from scratch for (left) full fine-tuning; (right) frozen backbone

## 4.4 MARGINAL RATE-OF-SUBSTITUTION BETWEEN MODALITIES

Having established that unpaired modalities boost representation learning, robustness, and transfer, we now ask a more fundamental question: what is the relative value of each modality? If both images and text provide views of the same semantic space, can we quantify their exchange rate—*How many words is an image worth?* Figure 8 and Figure 9 visualize image-text conversion ratios using test accuracy isolines on Oxford-Pets (Parkhi et al., 2012) dataset. These map the number of texts equivalent to an image for the same performance. Aligned CLIP encoder (1 image $\approx$ 228 words) is more efficient than non-aligned DINOv2 and OpenLLaMA encoders (1 image $\approx$ 1034 words). Indeed, in some cases, an image may quite literally be worth a thousand words. We also observe little or no additional benefit from adding more text beyond a few samples. This is likely because we

do not control for increasing complexity, so adding sentences does not guarantee extra information. Further results on additional datasets and key details, refer to Appendix E.4.

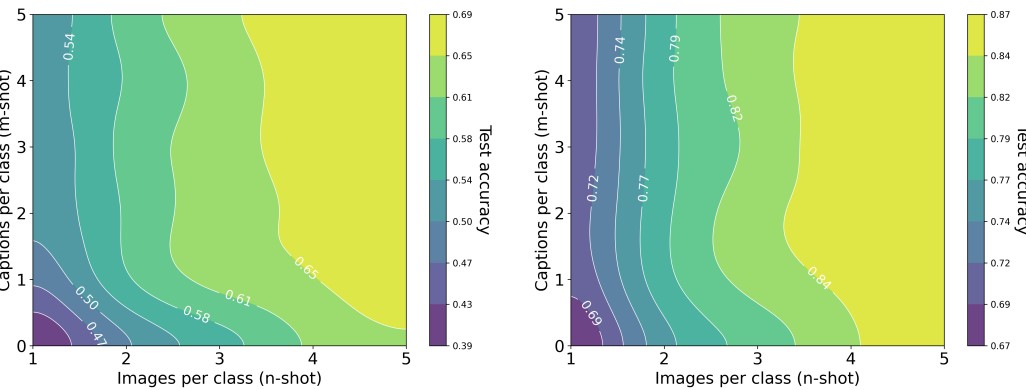

Figure 8: 1 img ≈ 228 words (CLIP)    Figure 9: 1 img ≈ 1034 words (DINOv2)

## 5 CONCLUSIONS AND LIMITATIONS

**Conclusions.** We propose and investigate *Unpaired Multimodal Representation Learning* for enhancing unimodal representations with unpaired multimodal data. Under linear assumptions, we theoretically show that unpaired data from multiple modalities strictly increases Fisher information along shared directions, resulting in a more accurate representation of the underlying world. Mechanistically, UNPAIRED MULTIMODAL LEARNER (UML) achieves this by accumulating gradients from different modalities on shared weights, which can be viewed as an operational analogue of the Fisher information gain. Empirically, we show performance gains across vision, text and audio benchmarks, and estimate conversion ratios between modalities. UML provides a new perspective on how to harness the abundance of unpaired data to learn better representations. This maybe especially useful in domains such as medical imaging, scientific data, and robotics, which contain rich auxiliary modalities like text, audio, or metadata that are not often paired with every instance of the primary modality.

**Limitations and Future Work.** While our experiments study learning from unpaired multimodal data under both the self-supervised and supervised settings, downstream evaluations are conducted primarily for classification. A natural extension of UML is to combine separate single-modality datasets to explicitly model missing modalities and partial overlap across modalities; this regime is already subsumed by our theoretical framework, and our experiments probe it in controlled settings, but a systematic empirical study remains an important direction for future work. Furthermore, we evaluate how multimodal data enhances image and audio classification; it remains to show if they can, in turn, offer useful information for textual tasks. Finally, our results do not necessarily extend to every possible setup of unpaired multimodal learning. In particular, we do not explicitly model or control gradient interference, modality collapse, or conflicting supervisory signals, which are known to induce optimization instability, negative transfer, and modality collapse in *paired* multimodal systems. Systematically characterizing these phenomena in the unpaired setting remains an important direction for future work.

## 6 ACKNOWLEDGMENTS

This research was sponsored by the Department of the Air Force Artificial Intelligence Accelerator under Cooperative Agreement Number FA8750-19-2-1000, in part by the NSF AI Institute TILOS (NSF CCF-2112665) and the Alexander von Humboldt Foundation. This work was also supported by a Packard Fellowship to P.I., and by ONR MURI grant N00014-22-1-2740. S.G. is supported by the MathWorks Engineering Fellowship. S.S. is supported by an NSF GRFP fellowship. The views and conclusions contained in this document are those of the authors and should not be interpreted as representing the official policies, either expressed or implied, of the Department of the Air Force or the U.S. Government. The U.S. Government is authorized to reproduce and distribute reprints for Government purposes, notwithstanding any copyright notation herein.

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

APPENDIX

# A  FURTHER RELATED WORKS

**Unpaired Multimodal Learning.**    Unpaired data has long been used for image-to-image (Zhu et al., 2017; Liu et al., 2017; Almahairi et al., 2018; Shi et al., 2023) and text-to-text translation (Lample et al., 2018) . More recently, several works have also proposed learning from unpaired data by inferring coarse- or fine-grained alignments through distribution matching or optimal transport objectives (Xi et al., 2024; Demetci et al., 2022; Ryu et al., 2024). In contrast, we leverage unpaired data for learning representations without the need for explicit or inferred alignment. (Timilsina et al., 2024; Sturma et al., 2023) theoretically analyze the problem of identifying shared latent components and causal structures in unaligned multimodal mixtures. Most closely related to our work is (Lin et al., 2023), which leverages coarse-grained text data such as class names to improve image classification on CLIP using a shared linear head. Another related line of works (Roth et al., 2023; Pratt et al., 2023; Menon & Vondrick, 2022; Gao et al., 2024) leverage prompting templates and pretrained LLMs to generate descriptive class captions, showing improved image classification performance with CLIP. Nonetheless, these methods operate on CLIP with pre-aligned representation spaces, whereas our approach also learns from unpaired data without assuming prior alignment. Several works have also proposed learning large multitask multimodal models with joint encoders and unified embedding spaces (Srivastava & Sharma, 2024; 2023; Zhang et al., 2023; Girdhar et al., 2022; Geng et al., 2022), often using joint training over separate tasks and/or masked prediction objectives. In a similar vein, (Chada et al., 2023) uses a stage-wise training strategy with both unpaired and paired data, and (Girdhar et al., 2023b) trains a single model across visual modalities. However, most of these methods rely on some amount of paired data for preliminary alignment and then leverage abundant modality-specific unpaired data for further improvement. In contrast, our approach demonstrates that a model can implicitly learn cross-modal correlations from purely unpaired data, without requiring explicit alignment as a prerequisite.

**Multimodal Representation Alignment.**    Our method relies on the notion of shared information and structure between unaligned modalities. Closely related to this are works demonstrating that unimodal representations trained without multimodal data are nevertheless converging. Huh et al. (2024) presents evidence that better-performing language models exhibit increased alignment to self-supervised vision models. Similarly, (Maniparambil et al., 2024) shows a latent space alignment between vision and text encoders across backbones and training paradigms, and uses the CKA metric to connect unaligned encoders zero-shot. Earlier works also note alignment between models trained with different datasets and modalities (Moschella et al., 2022; Norelli et al., 2023). Several works have also shown that a linear projection or MLP is sufficient to stitch together the latent spaces of pretrained vision and language models (Merullo et al., 2022; Liu et al., 2023; Koh et al., 2023). Zhai et al. (2022) extends this to training a text encoder to align to a frozen pretrained image model; this method was in turn used to integrate DINOv2, a large self-supervised vision model, with a text encoder (Jose et al., 2024).

**Modality Conflicts and Imbalance.**    A complementary line of work studies when joint multimodal training *fails* due to conflicts between modalities (Huang et al., 2021; Wang et al., 2020; Ma et al., 2025; Du et al., 2021; Xu et al., 2023; Fan et al., 2023; Sun et al., 2021; Zhang et al., 2024; Zhou et al., 2023; Wu et al., 2022). Here, *modality competition* typically refers to the phenomenon that one modality (often the easier or higher-SNR one) dominates optimization, causing the other modality to be under-utilized or even harmful to final performance. For example, Wang et al. (2020) shows that multimodal classifiers trained on *paired* inputs can underperform strong unimodal baselines because modalities converge and generalize at different rates, and propose gradient reweighting to address this. Fan et al. (2023) and related works explicitly treat modality imbalance as a competition for decision boundaries and introduce prototypical rebalancing to prevent a dominant modality from suppressing weaker ones. Ma et al. (2025) further attributes such conflicts to batch-level dynamics and proposes data remixing strategies to decouple and rebalance multimodal batches. All of these works focus on joint training with *paired* data (often training from scratch) and design mechanisms to mitigate conflict in that regime. A related line of work analyzes *modality collapse*, where representations degenerate to rely on a single modality, a phenomenon that can be especially severe under missing-modality patterns (Kim & Kim, 2024; Wu et al., 2024; Chaudhuri et al., 2025; Dai et al., 2025; Javaloy et al., 2022). These methods consider paired or partially paired training with systematically missing inputs, and design objectives that either correct the bias induced by absent modalities or predict

the representation of the missing modality. By contrast, our setting uses features from pretrained encoders or backbones initialized from pretrained models, together with fully unpaired data. In this regime, we do not observe the suboptimal modality competition that these works seek to mitigate.

## B  SUPPLEMENTARY EXPERIMENTAL DETAILS AND ASSETS DISCLOSURE

### B.1  ASSETS

We do not introduce new data in the course of this work. Instead, we use publicly available, widely used image datasets for the purposes of benchmarking and comparison.

### B.2  HARDWARE AND SETUP

Each experiment was conducted on 1 NVIDIA Tesla V100 GPUs, each with 32GB of accelerator RAM. The CPUs used were Intel Xeon E5-2698 v4 processors with 20 cores and 384GB of RAM. All experiments were implemented using the PyTorch deep learning framework.

### B.3  DATASETS

#### B.3.1  MULTIBENCH

We evaluate our approach on a diverse set of multimodal fusion datasets from MultiBench (Liang et al., 2021) spanning healthcare, sentiment, and humor detection:

- **CMU-MOSEI** (Moschella et al., 2022): The largest sentence-level multimodal sentiment and emotion dataset, containing 23,000 annotated monologue videos (over 65 hours of content from more than 1,000 speakers across 250 topics). Each video is labeled with sentiment intensity in the range $[-3, 3]$, which we cast into binary positive/negative sentiment classification.

- **CMU-MOSI** (Zadeh et al., 2016): A related multimodal sentiment dataset with 2,199 YouTube video clips, reflecting real-world opinionated speech. Sentiment intensities are annotated in the range $[-3, 3]$, and we again formulate the task as binary sentiment classification.

- **UR-FUNNY** (Hasan et al., 2019): A large-scale humor detection dataset derived from over 16,000 TED talk videos. It includes 8,257 humorous punchlines identified by laughter markers, paired with 8,257 negative examples drawn from non-humorous contexts, forming a balanced binary humor classification task.

- **MUSTARD** (Castro et al., 2019): A multimodal sarcasm detection dataset with 690 video clips from TV shows. Each sample is annotated for the presence or absence of sarcasm, yielding a challenging binary classification task.

- **MIMIC-III** (Johnson et al., 2016): A large-scale clinical dataset with records of over 40,000 ICU patients. It combines time-series physiological measurements (recorded hourly over a 24-hour window) with static demographic and tabular features. We use it for binary classification of patients into a group of ICD-9 codes.

#### B.3.2  IMAGE CLASSIFICATION BENCHMARKS

We evaluate on the following widely-used classification benchmarks: ImageNet (Deng et al., 2009), StanfordCars (Krause et al., 2013), UCF101 (Soomro et al., 2012), Caltech101 (Fei-Fei et al., 2004), Oxford Flowers (Nilsback & Zisserman, 2008), SUN397 (Xiao et al., 2010), DTD (Cimpoi et al., 2014), FGVCAircraft (Maji et al., 2013), OxfordPets (Parkhi et al., 2012), and Food101 (Bossard et al., 2014). More details about the dataset and splits is provided in Table 3.

#### B.3.3  CONSTRUCTING TEXT TEMPLATES

To construct conceptually related yet unpaired text data, we generate text templates that capture varying granularities of information about the dataset. Our first approach (*Vanilla*) uses the

Table 3: Detailed statistics of the 10 datasets for image classification.

| Dataset | Classes | Train | Val | Test |
|---|---|---|---|---|
| Caltech101 (Fei-Fei et al., 2004) | 100 | 4,128 | 1,649 | 2,465 |
| OxfordPets (Parkhi et al., 2012) | 37 | 2,944 | 736 | 3,669 |
| StanfordCars (Krause et al., 2013) | 196 | 6,509 | 1,635 | 8,041 |
| Oxford Flowers (Nilsback & Zisserman, 2008) | 102 | 4,093 | 1,633 | 2,463 |
| Food101 (Bossard et al., 2014) | 101 | 50,500 | 20,200 | 30,300 |
| FGVCAircraft (Maji et al., 2013) | 100 | 3,334 | 3,333 | 3,333 |
| SUN397 (Xiao et al., 2010) | 397 | 15,880 | 3,970 | 19,850 |
| DTD (Cimpoi et al., 2014) | 47 | 2,820 | 1,128 | 1,692 |
| UCF101 (Soomro et al., 2012) | 101 | 7,639 | 1,898 | 3,783 |
| ImageNet (Deng et al., 2009) | 1,000 | 1.28M | N/A | 50,000 |

straightforward template \a photo of a {}" with a natural language label for each category, resulting in a basic text description for each class. However, this simple textual corpus lacks fine-grained information necessary to distinguish between visually similar subcategories or to resolve contextually ambiguous terms. To address this, for the second template, we draw from the extensive literature on improving text prompts for zero-shot classification in CLIP (Gao et al., 2024; Menon & Vondrick, 2022; Pratt et al., 2023; Roth et al., 2023). Specifically, for the second approach (*GPT-3 Descriptions*), we adopt the text prompt generation strategy developed by Pratt et al. (2023), using large language models such as GPT-3 to generate diverse and contextually rich prompts for each image category. We use three generic hand-written sentences across the datasets:

```
Describe what a/the {} looks like:
             Describe a/the {} :
What are the identifying characteristics of a/the {}?
```

The blank portion of each template is populated with the category name, along with the category type for specialized datasets (e.g., "pet" + {} for Oxford Pets or "aircraft" + {} for FGVC Aircraft). The type specification is important for disambiguating categories with multiple interpretations. Some examples of these descriptions are provided in Table 4 for the Oxford Pets dataset.

### B.3.4   IMAGENET-ESC DATASET

**Experimental Setup.** We extend our results beyond vision and language to an audiovisual-language dataset: the ImageNet-ESC benchmark (Lin et al., 2023). This benchmark combines ImageNet (1000 object categories) and ESC-50 (50 environmental sound classes) by matching classes that logically correspond. For example, the dog (barking) class from ESC-50 aligns with various dog breeds from ImageNet, while the clock-alarm sound maps to both analog clock and digital clock. This alignment captures the relationship between visual objects, their sounds, and their textual descriptions. The benchmark consists of two versions: 1) ImageNet-ESC-27: A broader set including loosely matched visual-audio pairs (e.g., drinking-sipping to water bottle); 2) ImageNet-ESC-19: A more precise subset containing only accurate visual-audio matches.

### B.4   TRAINING PROTOCOL

### B.4.1   UNPAIRED MULTIMODAL REPRESENTATION LEARNING UNDER SELF-SUPERVISION

**MultiBench.** For the MIMIC dataset, which contains tabular and medical time-series inputs, we train models directly on the raw modality inputs. For the four video datasets (MOSEI, MOSI, UR-FUNNY, and MUSTARD), we train models on standard pre-extracted features from text, video, and audio modalities (Liang et al., 2021). All models are trained for 100 epochs, with hyperparameter search over learning rates $\{10^{-2}, 10^{-3}, 10^{-4}\}$. We report the best performance across learning rates, averaged over three random seeds. To train UML, each modality is first projected into a shared embedding space of dimension $d \in \{10, 40, 150, 300\}$ via a learned linear transformation. The projected inputs are processed by a shared 5-layer, 5-head autoregressive Transformer encoder,

Table 4: Sample text descriptions per class for Oxford Pets dataset

| Class | Examples |
|---|---|
| Wheaten Terrier | A wheaten terrier is a small, shaggy dog with a soft, silky coat. |
| | A wheaten terrier has a soft, wheat-colored coat that is low-shedding and hypoallergenic. |
| | The wheaten terrier is a medium-sized, hypoallergenic dog breed. |
| | A pet Wheaten Terrier usually has an intelligent expression and a soft, wheat-colored coat. |
| Great Pyrenees | A great pyrenees is a large, white, shaggy-coated dog. |
| | A Great Pyrenees is a large, fluffy dog with a calm, gentle disposition. |
| | The great pyrenees was originally bred to protect livestock from predators. |
| | Great Pyrenees are known for being very large, white dogs with thick fur. |
| Sphynx | A pet Sphynx typically has a small, wrinkled head and a hairless body. |
| | A Sphynx is a hairless cat breed known for its soft, warm skin. |
| | A Sphynx often displays large ears, pronounced cheekbones, and no fur. |
| | Sphynx are unique cats characterized by their lack of coat and wrinkled skin. |
| Birman | A Birman is a long-haired, color-pointed cat with a "mask" of darker fur on its face. |
| | A Birman has silky, pale cream to ivory fur with deep seal- or lilac-colored points. |
| | Birman cats possess striking blue eyes and contrasting white "gloves" on their paws. |
| | They are known for being gentle, affectionate, and smooth-coated companions. |
| Pomeranian | A Pomeranian is a small, fluffy dog with a thick double coat. |
| | Pomeranians are toy-sized, alert dogs with fox-like faces and plumed tails. |
| | A pet Pomeranian often comes in orange, black, white, or mixed coat colors. |
| | They are lively, outgoing, and known for their bold, friendly personalities. |

followed by a linear projection back to the original modality dimension. Training uses a next-token/patch embedding prediction objective, with both modalities sharing the Transformer backbone to encourage cross-modal synergies in the latent space. At inference time, we average the Transformer outputs across sequence length and use the resulting embeddings for linear probing on downstream classification tasks. All models (Unimodal baseline and UML) are trained for 100 epochs, with hyperparameter search over learning rates $\{10^{-2}, 10^{-3}, 10^{-4}\}$. For UML, in addition, we perform hyperparameter search over a curriculum parameter $\texttt{step} \in \{0, 30, 50, 70\}$. This parameter controls whether training begins on $X$ alone for the first $\texttt{step}$ epochs before switching to joint training, with $\texttt{step} = 0$ corresponding to joint training from the start. For each dataset, we select the best-performing model on the validation set and report test accuracy averaged over three random seeds.

**Standard Vision-Text Benchmarks.** We extract image and text embeddings using ViT-B/14 DINOv2 and OpenLLaMA-3B, respectively. As in MultiBench, patch and token embeddings are projected to a shared 256-dimensional space via modality-specific linear layers. A 4-layer, 4-head transformer serves as the shared encoder, with outputs projected back to the original embedding dimensions using modality-specific linear projections. We perform rigorous hyperparameter tuning for both the unimodal baseline and UML, and report average test accuracy of the best model across three seeds. Full hyperparameter ranges are listed in Table 5.

### B.4.2 Image Classification using Image and Unpaired Texts

For text, we use OpenLLaMA-3B as our default encoder and ablate against BERT-Large, RoBERTa-Large, GPT-2 Large, and the pre-aligned CLIP text encoder, keeping the text encoder frozen. For images, our main backbone is ViT-S/14 DINOv2, with ablations across other DINOv2 variants and the CLIP vision encoder. In the linear-probe setting, all encoder weights stay fixed and we train only a single linear classification head; in full fine-tuning, we jointly update the image backbone and that head, while still freezing the text encoder.

We optimize cross-entropy loss via AdamW (Loshchilov & Hutter, 2017) and perform an extensive grid search over learning rate, weight decay, cosine learning rate scheduling with linear warmup, dropout, and a learnable, modality-specific scaling on the logits. The results are reported for the

best-performing model on the validation dataset. We report results for the model achieving highest validation accuracy; the full hyperparameter ranges are in Table 5.

For full fine-tuning, we jointly update the image backbone and classification head with a fixed learning rate of $5 \times 10^{-5}$, batch size 64, and omit learnable modality-specific scaling, since it showed no benefit in this setting.

Table 5: Hyperparameter grid for linear probing.

| Hyperparameter | Values |
|---|---|
| Optimizer | adamw |
| Learning rate | {0.001, 1e-4} |
| Weight decay | {0.0, 0.01, 0.001} |
| LR scheduler | cosine |
| Batch size | {8, 32} |
| Max iterations | 12,800 |
| Warmup iterations | 50 |
| Warmup type | linear |
| Warmup min LR | 1e-5 |
| Dropout | {0.0} |
| Modality-specific learnable scaling | {False, True} |
| Early-stop patience | 10 |

### B.4.3 EVALUATION ON IMAGENET-ESC

Similar to our vision-language experiments, we perform few-shot evaluation using the 5-fold splits defined in the benchmark. Each fold contains 8 samples per class, with one fold used for training and validation and the remaining four for testing. We repeat the process over 5 random splits and report the average performance. For audio encoding, we use AudioCLIP with an ES-ResNeXT backbone (Guzhov et al., 2021). AudioCLIP is pretrained on AudioSet and generates audio embeddings in the same representation space as CLIP. Following the instructions in (Guzhov et al., 2021; Lin et al., 2023), we use train() mode in Pytorch to extract the features since eval() mode yields suboptimal embeddings. We evaluate our models on two tasks—audio classification and image classification—comparing the unimodal baseline against two multimodal variants in which the primary modality is each time augmented by one of the other modalities.

### B.4.4 TRANSFER LEARNING FROM LANGUAGE TO VISION

To adapt a language model to image classification, we embed image patches using a linear projection and add positional encodings to capture spatial structure. We then use transformer layers initialized from pretrained BERT, and finally, a 2-layer MLP classification head. Specifically, we split each image of size $224 \times 224$ into patches of size $16 \times 16$ with 196 patch tokens. Each patch is then projected into the model's embedding space of dimension $d$ (e.g. d=768 for GPT-2, $d = 1024$ for BERT) via a learned linear layer. We then prepend a learnable "[CLS]" token, add learned positional embeddings of shape $(N + 1) \times d$, and apply dropout with probability $p = 0.1$. This $(N + 1) \times d$ sequence is passed into the pretrained transformer stack (either GPT-2 or BERT), using a full bidirectional attention mask over all patch tokens and the CLS token. We extract the final hidden state corresponding to the CLS token and feed it through a two-layer MLP classification head.

During training, we evaluate two scenarios: 1) one where the pretrained backbone is frozen and only the patch embedding and linear head are trained, and 2) another where the backbone is initially frozen to align the trainable layers (patch embedding and head) with the pretrained language backbone, and then unfrozen after 2000 steps for end-to-end training. This approach allows us to test whether the semantic richness captured by language models provides a strong initialization, leading to better convergence and performance compared to training ViT from scratch.

## C  PROOFS OF THEORETICAL RESULTS

In this section, we present complete derivations and proofs of the main theoretical claims. Appendix C.1 gathers all definitions and background required for our arguments. Appendix C.2 formalizes the linear data-generating model, derives closed-form maximum-likelihood estimators for each modality and their joint estimator, and computes the corresponding block-wise Fisher information. Finally, Appendix C.3 provides the detailed proofs of our variance-reduction claims, showing rigorously how unpaired multimodal estimation strictly lowers estimator variance.

### C.1  BACKGROUND AND DEFINITIONS

In this section we revisit the mathematical definitions used in our theoretical analysis, including matrix-orderings, characterization of symmetric matrices and Fisher information.

**Definition 1** (Positive Semidefinite Matrix). A real symmetric matrix $A \in \mathbb{R}^{d \times d}$ is *positive semidefinite* if for all vectors $v \in \mathbb{R}^d$, $v^\top A\, v \geq 0$. Equivalently, all eigenvalues of $A$ are nonnegative. We denote the set of all $d \times d$ symmetric, positive-semidefinite matrices as $S^d_{\succeq 0}$.

**Definition 2** (Positive Definite Matrix). A real symmetric matrix $A \in \mathbb{R}^{d \times d}$ is *positive definite* if for every nonzero $v \in \mathbb{R}^d$, $v^\top A\, v > 0$. Equivalently, all eigenvalues of $A$ are strictly positive. We denote the set of all $d \times d$ symmetric, positive definite matrices as $S^d_{\succ 0}$.

**Definition 3** (Loewner Order). For two real symmetric matrices $A, B \in \mathbb{R}^{d \times d}$, we write $A \preceq B \iff B - A$ is positive semidefinite and $A \prec B \iff B - A$ is positive definite. This defines a partial order on the cone of symmetric matrices.

**Definition 4** (Fisher Information Matrix). Given a parametric family of densities $p(x; \theta)$ on data $x$, the *Fisher information matrix* at parameter $\theta$ is

$$I(\theta) = \mathbb{E}_{x \sim p(\cdot; \theta)} \big[ \nabla_\theta \log p(x; \theta)\, \nabla_\theta \log p(x; \theta)^\top \big].$$

Equivalently, for regular models, $I(\theta) = -\mathbb{E}\big[ \nabla^2_\theta \log p(x; \theta) \big]$.

### C.2  MAXIMUM LIKELIHOOD ESTIMATORS AND FISHER CONTRIBUTIONS

In this section we revisit our linear data–generating model, introduce notations for the $X$–only, $Y$–only and joint likelihoods, derive the closed-form MLEs $\widehat{\theta}_X$, $\widehat{\theta}_Y$ and $\widehat{\theta}_{X,Y}$, and formalize their information contributions towards estimating the ground truth parameters $\theta \equiv [\theta_c, \theta_x, \theta_y]^\top$.

**Data Generating Process.** Recall our linear data-generating process: Assume that all factors of variation in reality live in a single $d$-dimensional space $\mathcal{Z}^* \equiv \theta \in \mathbb{R}^d$ modeled using a linear data-generating pipeline. This parameter can further be decomposed as $\theta \equiv [\theta_c, \theta_x, \theta_y]^\top$ where $\theta_c \in \mathbb{R}^{d_c}, \theta_x \in \mathbb{R}^{d_x}, \theta_y \in \mathbb{R}^{d_y}$ and $d_c + d_x + d_y = d$. Here, $\theta_c$ captures the *common* (shared) parameters that affect both modalities, $\theta_x$ denotes the parameters that only affect modality $X$, and $\theta_y$ denotes the parameters that only affect modality $Y$. We observe two independent datasets, one from each modality $\{X_i\}_{i=1}^{N_x} \in \mathbb{R}^m$ and $\{Y_j\}_{j=1}^{N_y} \in \mathbb{R}^n$, each reflecting partial measurements of the ground truth latent space $\mathcal{Z}^*$:

$$X_i = A_{c,i}\, \theta_c + A_{x,i}\, \theta_x + \epsilon_{X,i}, \quad \epsilon_{X,i} \sim \mathcal{N}\big(0, \sigma_x^2 I_{m_i}\big) \tag{3}$$

$$Y_j = B_{c,j}\, \theta_c + B_{y,j}\, \theta_y + \epsilon_{Y,j}, \quad \epsilon_{Y,j} \sim \mathcal{N}\big(0, \sigma_y^2 I_{n_j}\big). \tag{4}$$

Here, $A_{c,i}, A_{x,i}, B_{c,j}, B_{y,j}$ are known design blocks capturing how each sample probes the latent factors and $\varepsilon_{X,i}, \varepsilon_{Y,j}$ represent the independent measurement noise.

In our linear setting, estimating the true latent state $\theta$—and hence the underlying reality $\mathcal{Z}^*$—is governed by the Fisher information matrix $I(\theta) = -\mathbb{E}\big[ \nabla^2_\theta \ell(\theta) \big]$, which measures how sharply the likelihood "curves" around the true $\theta$. High curvature along a particular axis means the data tightly constrain that component, driving down estimator variance there.

**Unimodal Estimators.** We first estimate $\theta$ using only the $X$–dataset. Stacking $\{X_i\}_{i=1}^{N_x}$ yields a design matrix $\mathcal{A}$ with block rows $[A_{c,i},\ A_{x,i},\ 0]$. The least-squares solution

$$\widehat{\theta}_X = \arg\min_{\theta} \sum_{i=1}^{N_x} \left\| X_i - A_{c,i}\,\theta_c - A_{x,i}\,\theta_x \right\|^2$$

omits $\theta_y$ entirely. Consequently, the Fisher information on $\theta_y$ vanishes, making it unidentifiable.

Analogously, stacking $\{Y_j\}_{j=1}^{N_y}$ defines $\mathcal{B}$ with block rows $[B_{c,j},\ 0,\ B_{y,j}]$ and yields

$$\widehat{\theta}_Y = \arg\min_{\theta} \sum_{j=1}^{N_y} \left\| Y_j - B_{c,j}\,\theta_c - B_{y,j}\,\theta_y \right\|^2.$$

This estimator doesn't depend on $\theta_x$, providing zero coverage for that component. Thus, each unimodal estimator entirely fails to recover the parameters exclusive to the omitted modality.

**Multimodal Estimators.** Despite the lack of one-to-one pairing, both $\{X_i\}$ and $\{Y_j\}$ share the common parameters $\theta_c$. Since the two distributions are conditionally independent, the joint likelihood factorizes as

$$\prod_{i=1}^{N_x} p(X_i \mid \theta_c, \theta_x) \ \times\ \prod_{j=1}^{N_y} p(Y_j \mid \theta_c, \theta_y).$$

Maximizing this yields the combined estimator

$$\widehat{\theta}_{X,Y} = \arg\min_{\theta_c,\theta_x,\theta_y} \left\{ \sum_{i=1}^{N_x} \| X_i - A_{c,i}\,\theta_c - A_{x,i}\,\theta_x \|^2 \ +\ \sum_{j=1}^{N_y} \| Y_j - B_{c,j}\,\theta_c - B_{y,j}\,\theta_y \|^2 \right\}.$$

Intuitively, there is no requirement to match up individual $(X_i, Y_j)$ pairs. Instead, the estimate for $\theta_c$ is improved by both modalities while remaining unpaired.

**Fisher Information.** In our linear model, each dataset contributes block-structured Fisher information. For the $X$–dataset:

$$I_X = \sum_{i=1}^{N_x} \begin{pmatrix} A_{c,i}^\top A_{c,i} & A_{c,i}^\top A_{x,i} & 0 \\ A_{x,i}^\top A_{c,i} & A_{x,i}^\top A_{x,i} & 0 \\ 0 & 0 & 0 \end{pmatrix},$$

and for the $Y$–dataset:

$$I_Y = \sum_{j=1}^{N_y} \begin{pmatrix} B_{c,j}^\top B_{c,j} & 0 & B_{c,j}^\top B_{y,j} \\ 0 & 0 & 0 \\ B_{y,j}^\top B_{c,j} & 0 & B_{y,j}^\top B_{y,j} \end{pmatrix}.$$

Because $X$ and $Y$ samples are independent, their curvature contributions add pointwise, resulting in the joint Fisher information being simply the sum of the unimodal blocks.

$$I_{X,Y} = I_X + I_Y = \begin{pmatrix} \sum_i A_{c,i}^\top A_{c,i} + \sum_j B_{c,j}^\top B_{c,j} & * & * \\ * & \sum_i A_{x,i}^\top A_{x,i} & 0 \\ * & 0 & \sum_j B_{y,j}^\top B_{y,j} \end{pmatrix},$$

where "$*$" denotes the cross-modal blocks. In particular, we have the shared-parameter block as

$$(I_{X,Y})_{\theta_c,\theta_c} = \sum_{i=1}^{N_x} A_{c,i}^\top A_{c,i} + \sum_{j=1}^{N_y} B_{c,j}^\top B_{c,j},$$

For clarity of exposition, we assume that the parameterisation is chosen such that the common factors $\theta_c$ are Fisher-orthogonal to the modality-specific factors $\theta_x$ and $\theta_y$ i.e. the cross Fisher blocks vanish. Under this assumption, the $\theta_c$ block of the Fisher matrix inverts blockwise to yield the covariance on $\theta_c$.

### C.3 Theorems and Proofs

The aim of this section is to detail the proofs of the theoretical results presented in the main manuscript. The key theoretical tools driving our analysis are already prepared in Appendix C.1 and Appendix C.2. Core to our theoretical analysis are a few lemmas around the Loewner-order monotonicity result for inverses that we prove below.

**Lemma 1** (Loewner Order reversal for inverses). *Let $M, N \in \mathbb{S}^d_{\succ 0}$ with $M \prec N$ (or $M \preceq N$). Then $N^{-1} \prec M^{-1}$ (or $N^{-1} \preceq M^{-1}$).*

*Proof.* Since $N \succ 0$, $N^{-1/2}$ exists and is nonsingular. Define $C := N^{-1/2}MN^{-1/2} \prec I$. Because a congruence with an invertible matrix preserves positive-definiteness, $C \succ 0$; hence $C^{-1}$ is well defined and $C^{-1} \succ I$ (the scalar map $x \mapsto x^{-1}$ is strictly decreasing on $(0, \infty)$). Undoing the congruence gives

$$M^{-1} = N^{-1/2}C^{-1}N^{-1/2} \succ N^{-1/2}IN^{-1/2} = N^{-1}. \qquad \square$$

**Lemma 2** (Inverse–monotonicity of the Moore–Penrose pseudoinverse). *Let $M, N \in \mathbb{S}^d_{\succeq 0}$ satisfy $M \prec N$ and $\ker M = \ker N =: K$. Then their pseudoinverses obey $N^\dagger \prec M^\dagger$.*

*Proof.* Set $S := K^\perp$ and let $P := P_S$ be the orthogonal projector onto $S$. Because $M$ and $N$ vanish on $K$, we have the decompositions $M = PMP$ and $N = PNP$. Restricted to $S$ both matrices are positive–definite:

$$\tilde{M} := PMP, \qquad \tilde{N} := PNP \in \mathbb{S}^{\dim S}_{\succ 0}, \qquad \tilde{M} \prec \tilde{N}.$$

Apply Lemma 1 to $\tilde{M}, \tilde{N}$ to obtain $\tilde{N}^{-1} \prec \tilde{M}^{-1}$ on $S$. The Moore–Penrose pseudoinverse equals the ordinary inverse on $S$ and is zero on $K$:

$$M^\dagger = P\tilde{M}^{-1}P, \qquad N^\dagger = P\tilde{N}^{-1}P.$$

Therefore $N^\dagger = P\tilde{N}^{-1}P \prec P\tilde{M}^{-1}P = M^\dagger$. $\qquad \square$

**Lemma 3** (Directional Loewner Order reversal). *Let $M, N \in \mathbb{S}^d_{\succ 0}$ with $M \preceq N$. If a non-zero vector $v$ satisfies $v^\top Mv < v^\top Nv$, then*

1. *For the vector $v$, it holds that $v^\top M^{-1}v \geq v^\top N^{-1}v$, with strict inequality $v^\top M^{-1}v > v^\top N^{-1}v$ if and only if $(N - M)M^{-1}v \neq 0$.*

2. *There exists a non-zero vector $u \in \mathbb{R}^d$ such that $u^\top M^{-1}u > u^\top N^{-1}u$.*

*Proof.* Denote the Loewner gap $\Delta := N - M \succeq 0$. Then, the assumption $v^\top Nv > v^\top Mv$ is equivalent to $v^\top \Delta v > 0$. Introduce the congruence–invariant normalisation $C := M^{-1/2}\Delta M^{-1/2} \succeq 0$. Now, using $\Delta = M^{1/2}CM^{1/2}$ and properties of inverse,

$$N = M^{1/2}(I + C)M^{1/2}, \qquad N^{-1} = M^{-1/2}(I + C)^{-1}M^{-1/2},$$

since $I + C \succ 0$ (because $C \succeq 0$ and $I \succ 0$). Thus,

$$\begin{aligned} M^{-1} - N^{-1} &= M^{-1/2}\big[I - (I + C)^{-1}\big]M^{-1/2} \\ &= M^{-1/2}C(I + C)^{-1}M^{-1/2}, \end{aligned}$$

because $(I - (I + C)^{-1})(I + C) = C$.

**(1) Directional inequality and strictness.**
Evaluating in the direction $v$, we have,

$$\begin{aligned} v^\top(M^{-1} - N^{-1})v &= v^\top M^{-1/2}(I + C)^{-1}CM^{-1/2}v \\ &= u^\top(I + C)^{-1}Cu \geq 0 \quad \text{(where } u = M^{-1/2}v) \end{aligned}$$

since $(I + C)^{-1} \in \mathbb{S}_{\succ 0}$ and $C \in \mathbb{S}_{\succeq 0}$. This proves that $v^\top M^{-1}v \geq v^\top N^{-1}v$. The inequality is strict if and only if $u \notin \ker(C)$. Since $\ker((I + C)^{-1}C) = \ker(C(I + C)^{-1}) = \ker(C)$, this is equivalent to $Cu \neq 0$. This can be further written as

$$Cu = M^{-1/2}\Delta M^{-1/2}u = M^{-1/2}\Delta M^{-1}v$$

As $M^{-1/2}$ is invertible, this holds if and only if $\Delta M^{-1}v \neq 0$, which proves 1.

**(2) Existence of a direction with strict reversal.**
Now, from the premise $v^\top \Delta v > 0$, it follows that $\Delta \neq 0$. Since $M \succ 0$, $M^{-1/2}$ is invertible, $C$ is also not the zero matrix. Since $C \succeq 0$, this means that $C$ must have at least one strictly positive eigenvalue. Let $\lambda > 0$ be such an eigenvalue, and let $z \neq 0$ be a corresponding eigenvector. Define, $x := M^{1/2}z \neq 0$. Thus, we have $x^\top(M^{-1} - N^{-1})x = z^\top C(I+C)^{-1}z = \frac{\lambda}{1+\lambda}\|z\|^2 > 0$, showing the existence of a non-zero vector $x$ such that $x^\top M^{-1}x > x^\top N^{-1}x$. $\qquad\square$

**Lemma 4** (Fisher information is positive semidefinite). *Let $\{p(x;\theta) : \theta \in \Theta \subset \mathbb{R}^d\}$ be a regular parametric family with score $S(\theta) := \nabla_\theta \log p(X;\theta)$, and Fisher information $I(\theta) := \mathbb{E}[S(\theta)S(\theta)^\top]$. Then $v^\top I(\theta)v \geq 0$ or equivalently $I(\theta) \in \mathbb{S}_{\succeq 0}^d$*

*Proof.* For any $v \in \mathbb{R}^d$,

$$v^\top I(\theta)v = v^\top \mathbb{E}[S(\theta)S(\theta)^\top]v = \mathbb{E}[(v^\top S(\theta))^2] \geq 0.$$

Thus $I(\theta)$ is positive semidefinite. $\qquad\square$

**Theorem 1** (Restatement of Theorem 1). *Let $\hat{\theta}_X, \hat{\theta}_Y$ be the least-squares estimators for $\theta$ using only $\{X_i\}$ and only $\{Y_j\}$ and let $\hat{\theta}_{X,Y}$ be the joint estimator using both unpaired datasets. Then, under the assumption that at least one $B_{c,j}$ where $j \in \{1,2,...N_y\}$ has full rank, the common-factor covariance satisfies the strict Loewner ordering i.e. $\mathrm{Var}(\hat{\theta}_{X,Y})_{\theta_c,\theta_c} \prec \mathrm{Var}(\hat{\theta}_X)_{\theta_c,\theta_c}$, or equivalently, the Fisher information on $\theta_c$ strictly increases when combining both modalities, despite not having sample-wise pairing: $(I_X + I_Y)_{\theta_c,\theta_c} \succ (I_X)_{\theta_c,\theta_c}$.*

*Proof.* Let $M_X = (I_X)_{\theta_c,\theta_c}$, $M_Y = (I_Y)_{\theta_c,\theta_c}$, and $M_{X,Y} = (I_{X,Y})_{\theta_c,\theta_c}$ be the Fisher information submatrices for the common parameters $\theta_c$. From the definitions provided and Lemma 4, we have $M_X = \sum_i A_{c,i}^\top A_{c,i} \succeq 0$ and $M_Y = \sum_j B_{c,j}^\top B_{c,j}$. By the theorem's assumption, at least one $B_{c,j}$ has full column rank, then for any $v \in \mathbb{R}^{d_c} \setminus \{0\}$, $v^\top B_{c,j}^\top B_{c,j}\,v = \|B_{c,j}v\|^2 > 0$. Now, since $M_Y$ is a sum of positive semidefinite matrices, at least one of which is positive definite, $M_Y$ must be positive definite. Thus, $M_Y \succ 0$.

Now since the joint Fisher information is $M_{X,Y} = M_X + M_Y$ and $M_Y \succ 0$, we have $M_{X,Y} - M_X = M_Y \succ 0$. This immediately proves the strict increase in Fisher information:

$$(I_X + I_Y)_{\theta_c,\theta_c} = M_{X,Y} \succ M_X = (I_X)_{\theta_c,\theta_c}.$$

In our Gaussian linear model, the least-squares estimators coincide with the maximum likelihood estimators, and their asymptotic covariance on $\theta_c$ is the inverse of the Fisher information, up to the usual $1/n$ scaling, which is common to all three estimators and irrelevant for comparisons. Thus, whenever $M_X$ is positive definite, $\mathrm{Var}(\hat{\theta}_X)_{\theta_c,\theta_c} = M_X^{-1}$ and $\mathrm{Var}(\hat{\theta}_{X,Y})_{\theta_c,\theta_c} = M_{X,Y}^{-1}$ Since $M_{X,Y} \succ M_X \succ 0$, from Lemma 1, we have $M_{X,Y}^{-1} \prec M_X^{-1}$, i.e.

$$\mathrm{Var}(\hat{\theta}_{X,Y})_{\theta_c,\theta_c} \prec \mathrm{Var}(\hat{\theta}_X)_{\theta_c,\theta_c}.$$

If $M_X$ is only positive semidefinite, interpret covariances via the Moore–Penrose pseudoinverse. Along directions $v \in \ker(M_X)$, the $X$-only estimator has infinite variance (no information), whereas $M_{X,Y} \succ 0$ yields a finite variance for the joint estimator. Hence adding the unpaired $Y$-modality strictly reduces the variance (or, equivalently, strictly increases the Fisher information) on the common factors $\theta_c$. $\qquad\square$

**Theorem 2** (Restatement of Theorem 2). *Let all notation be as in Theorem 1, and define $M_X := (I_X)_{\theta_c,\theta_c}$, $M_Y := (I_Y)_{\theta_c,\theta_c}$, and $M_{XY} := M_X + M_Y$. Let $v \in \mathbb{R}^{d_c} \setminus \{0\}$. If there exists at least one index $j \in \{1,2,...N_y\}$ such that $B_{c,j}v \neq 0$, then the following hold:*

    *1. The Fisher information strictly increases in direction $v$ i.e. $v^\top M_{XY}\,v > v^\top M_X v$.*

2. *The variance of the estimator in direction $v$ is strictly reduced i.e $v^\top \operatorname{Var}(\hat\theta_{X,Y})_{\theta_c,\theta_c} v <$ $v^\top \operatorname{Var}(\hat\theta_X)_{\theta_c,\theta_c} v$, if $v \notin \operatorname{range}(M_X)$. For $v \in \operatorname{range}(M_X)$, this strict inequality holds for $v$ under an additional invertibility condition and is always guaranteed for some $u \in \operatorname{range}(M_X)$ i.e. $\exists u$ s.t. $u^\top \operatorname{Var}(\hat\theta_{X,Y})_{\theta_c,\theta_c} u < u^\top \operatorname{Var}(\hat\theta_X)_{\theta_c,\theta_c} u$.*

*Proof.* By assumption, $\exists j$ such that $B_{c,j} v \neq 0$. Thus:

$$v^\top M_Y v = \sum_{j=1}^{N_y} \|B_{c,j} v\|^2 \geq \|B_{c,j} v\|^2 > 0.$$

Hence $M_Y$ is positive-definite in direction $v$, implying $M_{X,Y} \succ M_X$ in this direction, thus proving the first part of the theorem:

$$v^\top M_{XY} v = v^\top M_X v + v^\top M_Y v > v^\top M_X v,$$

For part (2), we break the analysis into two cases:

**Case 1:** $v \notin \operatorname{Range}(M_X)$. Then decompose $v = v_S + v_K$, where $v_S \in \operatorname{Range}(M_X)$ and $v_K \in \ker(M_X)$ with $v_K \neq 0$. The linear combination of parameters $v^\top \theta_c = v_S^\top \theta_c + v_K^\top \theta_c$. Since $v_K \in \ker(M_X)$, the component $v_K^\top \theta_c$ is not identifiable by the $X$-only model. Consequently, the asymptotic variance of an unbiased estimator for $v^\top \theta_c$ using only the $X$-dataset is infinite. We denote this as $v^\top \operatorname{Var}(\hat\theta_X)_{\theta_c,\theta_c} v = \infty$.

On the other hand, we already showed that $v^\top M_{XY} v > 0$, so $v \notin \ker(M_{XY})$. For a symmetric positive semidefinite matrix $M_{XY}$, its Moore–Penrose pseudoinverse $M_{XY}^\dagger$ is also symmetric positive semidefinite and satisfies $\ker(M_{XY}^\dagger) = \ker(M_{XY})$. Hence $v \notin \ker(M_{XY}^\dagger)$ and

$$0 < v^\top M_{XY}^\dagger v < \infty.$$

Since the asymptotic covariance of $\hat\theta_{X,Y}$ in direction $v$ is governed by $M_{XY}^\dagger$, we obtain

$$0 < v^\top \operatorname{Var}(\hat\theta_{X,Y})_{\theta_c,\theta_c} v < \infty = v^\top \operatorname{Var}(\hat\theta_X)_{\theta_c,\theta_c} v,$$

showing strict variance reduction along $v$ in this case.

**Case 2:** $v \in \operatorname{Range}(M_X)$. Let $S := \operatorname{Range}(M_X)$ and let $P_S$ be the orthogonal projector onto $S$. Because $M_X = M_X P_S$ and $M_{XY} = M_X + M_Y$, the restrictions

$$\tilde M_X := P_S M_X P_S, \qquad \tilde M_{XY} := P_S M_{XY} P_S = \tilde M_X + P_S M_Y P_S$$

are *positive-definite* on $S$; To see this, take any non-zero $w \in S$. Since $w \in \operatorname{range}(M_X)$, $P_S w = w$; hence

$$w^\top \tilde M_X w = w^\top M_X w > 0 \quad (P_S \text{ is identity when restricted to } S)$$

Thus $\tilde M_X \succ 0$ on $S$. Because $P_S M_Y P_S \succeq 0$, adding it preserves positive-definiteness, so

$$\tilde M_{XY} = \tilde M_X + P_S M_Y P_S \succeq \tilde M_X \succ 0 \quad \text{on } S.$$

Applying [Lemma 3](1) to $\tilde M_X$ and $\tilde M_{XY}$ on $S$ gives us $v^\top \tilde M_{XY}^{-1} v \leq v^\top \tilde M_X^{-1} v$. Strict inequality $v^\top \tilde M_{XY}^{-1} v < v^\top \tilde M_X^{-1} v$ holds if and only if the condition $C_v := ((\tilde M_{XY} - \tilde M_X)\tilde M_X^{-1} v \neq \mathbf{0})$ is met. Therefore, if condition $C_v$ holds, the directional variance along this constrained space $S$ is strictly reduced:

$$v^\top \operatorname{Var}(\hat\theta_{X,Y})_{\theta_c,\theta_c} v = v^\top \tilde M_{XY}^{-1} v < v^\top \tilde M_X^{-1} v = v^\top \operatorname{Var}(\hat\theta_X)_{\theta_c,\theta_c} v[1].$$

Further, from [Lemma 3](2), there exists some non-zero vector $u \in S$ such that $u^\top \tilde M_{XY}^{-1} u < u^\top \tilde M_X^{-1} u$. Thus we have,

$$u^\top \operatorname{Var}(\hat\theta_{X,Y})_{\theta_c,\theta_c} u < u^\top \operatorname{Var}(\hat\theta_X)_{\theta_c,\theta_c} u.$$

Thus, completing the proof.

$\square$

---

[1]We note that true asymptotic variance defined as $v^\top \operatorname{Var}(\hat\theta_{X,Y})_{\theta_c,\theta_c} v = v^\top M_{XY}^\dagger v$, $v^\top M_{XY}^\dagger v = v^\top \tilde M_{XY}^{-1} v$ if $S$ is an invariant subspace of $M_{XY}$ and $M_{XY}$ is block-diagonal with respect to $S$ and $S^\perp$ (i.e., $P_S M_{XY} P_{S^\perp} = \mathbf{0}$, which implies $P_S M_Y P_{S^\perp} = \mathbf{0}$).

**Corollary 1.** *Assume a direction $v \in \mathbb{R}^{d_c} \setminus \{0\}$ with $a = v^\top (I_X)_{\theta_c, \theta_c} v > 0$ and $b = v^\top (I_Y)_{\theta_c, \theta_c} v > 0$ where $v$ is the common eigenvector of $(I_X)_{\theta_c, \theta_c}$ and $(I_Y)_{\theta_c, \theta_c}$. Then the variance in direction $v$ contracts by the factor*

$$\frac{v^\top \mathrm{Var}(\hat{\theta}_{X,Y}) \, v}{v^\top \mathrm{Var}(\hat{\theta}_X) \, v} = \frac{1/(a+b)}{1/a} = \frac{a}{a+b} < 1,$$

*So the joint estimator achieves strictly lower error along $v$.*

*Proof.* Let $M_X = (I_X)_{\theta_c, \theta_c}$ and $M_Y = (I_Y)_{\theta_c, \theta_c}$. By assumption, $v$ is a common eigenvector of $M_X$ and $M_Y$. Thus, $M_X v = \lambda_X v$ and $M_Y v = \lambda_Y v$ for some eigenvalues $\lambda_X$ and $\lambda_Y$. From the assumptions, we have $\lambda_X = a/\|v\|^2 > 0$ and $\lambda_Y = b/\|v\|^2 > 0$. Since $M_X$ is symmetric and $M_X v = \lambda_X v$ with $\lambda_X > 0$, the pseudoinverse acts as $M_X^\dagger v = \lambda_X^{-1} v$. Therefore, the variance in direction $v$ for the $X$-only estimator is

$$v^\top \mathrm{Var}(\hat{\theta}_X)_{\theta_c, \theta_c} \, v = v^\top M_X^\dagger v = v^\top (\lambda_X^{-1} v) = \lambda_X^{-1} \|v\|^2 = a^{-1} \|v\|^4.$$

Since $v$ is a common eigenvector, it is also an eigenvector of $M_{XY} = M_X + M_Y$:

$$(M_X + M_Y) v = M_X v + M_Y v = \lambda_X v + \lambda_Y v = (\lambda_X + \lambda_Y) v.$$

The corresponding eigenvalue is $\lambda_{XY} = \lambda_X + \lambda_Y$. Since $\lambda_X > 0$ and $\lambda_Y > 0$, $\lambda_{XY} > 0$. Thus, $(M_X + M_Y)^\dagger v = (\lambda_X + \lambda_Y)^{-1} v$. The variance in direction $v$ for the joint estimator is

$$v^\top \mathrm{Var}(\hat{\theta}_{X,Y})_{\theta_c, \theta_c} \, v = v^\top (M_X + M_Y)^\dagger v = (\lambda_X + \lambda_Y)^{-1} \|v\|^2 = (a+b)^{-1} \|v\|^4.$$

Now, we form the ratio of these variances:

$$\frac{v^\top \mathrm{Var}(\hat{\theta}_{X,Y})_{\theta_c, \theta_c} \, v}{v^\top \mathrm{Var}(\hat{\theta}_X)_{\theta_c, \theta_c} \, v} = \frac{\lambda_X}{\lambda_X + \lambda_Y} = \frac{a}{a+b} < 1.$$

$\square$

**Corollary 2.** *Assume a direction $v \in \mathbb{R}^{d_c} \setminus \{0\}$ with $v^\top (I_X)_{\theta_c, \theta_c} v = 0$ and $v^\top (I_Y)_{\theta_c, \theta_c} v > 0$. Then $v^\top \mathrm{Var}(\hat{\theta}_X) \, v = \infty$ and $v^\top \mathrm{Var}(\hat{\theta}_{X,Y}) \, v < \infty$ i.e. a direction unidentifiable from $X$ alone becomes well-posed with even unpaired data from $Y$.*

*Proof.* This corollary follows directly from Case 1 of Theorem 2. The condition $v^\top (I_X)_{\theta_c, \theta_c} v = 0$ for $v \neq 0$ implies $v \in \ker((I_X)_{\theta_c, \theta_c})$, and thus $v \notin \mathrm{range}((I_X)_{\theta_c, \theta_c})$. Given the additional condition $v^\top (I_Y)_{\theta_c, \theta_c} v > 0$, the conclusions of Case 1 of the theorem apply directly. $\square$

**Corollary 3** (Variance Reduction for Eigenvectors of $M_X$). *Let $v \in \mathbb{R}^{d_c} \setminus \{0\}$ be an eigenvector of $M_X = (I_X)_{\theta_c, \theta_c}$ with a corresponding eigenvalue $\lambda_X > 0$. If the $Y$-dataset provides information in this direction $v$ (i.e., $v^\top M_Y v > 0$, where $M_Y = (I_Y)_{\theta_c, \theta_c}$), then the variance in direction $v$ is strictly reduced by incorporating the $Y$-dataset:*

$$v^\top \mathrm{Var}(\hat{\theta}_{X,Y})_{\theta_c, \theta_c} \, v < v^\top \mathrm{Var}(\hat{\theta}_X)_{\theta_c, \theta_c} \, v.$$

*Specifically, $v^\top \mathrm{Var}(\hat{\theta}_X)_{\theta_c, \theta_c} v = \lambda_X^{-1} \|v\|^2$.*

*Proof.* Let $M_X = (I_X)_{\theta_c, \theta_c}$ and $M_Y = (I_Y)_{\theta_c, \theta_c}$. Since $v$ is an eigenvector of $M_X$ with a positive eigenvalue $\lambda_X > 0$, it follows that $v \in \mathrm{Range}(M_X)$. Let $S = \mathrm{Range}(M_X)$. The variance using only the $X$-dataset in direction $v$ is given by

$$v^\top \mathrm{Var}(\hat{\theta}_X)_{\theta_c, \theta_c} v = v^\top M_X^\dagger v.$$

Because $v$ is an eigenvector of $M_X$ with $\lambda_X > 0$, $M_X^\dagger v = \lambda_X^{-1} v$. Thus,

$$v^\top \mathrm{Var}(\hat{\theta}_X)_{\theta_c, \theta_c} v = v^\top (\lambda_X^{-1} v) = \lambda_X^{-1} \|v\|^2.$$

This scenario falls under Case 2 of Theorem 2, specifically its conclusion regarding $v \in S$. According to that theorem, strict variance reduction $v^\top \mathrm{Var}(\hat{\theta}_{X,Y})_{\theta_c, \theta_c} v < v^\top \mathrm{Var}(\hat{\theta}_X)_{\theta_c, \theta_c} v$ occurs if the

condition $C_v = ((P_S M_Y P_S)(M_X|_S)^{-1} v \neq \mathbf{0})$ holds. Here, $P_S$ is the orthogonal projector onto $S$, and $M_X|_S$ is the restriction of $M_X$ to $S$, so $(M_X|_S)^{-1} v = \lambda_X^{-1} v$.

The condition $C_v$ thus becomes $(P_S M_Y P_S)(\lambda_X^{-1} v) \neq \mathbf{0}$. Since $\lambda_X > 0$, this is equivalent to $P_S M_Y P_S v \neq \mathbf{0}$. We are given that $v^\top M_Y v > 0$. As $v \in S$, $P_S v = v$. Therefore, $v^\top M_Y v = v^\top P_S M_Y P_S v > 0$. Let $A_S = P_S M_Y P_S$ restricted to $S$. $A_S$ is a positive semidefinite operator on $S$. The condition $v^\top A_S v > 0$ for $v \in S, v \neq 0$ implies that $A_S v \neq \mathbf{0}$ (because if $A_S v = \mathbf{0}$, then $v^\top A_S v = 0$, which contradicts $v^\top A_S v > 0$). Thus, $P_S M_Y P_S v \neq \mathbf{0}$, which means the condition $C_v$ is satisfied.

Since $v \in S$ and the condition $C_v$ for strict inequality is met, by Theorem 2, it follows that $v^\top \mathrm{Var}(\hat{\theta}_{X,Y})_{\theta_c,\theta_c} v < v^\top \mathrm{Var}(\hat{\theta}_X)_{\theta_c,\theta_c} v$. $\qquad\square$

**Theorem 3** (Restatement of Theorem 3). *Define for any $m$, $I_X^{(m)} = \sum_{i=1}^m A_{c,i}^\top A_{c,i}$ and $I_Y^{(m)} = \sum_{j=1}^m B_{c,j}^\top B_{c,j}$. If $\mathrm{range}\big(I_Y^{(m)}\big) \not\subseteq \mathrm{range}\big(I_X^{(m)}\big)$, then there exists a nonzero $v \in \mathbb{R}^{d_c}$ such that $v^\top I_Y^{(m)} v > v^\top I_X^{(m)} v$.*

*Proof.* Let $R_X := \mathrm{range}\big(I_X^{(m)}\big)$, $R_Y := \mathrm{range}\big(I_Y^{(m)}\big)$. By the assumption $R_Y \not\subseteq R_X$, choose a vector $w \in R_Y \setminus R_X$. Since $\mathbb{R}^{d_c}$ is a finite dimensional inner product space and $R_X$ is its finite dimensional subspace, we can decompose $w = w_\| + v$ with $w_\| \in R_X$ and $v \in R_X^\perp$. Because $w \notin R_X$, the orthogonal component $v$ is non-zero.

*(i) Term from $I_X^{(m)}$.* From the *Fundamental Theorem of Linear Algebra*, for any symmetric matrix $S$, $\ker S = \mathrm{range}(S)^\perp$; hence $R_X^\perp = \ker I_X^{(m)}$. Thus, $v^\top I_X^{(m)} v = 0$.

*(ii) Term from $I_Y^{(m)}$.* Because $w \in R_Y = \mathrm{range}\big(I_Y^{(m)}\big)$, there exists $u$ with $w = I_Y^{(m)} u$. Suppose, for contradiction, that $I_Y^{(m)} v = 0$. Then $v \in \ker I_Y^{(m)} = R_Y^\perp$, so $v \perp w$. But $w \cdot v = (w_\| + v) \cdot v = w_\| \cdot v + \|v\|^2 = \|v\|^2 > 0$ because $v \perp w_\|$ while $v \neq 0$. This contradicts $v \perp w$; therefore $I_Y^{(m)} v \neq 0$ and, by positive semidefiniteness,
$$v^\top I_Y^{(m)} v > 0.$$

Combining the above inequalities yields $v^\top I_Y^{(m)} v > v^\top I_X^{(m)} v$, with $v \neq 0$, which is the desired inequality. $\qquad\square$

### C.4 CONNECTION TO SHARING PARAMETERS ACROSS MODALITIES

The results in Section 3.1 establish that, in the linear model, the two modalities contribute additively to the Fisher information associated with the shared latent block $\theta_c$. This additivity arises solely from the structural fact that both observation channels depend on a common set of shared latents. To connect this to UML, we consider a likelihood model in which the parameters are partitioned into a shared block $\phi_c$ and modality-specific blocks, and the two modalities are conditionally independent given these parameters. Proposition 1 shows that, in any such model, the Fisher information on $\phi_c$ is exactly the sum of the unimodal Fisher blocks. Thus, the shared trunk in UML implements the same structural condition: the parameters $\phi_c$ are common to both losses, and therefore receive curvature contributions from both modalities. In this sense, weight sharing in UML is the architectural mechanism that reproduces the shared-parameter structure responsible for Fisher-information addition in our linearized theory. Below, we now formally state this result.

**Proposition 1** (Fisher Information Additivity for Shared Parameters). *Let $\phi_c \in \mathbb{R}^{d_c}$ be a shared parameter block and let $\phi_x, \phi_y$ be modality-specific parameters. Suppose*

$$p(x, y \mid \phi_c, \phi_x, \phi_y) = p_X(x \mid \phi_c, \phi_x)\, p_Y(y \mid \phi_c, \phi_y), \tag{5}$$

*so that $X$ and $Y$ are independent given $(\phi_c, \phi_x, \phi_y)$. Let $\{X_i\}_{i=1}^{N_x}$ and $\{Y_j\}_{j=1}^{N_y}$ be i.i.d. samples from the induced marginals. Denote by $\ell_X(\phi_c)$, $\ell_Y(\phi_c)$ and $\ell(\phi_c)$ the log-likelihoods in $\phi_c$ for the $X$-only,*

$Y$-only and joint experiments, respectively, and by $I_X(\phi_c)$, $I_Y(\phi_c)$ and $I_{X,Y}(\phi_c)$ the corresponding Fisher information matrices for $\phi_c$. Then, under standard regularity conditions,

$$I_{X,Y}(\phi_c) \; = \; I_X(\phi_c) + I_Y(\phi_c). \tag{6}$$

In particular, for any $v \in \mathbb{R}^{d_c} \setminus \{0\}$,

$$v^\top I_{X,Y}(\phi_c)v \; = \; v^\top I_X(\phi_c)v \; + \; v^\top I_Y(\phi_c)v. \tag{7}$$

*Proof.* For fixed $\phi_x, \phi_y$, the log-likelihood contributions are

$$\ell_X(\phi_c) := \sum_{i=1}^{N_x} \log p_X\big(X_i \mid \phi_c, \phi_x\big),$$

$$\ell_Y(\phi_c) := \sum_{j=1}^{N_y} \log p_Y\big(Y_j \mid \phi_c, \phi_y\big),$$

and $\ell(\phi_c) := \ell_X(\phi_c) + \ell_Y(\phi_c)$. Let

$$g_X(\phi_c) := \nabla_{\phi_c}\ell_X(\phi_c), \qquad g_Y(\phi_c) := \nabla_{\phi_c}\ell_Y(\phi_c), \qquad g(\phi_c) := \nabla_{\phi_c}\ell(\phi_c) = g_X(\phi_c) + g_Y(\phi_c)$$

be the corresponding score functions. The Fisher information for $\phi_c$ in the joint experiment is

$$I_{X,Y}(\phi_c) = \mathbb{E}\big[g(\phi_c)g(\phi_c)^\top\big],$$

where the expectation is taken under the joint law of $\{X_i\}_{i=1}^{N_x}$ and $\{Y_j\}_{j=1}^{N_y}$. Since $g_X$ depends only on $\{X_i\}$ and $g_Y$ only on $\{Y_j\}$, conditional independence of $X$ and $Y$ implies that $g_X$ and $g_Y$ are independent random vectors. Moreover, under standard regularity assumptions, each score has zero mean, $\mathbb{E}[g_X(\phi_c)] = \mathbb{E}[g_Y(\phi_c)] = 0$. Hence

$$\begin{aligned}
I_{X,Y}(\phi_c) &= \mathbb{E}\big[(g_X + g_Y)(g_X + g_Y)^\top\big] \\
&= \mathbb{E}[g_X g_X^\top] + \mathbb{E}[g_Y g_Y^\top] + \mathbb{E}[g_X g_Y^\top] + \mathbb{E}[g_Y g_X^\top] \\
&= \mathbb{E}[g_X g_X^\top] + \mathbb{E}[g_Y g_Y^\top] \\
&= I_X(\phi_c) + I_Y(\phi_c),
\end{aligned}$$

which is the desired identity. The directional statement follows by left- and right-multiplying with $v$. $\qquad\square$

*Remark* 1 (Connection to Section 3.1 and UML). The factorization

$$p(x, y \mid \phi_c, \phi_x, \phi_y) = p_X(x \mid \phi_c, \phi_x)\, p_Y(y \mid \phi_c, \phi_y)$$

is the standard conditional-independence assumption that, given the shared and modality-specific parameters, the two observation channels are driven by independent noise. This is exactly the structure of the linear-Gaussian model in Section 3.1 as well. There, $\theta_c$ plays the role of $\phi_c$, least-squares estimators coincide with maximum-likelihood estimators, and their covariance is asymptotically given by the inverse Fisher information, recovering Theorems 1 to 3. UML mirrors this shared-parameter structure by employing a common trunk (parameters $\phi_c$) updated from a loss of the form $\mathcal{L} = \mathcal{L}_X + \mathcal{L}_Y$, so that unpaired modalities contribute additively to the curvature experienced by the shared block.

## D   UML: ALGORITHM PSEUDOCODE

In this section we present the full pseudocode for UML for both the self-supervised and supervised settings as shown in Algorithm 1 and Algorithm 2 respectively.

---

**Algorithm 1** Pytorch Pseudocode for UML in the self-supervised setting

---

```
# f_img, f_text: image encoder, text encoder
# g_img, g_text: image decoder, text decoder
# h: shared backbone

while not converged: # training loop
    x_img = fetch_next(image_loader) # image minibatch
    x_text = fetch_next(text_loader) # text minibatch (random/unaligned)

    x_img = patchify_and_embed(x_img) # input image patch embeddings
    x_text = tokenize_and_embed(x_text) # input text token embeddings

    z_img = f_img(x_img) # image patch embeddings
    z_text = f_text(x_text) # text token embeddings

    y_img = g_img(h(z_img)) # predict image patch embeddings
    y_text = g_text(h(z_text)) # predict text patch embeddings

    # Next Token/Patch Embedding Prediction Loss
    loss_img = MSE(y_img[:,:-1,:], x_img[:,1:,:])
    loss_text = MSE(y_text[:,:-1,:], x_text[:,1:,:])
    loss = loss_img + lambda * loss_text # total loss

    loss.backward() # back-propagate
    update(h, f_img, f_text, g_img, g_text) # SGD update

# Define Mean Squared Error loss
def MSE(pred, target):
    return ((pred - target) ** 2).mean()
```

---

**Algorithm 2** Pytorch Pseudocode for UML in the supervised setting

---

```
# f_img: image encoder (frozen or trainable)
# f_text: text encoder (frozen)
# is_trainable: True if f_img is trainable else False
# h: classification head

while not converged: # training loop
    x_img = fetch_next(image_loader) # image minibatch
    x_text = fetch_next(text_loader) # text minibatch (random/unaligned)

    z_img = f_img(x_img) # image embeddings
    z_text = f_text(x_text) # text embeddings

    logits_img = h(z_img) # predict image labels
    logits_text = h(z_text) # predict text labels

    loss_img = CE(logits_img, labels_img) # image classification loss
    loss_text = CE(logits_text, labels_text) # text classification loss
    loss = loss_img + lambda * loss_text # total loss

    loss.backward() # back-propagate
    update(h, f_img) if is_trainable else update(h) # SGD update

# Define Cross-Entropy loss
def CE(logits, labels):
    return -sum(labels * log_softmax(logits, dim=1)) / len(labels)
```

---

# E  ADDITIONAL EXPERIMENTS

## E.1  IMPROVING IMAGE CLASSIFICATION USING UNPAIRED TEXTS (UNALIGNED ENCODERS)

In this section we report image-classification results on ten benchmarks (see Appendix B.3), covering three settings:

1. **Full-dataset fine-tuning**: train both the vision backbone and classification head (Appendix E.1.1).

2. **Full-dataset linear probe**: train only the classification head (Appendix E.1.2).

3. **Few-shot linear probe**: train only the classification head under few-shot conditions (Appendix E.1.3).

In each setting, we compare UML with baselines across all datasets and multiple DINO-initialized vision backbones. Our method has two variants: Ours (UML), where we alternately train with both image and unpaired text data (see Algorithm 2), and Ours (init) where we initialize the classifier with the average text embedding of each class, providing a strong prior to align image and class level information.

### E.1.1  SUPERVISED FINETUNING (ACROSS ARCHITECTURES)

In this section, we fine-tune both the vision backbone and the linear classifier on ten downstream tasks, comparing UML against strong image-only baselines. We evaluate four DINO-initialized backbones:

- ViT-B/16 in Table 6

- ViT-B/8 in  Table 7

- DINOv2 ViT-S/14 in Table 8

- DINOv2 ViT-B/14 in Table 9

Results for DINOv2 ViT-L/14 are omitted due to computational constraints. Across all backbones, UML consistently improves over the image-only baseline by leveraging unpaired text embeddings. For some backbones such as DINOv2 VIT-B/16, our head-initialization variant (*Ours (init)*) outperforms training using unpaired multimodal data from scratch (*Ours*), while in others it does not.

Table 6: **Full finetuning on classification with ViT-B/16 DINO and OpenLLaMA-3B**. We compare our proposed approach with the image-only baseline when fine-tuning on the target dataset. All vision encoders are initialized from DINO weights, and our approach leverages unpaired text data using OpenLLaMA-3B embeddings.

| | Dataset | | | | | | | | | |
| Method | Stanford Cars | SUN397 | FGVC Aircraft | DTD | UCF101 | Food101 | Oxford Pets | Oxford Flowers | Caltech101 | Average |
|---|---|---|---|---|---|---|---|---|---|---|
| Unimodal | 78.41 | 63.99 | 62.12 | 74.17 | 81.43 | 82.38 | 92.00 | 98.24 | 96.31 | 81.01 |
| Ours | **82.56** | 67.04 | 67.38 | **76.42** | 84.06 | **81.79** | **93.20** | **98.98** | **97.04** | **83.16** |
| Ours (init) | 81.95 | **67.12** | **68.29** | 73.84 | **84.31** | 81.12 | 92.60 | 98.73 | 96.84 | 82.76 |

Table 7: **Full finetuning on classification with ViT-B/8 DINO and OpenLLaMA-3B**. We compare our proposed approach with the image-only baseline when fine-tuning on the target dataset. All vision encoders are initialized from DINO weights, and our approach leverages unpaired text data using OpenLLaMA-3B embeddings.

| Method | Stanford Cars | SUN397 | FGVC Aircraft | DTD | UCF101 | Food101 | Oxford Pets | Oxford Flowers | Caltech101 | Average |
|---|---|---|---|---|---|---|---|---|---|---|
| Unimodal | 85.67 | 68.04 | 72.60 | 76.65 | 83.94 | **85.32** | 93.06 | 99.22 | 96.82 | 84.59 |
| Ours | **87.95** | **70.28** | 75.31 | **77.19** | 85.59 | 84.83 | 93.05 | **99.43** | 97.12 | 85.64 |
| Ours (init) | 87.44 | 70.03 | **76.09** | 76.24 | **86.49** | 84.71 | **93.81** | 99.27 | **97.16** | **85.69** |

Table 8: **Full finetuning on classification with ViT-S/14 DINOv2 and OpenLLaMA-3B**. We compare our proposed approach with the image-only baseline when fine-tuning on the target dataset. All vision encoders are initialized from DINOv2 weights, and our approach leverages unpaired text data using OpenLLaMA-3B embeddings.

| Method | Stanford Cars | SUN397 | FGVC Aircraft | DTD | UCF101 | Food101 | Oxford Pets | Oxford Flowers | Caltech101 | Average |
|---|---|---|---|---|---|---|---|---|---|---|
| Unimodal | 79.45 | 66.20 | 66.99 | 72.16 | 83.18 | 80.65 | 90.67 | 99.18 | 95.45 | 81.54 |
| Ours | 84.87 | **66.72** | 71.54 | 74.14 | **84.77** | 81.16 | **91.87** | 99.55 | 97.03 | 83.52 |
| Ours (init) | **86.39** | 66.03 | **73.44** | **74.27** | 84.69 | **81.97** | 91.72 | **99.82** | **97.60** | **83.99** |

Table 9: **Full finetuning on classification with ViT-B/14 DINOv2 and OpenLLaMA-3B**. We compare our proposed approach with the image-only baseline when fine-tuning on the target dataset. All vision encoders are initialized from DINOv2 weights, and our approach leverages unpaired text data using OpenLLaMA-3B embeddings.

| Method | Stanford Cars | SUN397 | FGVC Aircraft | DTD | UCF101 | Food101 | Oxford Pets | Oxford Flowers | Caltech101 | Average |
|---|---|---|---|---|---|---|---|---|---|---|
| Unimodal | 89.62 | **71.45** | 77.29 | 73.88 | 88.00 | 82.94 | 94.55 | **99.88** | 97.69 | 86.14 |
| Ours | **90.93** | 70.97 | 80.02 | 75.83 | 87.52 | **86.25** | **94.74** | **99.88** | 97.57 | **87.08** |
| Ours (init) | 90.73 | 70.92 | **80.23** | **75.87** | **87.6**0 | 83.43 | 94.47 | 99.80 | **97.93** | 86.77 |

### E.1.2 LINEAR PROBING (ACROSS ARCHITECTURES)

In this section, we train only the linear classifier, on top of the frozen vision and language backbone, on ten downstream tasks, comparing UML against strong image-only baselines. We evaluate five DINO-initialized backbones:

- ViT-B/16 in Table 10

- ViT-B/8 in Table 11

- DINOv2 ViT-S/14 in Table 12

- DINOv2 ViT-B/14 in Table 13

- DINOv2 ViT-L/14 in Table 14

Across all backbones, UML consistently improves over the image-only baseline by leveraging unpaired text embeddings. For all backbones, our head-initialization variant (*Ours (init)*) outperforms training using unpaired multimodal data from scratch (*Ours*).

Table 10: **Full linear probing on classification with ViT-B/16 DINO and OpenLLaMA-3B**. We compare our proposed approach with the image-only baseline when training a linear probe on the target dataset. All vision encoders are initialized from DINO weights, and our approach leverages unpaired text data using OpenLLaMA-3B embeddings.

| Method | Stanford Cars | SUN397 | FGVC Aircraft | DTD | UCF101 | Food101 | Oxford Pets | Oxford Flowers | Caltech101 | Average |
|---|---|---|---|---|---|---|---|---|---|---|
| Unimodal | 67.10 | 64.63 | 56.02 | 72.42 | 81.27 | 74.96 | 93.07 | 98.32 | 95.01 | 78.08 |
| Ours | **68.71** | 65.14 | 57.42 | 72.95 | 82.06 | 75.30 | 93.18 | **98.46** | 96.19 | 78.82 |
| Ours (init) | 68.60 | **65.59** | **57.98** | **73.11** | **82.40** | **75.73** | **93.62** | 98.42 | **96.35** | **79.09** |

Table 11: **Full linear probing on classification with ViT-B/8 DINO and OpenLLaMA-3B**. We compare our proposed approach with the image-only baseline when training a linear probe on the target dataset. All vision encoders are initialized from DINO weights, and our approach leverages unpaired text data using OpenLLaMA-3B embeddings.

| Method | Stanford Cars | SUN397 | FGVC Aircraft | DTD | UCF101 | Food101 | Oxford Pets | Oxford Flowers | Caltech101 | Average |
|---|---|---|---|---|---|---|---|---|---|---|
| Unimodal | 72.01 | 67.19 | 62.02 | 76.18 | 82.95 | 78.57 | 91.99 | **98.78** | 96.23 | 80.66 |
| Ours | **72.93** | 68.17 | 63.49 | **77.13** | 83.16 | 79.87 | **92.59** | 98.50 | **96.47** | 81.37 |
| Ours (init) | 72.81 | **68.36** | **64.09** | 76.48 | **83.72** | **80.01** | 92.50 | 98.74 | 96.43 | **81.46** |

Table 12: **Full linear probing on classification with ViT-S/14 DINOv2 and OpenLLaMA-3B**. We compare our proposed approach with the image-only baseline when training a linear probe on the target dataset. All vision encoders are initialized from DINOv2 weights, and our approach leverages unpaired text data using OpenLLaMA-3B embeddings.

| | Dataset | | | | | | | | | |
| Method | Stanford Cars | SUN397 | FGVC Aircraft | DTD | UCF101 | Food101 | Oxford Pets | Oxford Flowers | Caltech101 | Average |
|---|---|---|---|---|---|---|---|---|---|---|
| Unimodal | 77.48 | 70.72 | 66.28 | 78.25 | 82.64 | 84.39 | 94.29 | 99.62 | 97.00 | 83.40 |
| Ours | 78.45 | 71.53 | 67.33 | 78.70 | 83.51 | 84.67 | 94.70 | 99.82 | 97.11 | 83.98 |
| Ours (init) | **78.58** | **72.24** | **67.50** | **79.51** | **83.57** | **84.74** | **94.78** | **99.89** | **97.15** | **84.22** |

Table 13: **Full linear probing on classification with ViT-B/14 DINOv2 and OpenLLaMA-3B**. We compare our proposed approach with the image-only baseline when training a linear probe on the target dataset. All vision encoders are initialized from DINOv2 weights, and our approach leverages unpaired text data using OpenLLaMA-3B embeddings.

| | Dataset | | | | | | | | | |
| Method | Stanford Cars | SUN397 | FGVC Aircraft | DTD | UCF101 | Food101 | Oxford Pets | Oxford Flowers | Caltech101 | Average |
|---|---|---|---|---|---|---|---|---|---|---|
| Unimodal | 85.46 | 75.42 | 72.34 | 79.73 | **87.26** | 88.70 | 95.56 | 99.76 | 97.81 | 86.89 |
| Ours | 85.40 | 75.22 | **75.22** | 80.73 | 87.21 | **89.02** | 95.83 | **99.88** | 97.85 | 87.37 |
| Ours (init) | **85.74** | **75.70** | 74.17 | **81.32** | **87.26** | 88.78 | **95.78** | **99.88** | **97.93** | **87.40** |

Table 14: **Full linear probing on classification with ViT-L/14 DINOv2 and OpenLLaMA-3B**. We compare our proposed approach with the image-only baseline when training a linear probe on the target dataset. All vision encoders are initialized from DINOv2 weights, and our approach leverages unpaired text data using OpenLLaMA-3B embeddings.

| | Dataset | | | | | | | | | |
| Method | Stanford Cars | SUN397 | FGVC Aircraft | DTD | UCF101 | Food101 | Oxford Pets | Oxford Flowers | Caltech101 | Average |
|---|---|---|---|---|---|---|---|---|---|---|
| Unimodal | 88.16 | 77.26 | 74.32 | 81.56 | 89.82 | 90.95 | 96.27 | 99.84 | 97.97 | 88.46 |
| Ours | **88.45** | 77.20 | 76.93 | 82.39 | **90.19** | 91.09 | **96.51** | **99.92** | **98.01** | 88.97 |
| Ours (init) | 87.99 | **77.75** | **77.20** | **82.51** | 90.17 | **91.29** | 96.32 | **99.92** | 97.93 | **89.01** |

### E.1.3 FEW-SHOT LINEAR PROBING (ACROSS ARCHITECTURES)

In this section, we train only the linear classifier, on top of the frozen vision and language backbone, for few-shot classification on ten downstream tasks, comparing UML against strong image-only baselines. We evaluate five DINO-initialized backbones: ViT-B/16 in Table 16, ViT-B/8 in Table 15, DINOv2 ViT-S/14 in Table 17, DINOv2 ViT-B/14 in Table 19, DINOv2 ViT-L/14 in Table 19. Across all backbones, UML consistently improves over the image-only baseline by leveraging unpaired text embeddings. For all backbones, our head-initialization variant (*Ours (init)*) outperforms training using unpaired multimodal data from scratch (*Ours*).

Table 15: **Linear evaluation of frozen features on 11 fine-grained benchmarks for few-shot learning.** We compare our proposed approach with the image-only baseline by training a linear classifier on top of frozen VIT-B/8 DINO features. Our method leverages unpaired text data using OpenLLaMA-3B

| Train Shot | Method | Stanford Cars | Sun397 | Fgvc Aircraft | Dtd | Ucf101 | Food101 | Imagenet | Oxford Pets | Oxford Flowers | Caltech101 | Average |
|---|---|---|---|---|---|---|---|---|---|---|---|---|
| 1 | Unimodal | 7.40 | 26.37 | 12.16 | 28.62 | 39.75 | 19.23 | 42.81 | 54.97 | 58.22 | 74.13 | 36.37 |
| | Ours | 7.71 | 28.01 | 13.56 | 33.22 | 42.08 | 21.13 | 43.27 | 55.85 | 58.61 | 77.51 | 38.10 |
| | Ours (init) | 9.24 | 34.23 | 14.49 | 36.27 | 47.55 | 24.81 | 46.75 | 60.09 | 61.59 | 80.23 | 41.52 |
| 2 | Unimodal | 14.43 | 37.96 | 20.28 | 39.80 | 53.03 | 30.62 | 54.75 | 68.12 | 77.59 | 81.91 | 47.85 |
| | Ours | 15.71 | 40.74 | 21.04 | 43.74 | 55.86 | 33.52 | 54.49 | 69.86 | 77.18 | 84.52 | 49.67 |
| | Ours (init) | 16.94 | 45.16 | 22.17 | 45.43 | 59.02 | 35.89 | 56.78 | 71.57 | 77.94 | 86.06 | 51.70 |
| 4 | Unimodal | 25.67 | 49.23 | 29.39 | 52.52 | 64.27 | 43.82 | 61.64 | 75.85 | 87.41 | 90.36 | 58.02 |
| | Ours | 27.30 | 51.23 | 31.43 | 54.31 | 66.72 | 45.58 | 61.51 | 77.51 | 87.96 | 91.36 | 59.49 |
| | Ours (init) | 28.54 | 53.68 | 31.31 | 56.13 | 67.47 | 47.40 | 62.84 | 79.10 | 88.29 | 91.98 | 60.67 |
| 8 | Unimodal | 41.04 | 56.86 | 40.03 | 61.15 | 72.39 | 54.47 | 66.10 | 82.30 | 93.95 | 92.28 | 66.06 |
| | Ours | 43.76 | 58.14 | 42.56 | 63.12 | 73.13 | 56.30 | 66.36 | 84.27 | 94.25 | 92.71 | 67.46 |
| | Ours (init) | 44.16 | 59.80 | 42.30 | 64.46 | 74.30 | 57.07 | 67.18 | 84.85 | 94.00 | 93.24 | 68.14 |
| 16 | Unimodal | 57.72 | 61.74 | 52.63 | 67.69 | 76.18 | 62.63 | 68.87 | 87.31 | 96.41 | 94.27 | 72.54 |
| | Ours | 60.11 | 63.21 | 54.53 | 69.33 | 78.13 | 63.74 | 69.44 | 87.73 | 96.89 | 94.54 | 73.76 |
| | Ours (init) | 60.36 | 64.26 | 54.81 | 70.27 | 78.76 | 64.13 | 70.05 | 88.23 | 96.63 | 94.73 | 74.22 |

Table 16: **Linear evaluation of frozen features on 11 fine-grained benchmarks for few-shot learning.** We compare our proposed approach with the image-only baseline by training a linear classifier on top of frozen VIT-B/16 DINO features. Our method leverages unpaired text data using OpenLLaMA-3B

| Train Shot | Method | Stanford Cars | Sun397 | Fgvc Aircraft | Dtd | Ucf101 | Food101 | Imagenet | Oxford Pets | Oxford Flowers | Caltech101 | Average |
|---|---|---|---|---|---|---|---|---|---|---|---|---|
| 1 | Unimodal | 6.28 | 22.43 | 9.72 | 29.22 | 37.85 | 15.40 | 38.67 | 60.12 | 54.62 | 73.25 | 34.76 |
| | Ours | 7.89 | 26.08 | 10.41 | 32.45 | 40.27 | 18.14 | 39.28 | 60.88 | 58.32 | 75.66 | 36.94 |
| | Ours (init) | 8.96 | 31.34 | 12.12 | 34.22 | 44.32 | 21.46 | 42.68 | 66.39 | 60.37 | 79.74 | 40.16 |
| 2 | Unimodal | 12.64 | 35.64 | 14.98 | 38.93 | 51.14 | 26.05 | 50.34 | 70.84 | 75.61 | 83.16 | 45.93 |
| | Ours | 14.38 | 38.62 | 17.00 | 40.37 | 54.28 | 29.24 | 50.83 | 72.88 | 77.14 | 85.95 | 48.07 |
| | Ours (init) | 15.99 | 42.31 | 17.65 | 42.89 | 56.46 | 32.15 | 52.90 | 74.82 | 77.32 | 87.34 | 49.98 |
| 4 | Unimodal | 22.60 | 45.95 | 24.27 | 50.30 | 63.00 | 38.51 | 57.99 | 80.14 | 85.60 | 89.67 | 55.80 |
| | Ours | 24.83 | 48.62 | 25.76 | 52.64 | 64.39 | 40.74 | 57.96 | 80.92 | 87.20 | 91.17 | 57.42 |
| | Ours (init) | 25.83 | 51.01 | 26.35 | 55.06 | 65.86 | 42.69 | 59.32 | 82.23 | 87.83 | 91.99 | 58.82 |
| 8 | Unimodal | 37.68 | 52.94 | 33.67 | 59.18 | 70.62 | 49.48 | 62.97 | 85.26 | 92.83 | 93.17 | 63.78 |
| | Ours | 39.31 | 55.31 | 35.56 | 60.48 | 71.88 | 50.46 | 63.08 | 86.25 | 93.23 | 93.47 | 64.90 |
| | Ours (init) | 40.50 | 57.03 | 35.64 | 62.27 | 73.18 | 51.50 | 64.09 | 86.93 | 93.59 | 93.71 | 65.84 |
| 16 | Unimodal | 52.48 | 58.27 | 45.34 | 64.81 | 75.72 | 56.24 | 66.36 | 88.57 | 95.90 | 94.27 | 69.80 |
| | Ours | 55.84 | 60.57 | 47.70 | 66.21 | 76.81 | 58.26 | 66.47 | 89.60 | 96.55 | 95.12 | 71.31 |
| | Ours (init) | 55.82 | 61.73 | 48.14 | 67.02 | 77.39 | 58.76 | 67.08 | 90.53 | 96.62 | 94.98 | 71.81 |

Table 17: **Linear evaluation of frozen features on 11 fine-grained benchmarks for few-shot learning.** We compare our proposed approach with the image-only baseline by training a linear classifier on top of frozen VIT-S/14 DINOv2 features. Our method leverages unpaired text data using OpenLLaMA-3B

| Train Shot | Method | Stanford Cars | Sun397 | Fgvc Aircraft | Dtd | Ucf101 | Food101 | Imagenet | Oxford Pets | Oxford Flowers | Caltech101 | Average |
|---|---|---|---|---|---|---|---|---|---|---|---|---|
| 1 | Unimodal | 13.18 | 34.15 | 14.09 | 36.60 | 46.74 | 35.18 | 36.48 | 63.51 | 89.62 | 76.66 | 44.62 |
| | Ours | 14.95 | 37.25 | 14.88 | 38.93 | 49.18 | 37.91 | 38.35 | 68.92 | 91.42 | 84.04 | 47.58 |
| | Ours (init) | 16.49 | 41.79 | 15.63 | 42.04 | 52.33 | 42.27 | 42.69 | 73.59 | 93.64 | 84.52 | 50.50 |
| 2 | Unimodal | 24.68 | 47.88 | 23.09 | 47.75 | 56.81 | 48.54 | 50.41 | 75.32 | 96.02 | 86.90 | 55.73 |
| | Ours | 26.93 | 49.65 | 24.29 | 50.99 | 61.67 | 51.77 | 51.31 | 79.44 | 96.90 | 89.80 | 58.28 |
| | Ours (init) | 28.65 | 53.15 | 24.78 | 53.25 | 63.86 | 54.44 | 54.21 | 81.41 | 97.63 | 90.55 | 60.19 |
| 4 | Unimodal | 38.76 | 57.51 | 32.10 | 59.69 | 67.75 | 60.79 | 58.73 | 83.89 | 98.59 | 93.48 | 65.12 |
| | Ours | 41.69 | 58.87 | 33.38 | 61.58 | 69.60 | 62.69 | 59.69 | 86.27 | 98.84 | 94.56 | 66.71 |
| | Ours (init) | 43.17 | 60.89 | 33.86 | 62.43 | 71.13 | 63.88 | 61.38 | 87.36 | 99.17 | 94.96 | 67.82 |
| 8 | Unimodal | 54.56 | 63.00 | 45.05 | 64.78 | 74.19 | 68.06 | 64.53 | 88.68 | 99.27 | 94.35 | 71.65 |
| | Ours | 56.27 | 64.57 | 45.98 | 66.31 | 75.19 | 69.22 | 65.14 | 89.78 | 99.27 | 95.42 | 72.71 |
| | Ours (init) | 57.91 | 65.82 | 47.40 | 67.81 | 75.99 | 69.71 | 66.40 | 90.29 | 99.54 | 95.84 | 73.67 |
| 16 | Unimodal | 67.96 | 67.35 | 55.89 | 71.36 | 77.92 | 73.24 | 68.14 | 90.73 | 99.63 | 96.43 | 76.22 |
| | Ours | 69.42 | 68.50 | 58.54 | 72.24 | 78.69 | 73.80 | 68.70 | 91.87 | 99.72 | 96.63 | 77.80 |
| | Ours (init) | 70.32 | 69.19 | 58.74 | 73.17 | 79.58 | 74.51 | 69.44 | 92.47 | 99.82 | 96.80 | 78.81 |

Table 18: **Linear evaluation of frozen features on 10 fine-grained benchmarks for few-shot learning with DINOv2 ViT-B/14.** We compare our proposed approach with the image-only baseline by training a linear classifier on top of frozen VIT-B/14 DINOv2 features. Our method leverages unpaired text data using OpenLLaMA-3B

| Train Shot | Method | Stanford Cars | Sun397 | Fgvc Aircraft | Dtd | Ucf101 | Food101 | Imagenet | Oxford Pets | Oxford Flowers | Caltech101 | Average |
|---|---|---|---|---|---|---|---|---|---|---|---|---|
| 1 | Unimodal | 22.42 | 43.03 | 15.79 | 38.85 | 58.57 | 48.71 | 52.26 | 76.47 | 97.12 | 83.64 | 53.69 |
| | Ours | 23.10 | 45.12 | 16.22 | 42.69 | 61.05 | 51.30 | 52.45 | 78.14 | 98.08 | 87.68 | 55.58 |
| | Ours (init) | 25.47 | 48.56 | 16.83 | 45.31 | 63.53 | 54.16 | 55.56 | 81.08 | 97.94 | 88.13 | 57.66 |
| 2 | Unimodal | 35.17 | 55.41 | 25.54 | 51.16 | 69.49 | 62.13 | 62.35 | 84.31 | 99.58 | 89.55 | 63.47 |
| | Ours | 37.38 | 56.98 | 25.88 | 54.65 | 70.61 | 63.89 | 63.21 | 85.50 | 99.70 | 92.02 | 64.98 |
| | Ours (init) | 38.78 | 59.81 | 26.00 | 55.61 | 71.38 | 66.54 | 65.06 | 86.49 | 99.62 | 92.79 | 66.21 |
| 4 | Unimodal | 51.40 | 63.68 | 34.25 | 61.25 | 76.32 | 71.60 | 68.86 | 89.05 | 99.76 | 94.51 | 71.07 |
| | Ours | 54.26 | 64.65 | 35.52 | 62.63 | 76.87 | 72.33 | 69.14 | 90.00 | 99.70 | 95.51 | 72.06 |
| | Ours (init) | 55.01 | 66.55 | 35.14 | 63.97 | 77.57 | 73.25 | 70.30 | 90.31 | 99.57 | 95.65 | 72.73 |
| 8 | Unimodal | 66.01 | 68.88 | 48.17 | 66.67 | 79.92 | 76.26 | 72.48 | 90.97 | 99.80 | 95.54 | 76.47 |
| | Ours | 68.53 | 69.75 | 50.88 | 68.46 | 81.44 | 77.34 | 73.12 | 92.39 | 99.70 | 96.20 | 77.78 |
| | Ours (init) | 67.91 | 70.66 | 51.26 | 69.56 | 81.85 | 77.95 | 73.75 | 92.50 | 99.68 | 96.51 | 78.16 |
| 16 | Unimodal | 77.31 | 72.17 | 62.38 | 73.76 | 83.80 | 80.74 | 75.15 | 93.34 | 99.81 | 97.40 | 81.59 |
| | Ours | 78.92 | 72.80 | 64.51 | 75.16 | 84.62 | 81.00 | 75.46 | 92.92 | 99.59 | 97.38 | 82.24 |
| | Ours (init) | 78.52 | 73.18 | 65.81 | 75.65 | 84.77 | 81.18 | 75.82 | 93.28 | 99.78 | 97.57 | 82.56 |

Table 19: **Linear evaluation of frozen features on 10 fine-grained benchmarks for few-shot learning with DINOv2 ViT-L/14.** We compare our proposed approach with the image-only baseline by training a linear classifier on top of frozen VIT-L/14 DINOv2 features. Our method leverages unpaired text data using OpenLLaMA-3B

| Train Shot | Method | Stanford Cars | Sun397 | Fgvc Aircraft | Dtd | Ucf101 | Food101 | Imagenet | Oxford Pets | Oxford Flowers | Caltech101 | Average |
|---|---|---|---|---|---|---|---|---|---|---|---|---|
| 1 | Unimodal | 24.89 | 48.36 | 17.69 | 38.77 | 66.46 | 59.27 | 57.50 | 79.83 | 98.13 | 82.96 | 57.39 |
| | Ours | 25.88 | 49.63 | 18.08 | 42.93 | 69.18 | 60.12 | 58.37 | 83.51 | 98.42 | 86.23 | 59.24 |
| | Ours (init) | 27.90 | 52.86 | 18.95 | 43.18 | 70.98 | 63.17 | 60.80 | 83.86 | 98.59 | 88.17 | 60.85 |
| 2 | Unimodal | 39.95 | 58.95 | 26.87 | 50.18 | 75.79 | 70.74 | 67.14 | 84.71 | 99.74 | 89.82 | 66.39 |
| | Ours | 41.22 | 60.82 | 27.15 | 53.01 | 76.61 | 72.07 | 67.90 | 86.07 | 99.72 | 91.95 | 67.65 |
| | Ours (init) | 42.93 | 63.36 | 28.14 | 54.96 | 77.72 | 73.87 | 69.20 | 87.13 | 99.81 | 91.71 | 68.88 |
| 4 | Unimodal | 56.49 | 66.37 | 38.59 | 59.08 | 80.84 | 77.39 | 72.41 | 89.90 | 99.73 | 94.44 | 73.52 |
| | Ours | 58.19 | 67.36 | 39.57 | 61.78 | 81.36 | 78.19 | 72.82 | 90.99 | 99.76 | 95.27 | 74.53 |
| | Ours (init) | 58.60 | 68.84 | 39.19 | 62.77 | 81.50 | 78.99 | 73.63 | 90.74 | 99.88 | 96.02 | 75.02 |
| 8 | Unimodal | 70.00 | 70.71 | 51.57 | 66.47 | 83.84 | 81.69 | 76.02 | 93.53 | 99.89 | 95.55 | 78.93 |
| | Ours | 71.63 | 71.59 | 55.13 | 67.91 | 84.47 | 82.12 | 76.43 | 93.62 | 99.88 | 96.36 | 79.91 |
| | Ours (init) | 72.02 | 72.51 | 55.49 | 69.03 | 84.57 | 82.52 | 76.78 | 93.80 | 99.89 | 96.73 | 80.33 |
| 16 | Unimodal | 80.84 | 73.83 | 64.13 | 73.96 | 87.43 | 84.58 | 77.78 | 94.69 | 99.91 | 97.36 | 83.45 |
| | Ours | 81.85 | 74.39 | 69.45 | 74.70 | 87.35 | 84.58 | 78.35 | 94.59 | 99.89 | 97.61 | 84.28 |
| | Ours (init) | 82.76 | 74.80 | 69.42 | 74.88 | 87.65 | 84.96 | 78.58 | 94.42 | 99.81 | 97.62 | 84.49 |

## E.2 IMPROVING IMAGE CLASSIFICATION USING UNPAIRED TEXTS (ALIGNED ENCODERS)

### E.2.1 SUPERVISED FINETUNING

In this section, we fine-tune both the vision backbone and the linear classifier on nine downstream tasks, comparing UML against strong image-only baselines. We evaluate two different backbones: ResNet-50 and VIT-B/16.

As shown in Table 20, across all backbones, UML consistently improves over the image-only baseline by leveraging unpaired text embeddings. Further, our head-initialization variant (*Ours (init)*) outperforms training using unpaired multimodal data from scratch (*Ours*).

Table 20: **Supervised finetuning on 9 fine-grained classification benchmarks with CLIP**. We compare our proposed approach with the image-only baseline when fine-tuning on the target dataset. All vision encoders are initialized from CLIP ResNet50 weights, and our approach leverages unpaired text data using the corresponding CLIP text encoder.

| Method | Stanford Cars | Sun397 | Fgvc Aircraft | Dtd | Ucf101 | Food101 | Oxford Pets | Oxford Flowers | Caltech101 | Average |
|---|---|---|---|---|---|---|---|---|---|---|
| Unimodal | 36.12 | 25.93 | 37.70 | 51.06 | 52.49 | 69.24 | 63.17 | 88.42 | 83.61 | 56.42 |
| Ours | 37.00 | 24.05 | 41.34 | 55.67 | 60.48 | 69.77 | 74.49 | 92.57 | 84.79 | 60.02 |
| Ours (init) | **72.75** | **62.33** | **66.58** | **56.50** | **67.54** | **76.95** | **86.97** | **94.80** | **87.95** | **74.71** |

Table 21: **Full linear probing on classification with CLIP ResNet-50 Image Encoder and Text encoder**. We compare our proposed approach with the image-only baseline when training a linear probe on the target dataset. All vision encoders are initialized from ResNet-50 weights, and our approach leverages unpaired text data using the corresponding CLIP text embeddings.

| | Dataset | | | | | | | | | |
|---|---|---|---|---|---|---|---|---|---|---|
| Method | Stanford Cars | SUN397 | FGVC Aircraft | DTD | UCF101 | Food101 | Oxford Pets | Oxford Flowers | Caltech101 | Average |
| Unimodal | 76.36 | 70.97 | 41.88 | 72.81 | 81.23 | 81.60 | 88.39 | **97.89** | 92.78 | 78.21 |
| Ours | 77.23 | 71.18 | 42.66 | 71.81 | 81.81 | 81.51 | 87.84 | 97.65 | 93.01 | 78.30 |
| Ours (init) | **79.14** | **73.83** | **42.81** | **73.76** | **82.13** | **82.44** | **90.90** | 97.69 | **94.19** | **79.65** |

### E.2.2 LINEAR PROBING

In this section, we train only the linear classifier, on top of the frozen vision and language backbone from CLIP, on ten downstream tasks, comparing UML against strong image-only baselines.

As shown in Table 21, UML consistently improves over the image-only baseline by leveraging unpaired text embeddings. Further, our head-initialization variant (*Ours (init)*) outperforms training using unpaired multimodal data from scratch (*Ours*).

### E.2.3 FEW-SHOT LINEAR PROBING (ACROSS ARCHITECTURES)

In this section, we train only the linear classifier, on top of the frozen vision and language backbone from CLIP, for few-shot classification on ten downstream tasks, comparing UML against strong image-only baselines. We evaluate two different backbones: ResNet-50 and VIT-B/16.

As shown in Table 22 and Table 23, across both backbones, UML consistently improves over the image-only baseline by leveraging unpaired text embeddings. Further, our head-initialization variant (*Ours (init)*) outperforms training using unpaired multimodal data from scratch (*Ours*).

### E.3 IMPROVING VISUAL ROBUSTNESS USING UNPAIRED TEXTS

In this section, we evaluate the robustness of models trained with UML to test-time distribution shifts. We train a k-shot linear probe (where $k \in \{1, 2, 4, 8\}$) with DINOv2 on ImageNet and evaluate across four distribution-shifted target datasets: ImageNet-V2, ImageNet-Sketch, ImageNet-A, and ImageNet-R. Our method consistently improves robustness over the unimodal baseline (Figure 10, Figure 11, Figure 12 and Figure 13) across different training shots, indicating that language priors help capture more transferable features.

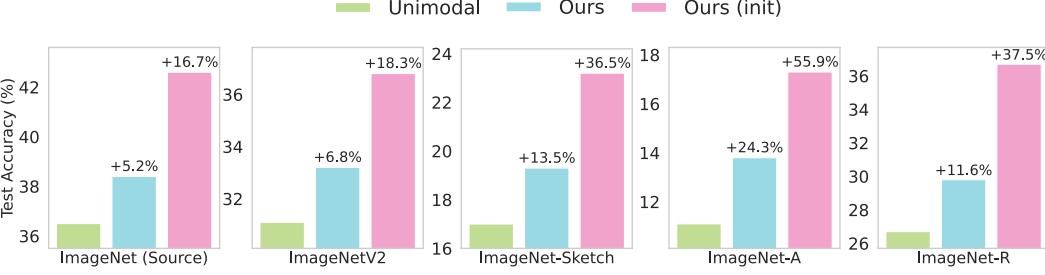

Figure 10: **Robustness under test-time distribution shifts.** Our approach (trained on 1-shot) is much more robust than its unimodal counterpart across four distribution-shuffled target test sets.

Table 22: **Linear evaluation of frozen features on 10 fine-grained benchmarks for few-shot learning.** We compare our proposed approach with the image-only baseline by training a linear classifier on top of frozen CLIP ResNet50 features. Our method leverages unpaired text data using the corresponding CLIP text encoder

| | | Dataset | | | | | | | | | | |
|---|---|---|---|---|---|---|---|---|---|---|---|---|
| Train Shot | Method | Stanford Cars | Sun397 | Fgvc Aircraft | Dtd | Ucf101 | Food101 | Oxford Pets | Imagenet | Oxford Flowers | Caltech101 | Average |
| 1 | Unimodal | 23.24 | 29.14 | 12.38 | 30.24 | 37.55 | 27.26 | 34.61 | 21.36 | 59.07 | 66.52 | 34.14 |
| | Ours | 36.32 | 45.40 | 16.84 | 40.92 | 53.19 | 49.76 | 53.03 | 36.48 | 68.56 | 76.80 | 47.73 |
| | Ours (init) | 57.88 | 64.59 | 22.23 | 50.85 | 65.99 | 76.73 | 86.59 | 60.92 | 81.08 | 83.79 | 65.06 |
| 2 | Unimodal | 38.37 | 43.83 | 18.63 | 40.33 | 53.25 | 44.60 | 47.75 | 32.62 | 75.03 | 78.90 | 47.33 |
| | Ours | 46.64 | 53.53 | 20.81 | 48.35 | 62.01 | 56.67 | 60.64 | 42.21 | 77.97 | 84.58 | 55.34 |
| | Ours (init) | 61.86 | 65.90 | 24.19 | 55.30 | 70.39 | 77.07 | 87.40 | 61.40 | 86.20 | 85.94 | 67.57 |
| 4 | Unimodal | 51.34 | 54.38 | 23.08 | 52.07 | 64.06 | 57.29 | 61.32 | 41.72 | 86.16 | 85.41 | 57.68 |
| | Ours | 55.21 | 59.48 | 24.77 | 56.78 | 67.65 | 62.68 | 67.31 | 47.04 | 86.46 | 87.23 | 61.46 |
| | Ours (init) | 65.80 | 68.11 | 27.49 | 60.13 | 73.62 | 77.79 | 86.54 | 62.37 | 91.60 | 87.57 | 70.10 |
| 8 | Unimodal | 61.74 | 61.47 | 30.22 | 60.15 | 70.16 | 64.63 | 68.94 | 49.48 | 92.20 | 89.14 | 64.81 |
| | Ours | 62.75 | 63.70 | 30.69 | 61.84 | 70.74 | 67.73 | 73.62 | 52.14 | 92.31 | 89.89 | 66.54 |
| | Ours (init) | 69.78 | 69.61 | 31.62 | 64.13 | 77.24 | 78.58 | 89.07 | 63.34 | 94.21 | 91.58 | 72.92 |
| 16 | Unimodal | 70.94 | 65.53 | 35.91 | 64.30 | 75.13 | 70.67 | 78.49 | 55.07 | 95.21 | 91.26 | 70.25 |
| | Ours | 71.58 | 67.08 | 36.23 | 65.62 | 76.09 | 71.63 | 79.52 | 56.92 | 95.44 | 91.94 | 71.20 |
| | Ours (init) | 74.56 | 71.33 | 37.13 | 68.09 | 78.66 | 79.06 | 89.71 | 64.31 | 96.17 | 93.31 | 75.23 |

Table 23: **Linear evaluation of frozen features on 10 fine-grained benchmarks for few-shot learning.** We compare our proposed approach with the image-only baseline by training a linear classifier on top of frozen CLIP VIT-B/16 features. Our method leverages unpaired text data using the corresponding CLIP text encoder

| | | Dataset | | | | | | | | | | |
|---|---|---|---|---|---|---|---|---|---|---|---|---|
| Train Shot | Method | Stanford Cars | Sun397 | Fgvc Aircraft | Dtd | Ucf101 | Food101 | Oxford Pets | Imagenet | Oxford Flowers | Caltech101 | Average |
| 1 | Unimodal | 31.53 | 33.51 | 17.76 | 31.72 | 43.64 | 39.40 | 37.43 | 27.65 | 67.95 | 71.68 | 40.23 |
| | Ours | 48.28 | 53.44 | 22.06 | 47.04 | 63.40 | 63.92 | 60.95 | 47.35 | 77.82 | 83.14 | 56.74 |
| | Ours (init) | 67.76 | 70.13 | 32.26 | 55.16 | 75.02 | 84.25 | 90.91 | 69.50 | 87.58 | 88.87 | 72.14 |
| 2 | Unimodal | 48.45 | 48.70 | 23.38 | 42.04 | 60.08 | 58.30 | 53.56 | 41.68 | 82.01 | 83.20 | 54.14 |
| | Ours | 57.89 | 59.95 | 27.19 | 52.27 | 69.60 | 71.18 | 66.78 | 54.24 | 87.43 | 90.20 | 63.67 |
| | Ours (init) | 70.75 | 71.52 | 33.99 | 60.17 | 78.37 | 85.39 | 90.67 | 70.19 | 92.18 | 90.09 | 74.33 |
| 4 | Unimodal | 61.64 | 60.66 | 31.01 | 54.37 | 70.49 | 71.91 | 69.35 | 52.15 | 90.99 | 91.08 | 65.36 |
| | Ours | 66.24 | 65.56 | 32.98 | 59.95 | 74.16 | 76.19 | 75.92 | 58.50 | 91.32 | 93.23 | 69.40 |
| | Ours (init) | 74.58 | 73.54 | 37.38 | 64.30 | 81.10 | 86.05 | 91.64 | 70.89 | 94.80 | 93.70 | 76.80 |
| 8 | Unimodal | 71.76 | 66.67 | 38.47 | 61.96 | 77.11 | 78.16 | 78.25 | 59.90 | 95.20 | 92.98 | 72.05 |
| | Ours | 72.77 | 69.50 | 39.09 | 64.89 | 79.01 | 80.07 | 80.85 | 62.63 | 94.98 | 94.36 | 73.82 |
| | Ours (init) | 78.43 | 75.07 | 41.77 | 68.50 | 83.41 | 86.87 | 92.55 | 71.97 | 96.94 | 95.27 | 79.08 |
| 16 | Unimodal | 78.76 | 71.49 | 44.74 | 68.79 | 80.43 | 82.08 | 85.16 | 63.87 | 96.97 | 94.54 | 76.68 |
| | Ours | 79.40 | 72.19 | 45.06 | 69.41 | 81.97 | 82.12 | 85.92 | 64.93 | 96.49 | 95.28 | 77.28 |
| | Ours (init) | 82.38 | 76.51 | 47.14 | 72.13 | 84.66 | 86.60 | 92.68 | 72.79 | 97.70 | 96.08 | 80.87 |

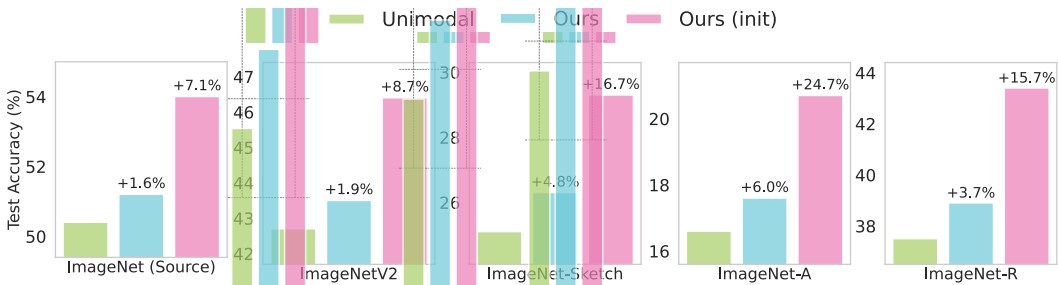

Figure 11: **Robustness under test-time distribution shifts.** Our approach (trained on 2-shots) is much more robust than its unimodal counterpart across four distribution-shuffled target test sets.

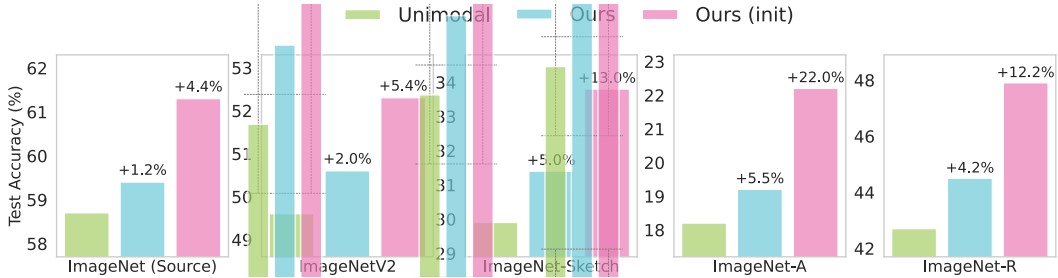

Figure 12: **Robustness under test-time distribution shifts.** Our approach (trained on 4-shots) is much more robust than its unimodal counterpart across four distribution-shuffled target test sets.

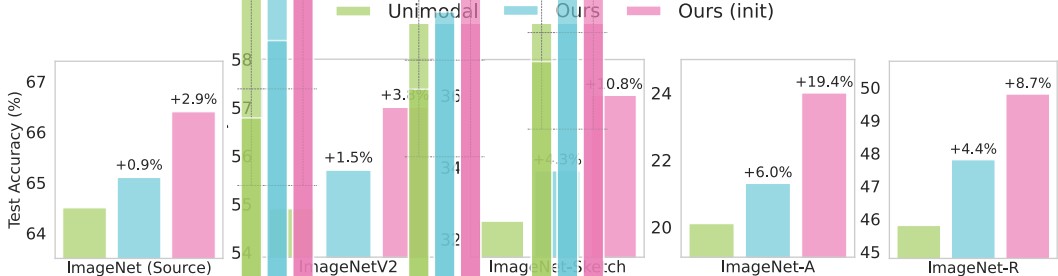

Figure 13: **Robustness under test-time distribution shifts.** Our approach (trained on 8-shots) is much more robust than its unimodal counterpart across four distribution-shuffled target test sets.

### E.4 MARGINAL RATE-OF-SUBSTITUTION BETWEEN MODALITIES

*How many words is an image worth?* In this section, we extend our results to evaluate image-text conversion ratios using test accuracy isolines on the remaining eight datasets. We measure these global equivalence ratios by fitting a plane to the accuracy values given the number of image and text shots. Figures 14 to 21 demonstrate the conversion ratios for DINOv2 VIT-S/14 as the vision backbone and OpenLLaMa-3B as the text backbone (unaligned encoders). Analogously, Figures 22 to 29 show the same ratios for CLIP ResNet-50 as the vision and text encoders (aligned encoders). As expected, with the fully aligned CLIP backbone, each image equates to far fewer text prompts than under the unaligned DINO setting, showing the higher efficiency of aligned embeddings.

### E.4.1 UNALIGNED ENCODERS

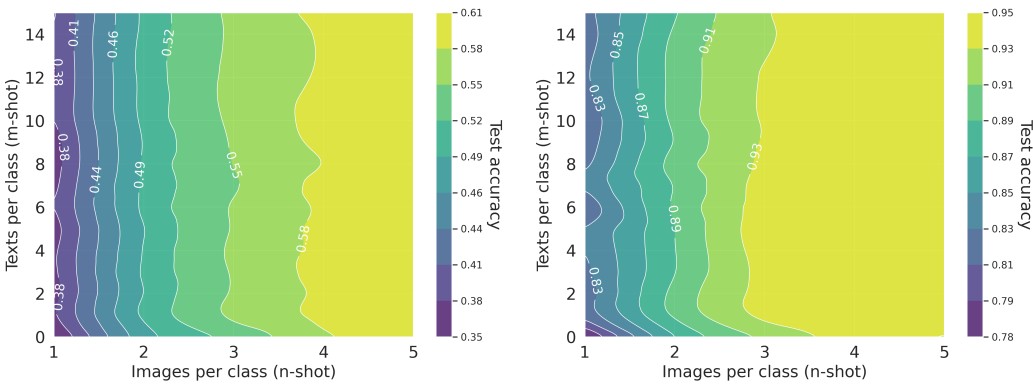

Figure 14: **SUN397.** 1 img ≈ 1568 words    Figure 15: **Caltech101.** 1 img ≈ 1248 words

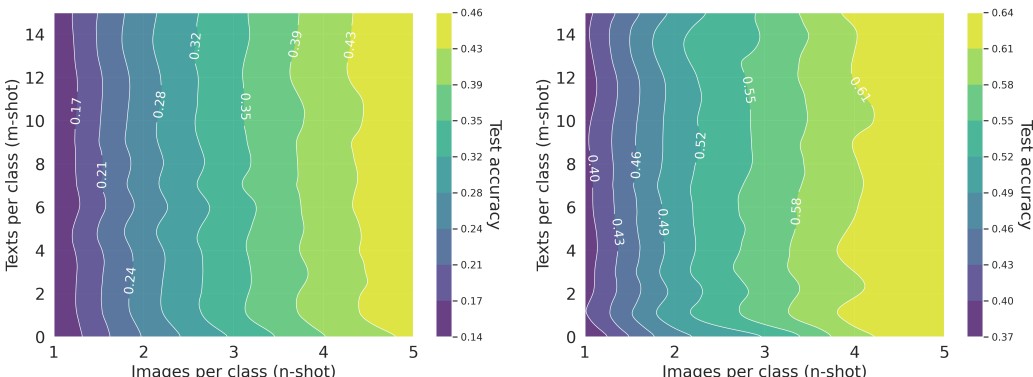

Figure 16: **Stanford Cars.** 1 img ≈ 1799 words    Figure 17: **DTD.** 1 img ≈ 2309 words

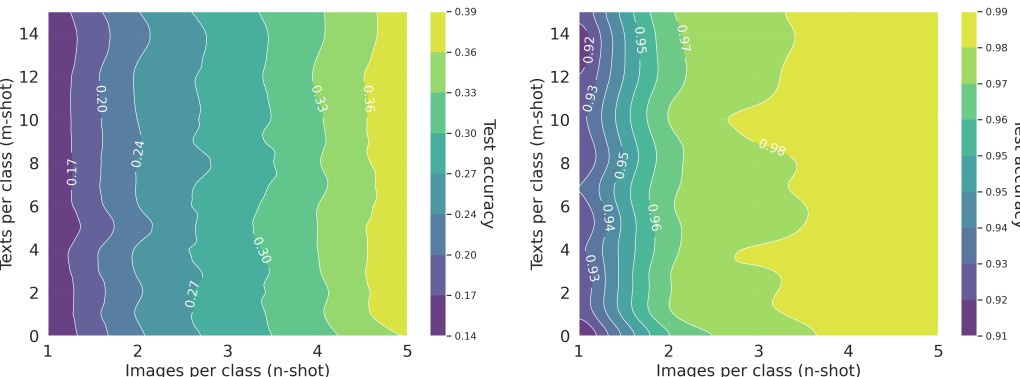

Figure 18: **FGVC Aircraft.** 1 img ≈ 3220 words  Figure 19: **Oxford Flowers.** 1 img ≈ 1895 words

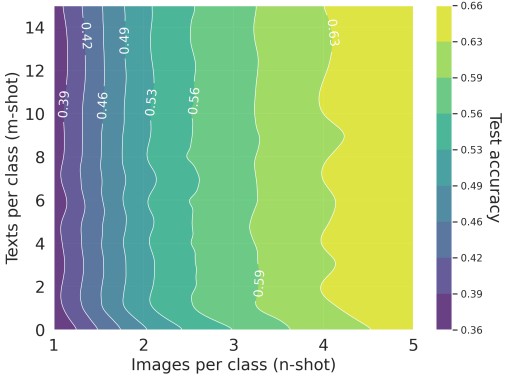

Figure 20: **Food101.** 1 img ≈ 2608 words

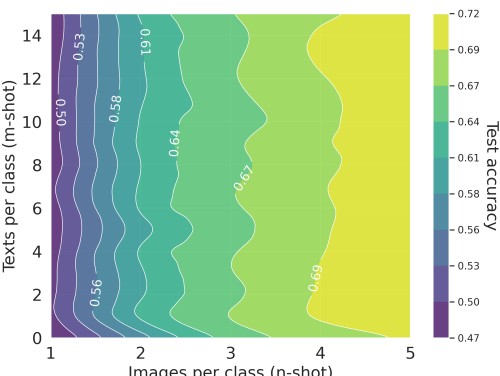

Figure 21: **UCF101.** 1 img ≈ 2617 words

### E.4.2 ALIGNED ENCODERS (CLIP)

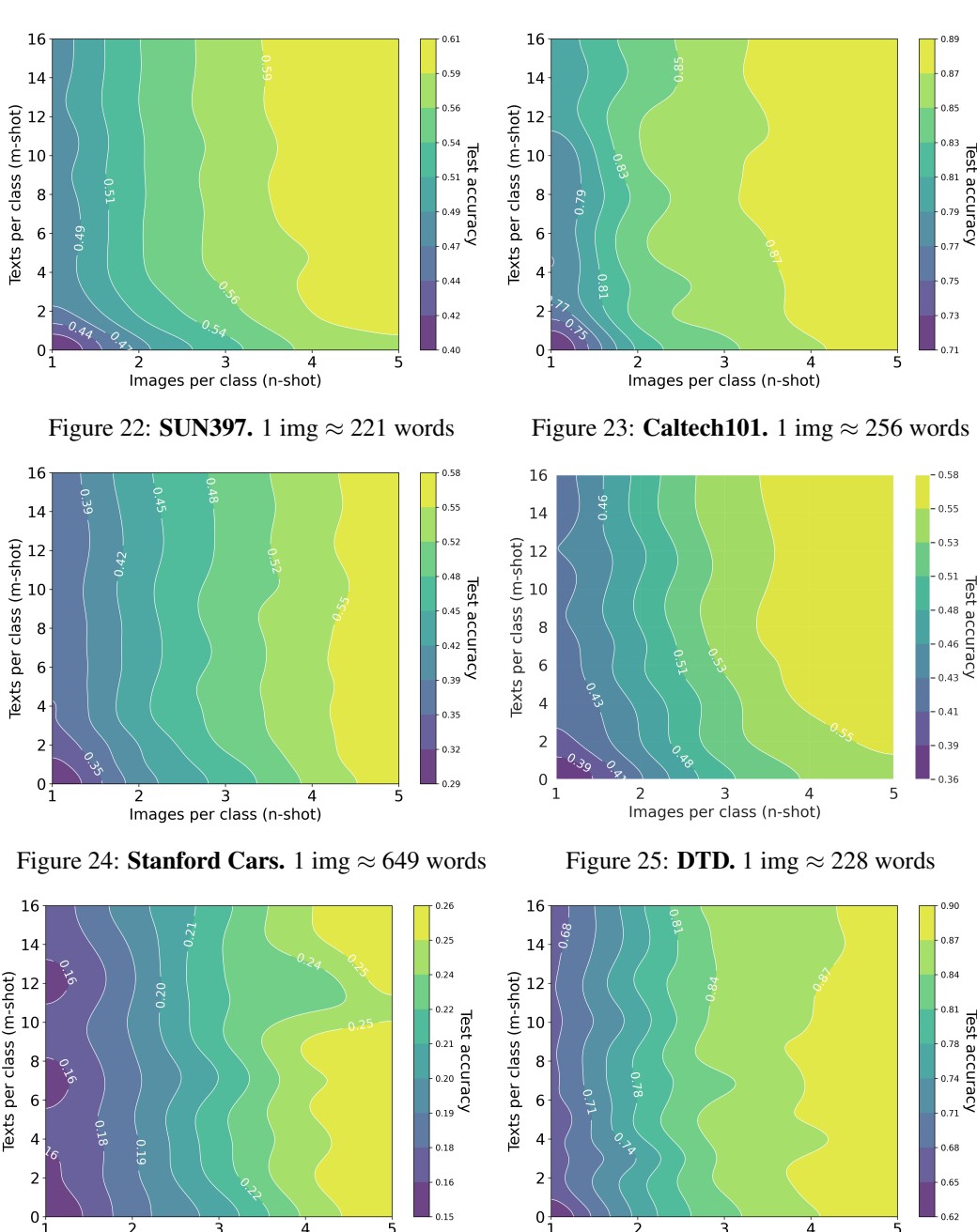

Figure 22: **SUN397.** 1 img ≈ 221 words

Figure 23: **Caltech101.** 1 img ≈ 256 words

Figure 24: **Stanford Cars.** 1 img ≈ 649 words

Figure 25: **DTD.** 1 img ≈ 228 words

Figure 26: **FGVC Aircraft.** 1 img ≈ 691 words    Figure 27: **Oxford Flowers.** 1 img ≈ 851 words

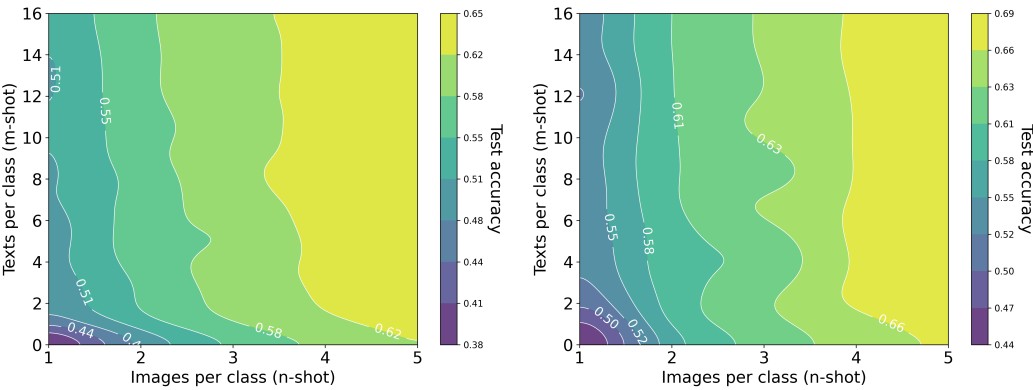

Figure 28: **Food101.** 1 img ≈ 202 words      Figure 29: **UCF101.** 1 img ≈ 393 words

### E.5 IMPACT OF SCALING VISION BACKBONE

In this section, we study how our method's performance scales with the size and architecture of the vision backbone. In addition to ViT-S/14 DINOv2, we extend our analysis to a range of ViT-based architectures, including ViT-B/14 and ViT-L/14 DINOv2 and ViT-B/16 and ViT-B/8 DINO models. To ensure a fair comparison, we follow the same training protocol as in previous experiments. Our method consistently outperforms the unimodal baselines in every setting. In few-shot linear probing across ViT-B/8, ViT-B/16, DINOv2-ViTs and ViT-L/14 backbones (Tables 15 to 19), we see clear gains. The same holds for full-dataset end-to-end fine-tuning of both encoder and head (Tables 6 to 9), and even when only the linear classifier is trained on the full splits (Tables 10 to 14).

### E.6 IMPACT OF VARYING TEXT ENCODERS

In this section, we study how our method's performance varies with different language models used for generating text embeddings. Through this experiment, we aim to understand how differences in embedding quality and model capacity affect the integration of textual information in our multimodal setup. Specifically, we cover LLMs with diverse architectures and scales, including BERT-Large, RoBERTa-Large and GPT-2 Large. As shown in Figure 30, adding unpaired text embeddings shows a significant boost in 1-shot accuracy and still decent gains at 16 shots on SUN397 dataset. Overall, OpenLLaMA-3B outperforms all other language models.

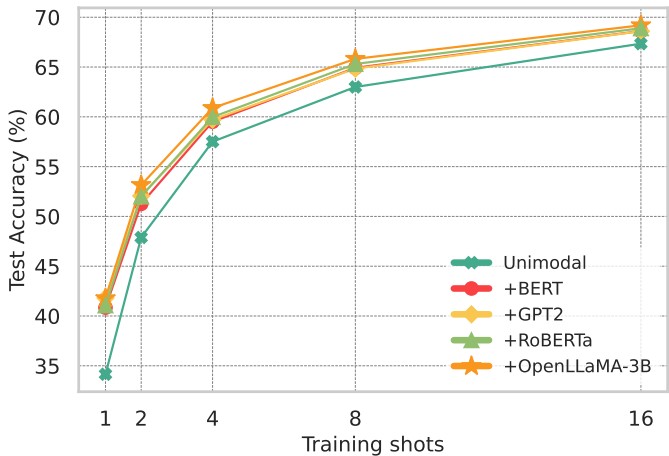

Figure 30: Few-shot classification accuracy on SUN397 using UML with unpaired, frozen embeddings from various pretrained language models.

### E.7 LEARNING WITH COARSE-GRAINED VS. FINE-GRAINED TEXTUAL CUES

Understanding the type of information extracted from textual cues is crucial to assessing the effectiveness of our multimodal approach. A key question is whether the model merely utilizes class names or goes beyond to capture richer, more descriptive features. To investigate this, we compare the performance of our method using two types of text templates: a vanilla template that consists solely of the class name (e.g., "a photo of a [class]") and descriptive templates generated from GPT-3, as detailed in Section Appendix B. As shown in Figure 31 and Figure 32, both multimodal approaches consistently outperform the unimodal baseline, with descriptions from GPT-3 offering a more substantial performance gain. This shows that leveraging richer, contextually diverse text cues can significantly enhance model performance, even in low-shot learning scenarios.

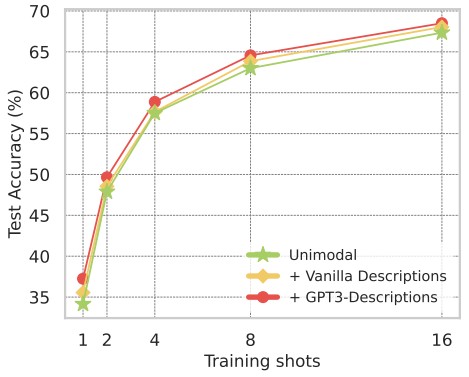
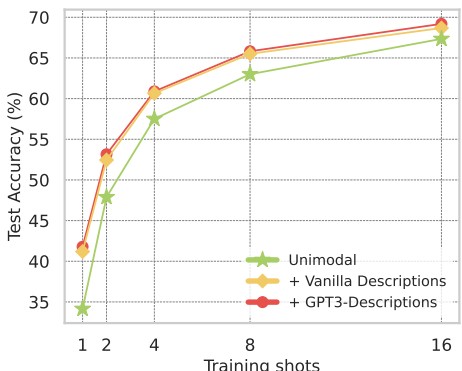

Figure 31: Few-shot SUN397 accuracy with UML using two levels of textual granularity: (a) vanilla class descriptions and (b) GPT-3–generated fine-grained descriptions.

Figure 32: Few-shot SUN397 accuracy with UML (init) using two levels of textual granularity: (a) vanilla class descriptions and (b) GPT-3–generated fine-grained descriptions.

### E.8 IMPACT ON PERFORMANCE WITH INCREASING UNPAIRED TEXT PROMPTS

Here, we investigate how classification accuracy evolves as we augment each image with an increasing number of unpaired text prompts . Figure 33 shows these accuracy curves as we vary the number of unpaired text prompts per image shot across five image-shot budgets. In every regime, our multimodal initialization ("Ours (init)") outperforms training the head from scratch, with most of the gain coming from the first few prompts and gains tapering off thereafter. Note that we do not enforce diversity or novelty in the unpaired text prompts—simply adding more sentences does not guarantee additional information.

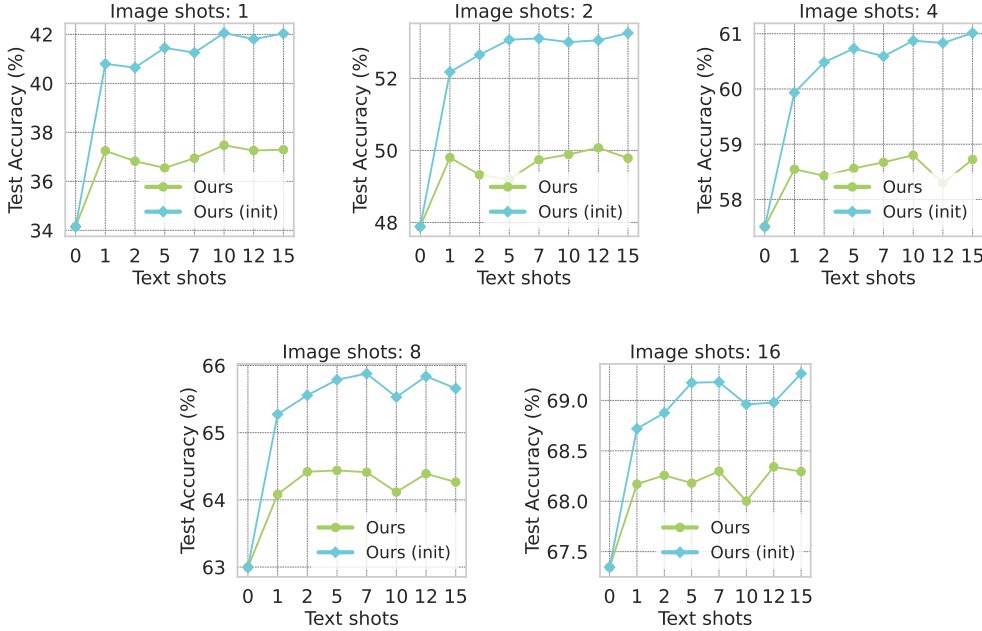

Figure 33: Classification accuracy as a function of the number of text prompts per image shot for the SUN397 Dataset.

### E.9 EFFECT OF UNRELATED AUXILIARY MODALITIES

While our main experiments investigate the benefits of incorporating semantically related auxiliary modalities, an important question is what happens when the auxiliary data is unrelated or even adversarial. In our unpaired setting, this requires reasoning not about individual mismatched pairs, but about the relationship between entire data distributions. To meaningfully study the adversarial case, one would need a principled definition of "negative correlation" between modality distributions, as well as metrics for quantifying it. We leave such a rigorous framework to future work.

Instead, here, we evaluate a simpler but informative scenario: the *independent* case, where the auxiliary modality is semantically unrelated to the target modality. We use our few-shot learning setup to train UML with image data from SUN397 and text data from Stanford-Cars, two semantically unrelated datasets. Table 24 reports results for few-shot image classification on the SUN397 dataset. When the auxiliary text is unrelated to the image domain, performance does not improve over the unimodal baseline. In contrast, semantically related text provides consistent gains across all shot counts.

Table 24: Training UML with unrelated auxiliary text from the Stanford Cars dataset does not yield performance gains on image classification on the SUN397 dataset. However, semantically correlated text (from SUN397) consistently improves accuracy across few-shot settings.

| Method | 1-shot | 2-shot | 4-shot | 8-shot | 16-shot |
|---|---|---|---|---|---|
| Unimodal (Image) | 34.15 | 47.88 | 57.51 | 63.00 | 67.35 |
| UML (Image + Unrelated Text) | 35.27 | 47.12 | 57.50 | 62.45 | 67.25 |
| UML (Image + Related Text) | **41.79** | **53.15** | **60.89** | **65.82** | **69.19** |

### E.10 EXTENSION TO MORE THAN TWO MODALITIES

Our framework naturally extends beyond two modalities. We validate this by extending our image and audio classification experiments on ImageNet-ESC to use all three modalities: image, audio, and

text. Training alternates batches from each modality while applying modality-specific classification losses, consistent with our two-modality setup.

Tables 25 and 26 summarize results on audio and image classification. In both cases, incorporating additional unpaired modalities consistently improves performance over unimodal or pairwise settings. These findings demonstrate that the performance benefits of UML extend robustly to more than two modalities.

From a theoretical perspective, our results also generalize directly. Since modality-specific observations are conditionally independent given the ground-truth latent $\mathcal{Z}^*$, their joint contribution to the Fisher information reduces to the sum of the unimodal blocks. Consequently, the total contribution of all auxiliary modalities (excluding the primary modality $X$) can be obtained by summing their individual Fisher information matrices.

Table 25: Training UML with all three modalities from ImageNet-ESC outperforms unimodal or pairwise training on **audio classification**. Numbers in parentheses denote relative improvements over the unimodal baseline.

| Dataset | Method | 1-shot | 2-shot | 4-shot |
|---------|--------|--------|--------|--------|
| ESC-19 | Audio-Only | 28.78 | 39.85 | 52.22 |
| | Audio + Image | 34.59 | 44.13 | 50.00 |
| | Audio + Text | 35.47 | 52.19 | 52.90 |
| | Audio + Image + Text | **44.46** (+54.4%) | **51.48** (+29.2%) | **56.57** (+8.3%) |
| ESC-27 | Audio-Only | 25.65 | 35.99 | 44.79 |
| | Audio + Image | 37.15 | 42.86 | 51.15 |
| | Audio + Text | 41.97 | 47.02 | 53.85 |
| | Audio + Image + Text | **44.68** (+74.2%) | **48.03** (+33.4%) | **54.16** (+20.9%) |

Table 26: Training UML with all three modalities from ImageNet-ESC outperforms unimodal or pairwise training on **image classification**. Numbers in parentheses denote relative improvements over the unimodal baseline.

| Dataset | Method | 1-shot | 2-shot | 4-shot |
|---------|--------|--------|--------|--------|
| ESC-19 | Image-Only | 60.28 | 74.10 | 78.70 |
| | Image + Audio | 64.63 | 76.17 | 85.02 |
| | Image + Text | 88.84 | 89.71 | 92.07 |
| | Image + Audio + Text | **90.55** (+50.2%) | **91.08** (+22.9%) | **91.72** (+16.5%) |
| ESC-27 | Image-Only | 55.33 | 65.60 | 77.85 |
| | Image + Audio | 59.75 | 70.93 | 78.14 |
| | Image + Text | 86.39 | 88.91 | 90.14 |
| | Image + Audio + Text | **88.22** (+59.5%) | **88.96** (+35.6%) | **91.78** (+17.9%) |

### E.11    EFFECT OF RATIO OF MODALITY BATCHES

In our main experiments, UML was trained with a simple 1:1 alternation of batches across modalities. To study the effect of this schedule, we ablate the ratio of text to image batches, denoted by $r$, on SUN397 using ViT-S/14 DINOv2 (vision) and OpenLLaMA-3B (text). We evaluate both in the (a) linear probe setting and the (b) full finetuning setting.

Across both settings and for both UML and UML (init), we observe that the choice of $r$ has little impact on performance. The gains primarily arise from the presence of auxiliary information rather than the exact frequency of its appearance during training.

Table 27: Few-shot linear probing with different ratios of text-to-image batches on SUN397 using ViT-S/14 DINOv2 and OpenLLaMA-3B. Performance remains stable across ratios $r$, indicating robustness of UML to batch scheduling.

| Method | Shot | $r = 0.25$ | $r = 0.5$ | $r = 1.0$ | $r = 2.0$ | $r = 4.0$ |
|---|---|---|---|---|---|---|
| UML | 2-shot | 49.03 | 49.23 | 49.65 | 49.56 | 49.97 |
| | 4-shot | 59.05 | 59.06 | 58.87 | 58.92 | 58.70 |
| | 8-shot | 64.63 | 65.24 | 64.57 | 64.98 | 65.29 |
| | 16-shot | 68.55 | 68.79 | 68.50 | 68.86 | 68.36 |
| UML (init) | 1-shot | 42.60 | 42.13 | 41.79 | 41.95 | 42.04 |
| | 2-shot | 52.83 | 53.01 | 53.15 | 53.04 | 52.81 |
| | 4-shot | 61.10 | 61.09 | 60.89 | 60.63 | 60.99 |
| | 8-shot | 65.29 | 65.32 | 65.82 | 65.15 | 65.21 |
| | 16-shot | 69.19 | 68.91 | 69.19 | 68.51 | 68.71 |

Table 28: Full finetuning of the vision encoder and linear head with different text-to-image batch ratios $r$ on SUN397. Results show that performance is largely insensitive to $r$.

| Method | $r = 0.25$ | $r = 0.5$ | $r = 1.0$ | $r = 2.0$ | $r = 4.0$ |
|---|---|---|---|---|---|
| UML | 66.44 | 66.80 | 66.72 | 67.60 | 66.44 |
| UML (init) | 66.58 | 65.41 | 66.03 | 65.86 | 64.25 |

### E.12 EFFECT OF FREEZING VS. UNFREEZING THE TEXT ENCODER

In all our main experiments, we freeze the text encoder. This design choice allows us to isolate the role of the auxiliary modality, ensuring that improvements in the primary modality (e.g., vision) arise from cross-modal transfer rather than joint training of both encoders.

In principle, however, one can also unfreeze the text encoder and update it during training. To study this, we ablate freezing versus unfreezing on Stanford Cars and SUN397 using ViT-S/14 DINOv2 (vision) and OpenLLaMA-3B (text). As shown in Table 29, unfreezing the text encoder improves performance on Stanford Cars, but slightly reduces performance on SUN397—likely due to the larger number of trainable parameters and increased optimization complexity.

Table 29: Effect of freezing versus unfreezing the text encoder when training with UML. Freezing stabilizes training and often yields slightly stronger gains.

| Method | Stanford Cars | SUN397 |
|---|---|---|
| Unimodal (Image Only) | 79.45 | 66.20 |
| UML (Unfrozen Text Encoder) | 84.23 | 65.80 |
| UML (Frozen Text Encoder) | **84.87** | **66.72** |

### E.13 ADDITIONAL EXPERIMENTS FOR AUDIO-VISUAL SETTING

In this section, we extend our unpaired multimodal framework to the tri-modal ImageNet–ESC benchmark, examining how unpaired audio and text signals can enhance image classification under both aligned (Appendix E.13.2) and unaligned encoders(Appendix E.13.1). We then reverse the setting, showing that unpaired visual and textual context likewise improves audio classification (Appendix E.13.3).

#### E.13.1 IMPROVING IMAGE CLASSIFICATION WITH UNPAIRED AUDIO AND TEXT (UNALIGNED ENCODERS)

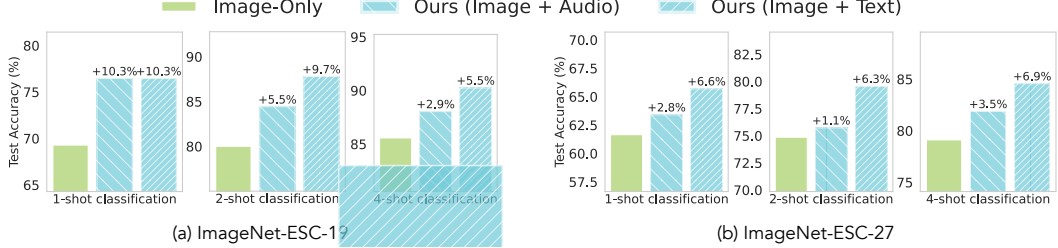

Figure 34: UML improves image classification using unpaired audio and text samples on both ImageNet-ESC-19 and ImageNet-ESC-27 benchmarks when trained on top of DINOv2 VIT-S/14 and OpenLLaMa-3B.

### E.13.2 IMPROVING IMAGE CLASSIFICATION WITH UNPAIRED AUDIO AND TEXT (ALIGNED ENCODERS)

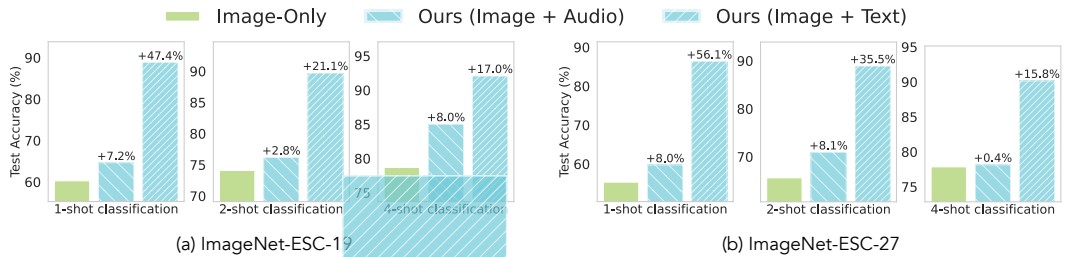

Figure 35: UML improves image classification using unpaired audio and text samples on both ImageNet-ESC-19 and ImageNet-ESC-27 benchmarks when trained on top of CLIP ResNet-50 image and text encoders

### E.13.3 IMPROVING AUDIO CLASSIFICATION WITH UNPAIRED IMAGE AND TEXT (ALIGNED ENCODERS)

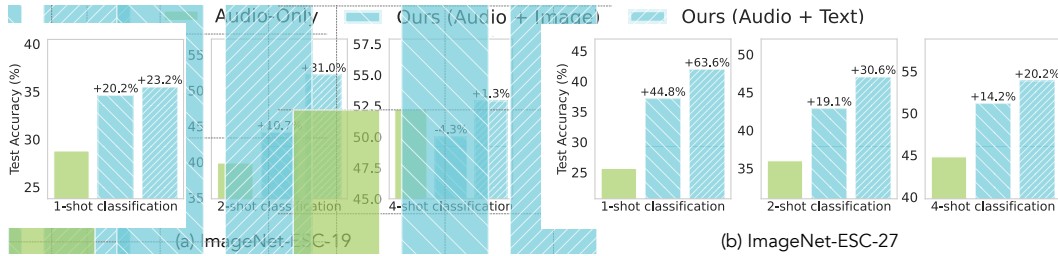

Figure 36: UML improves audio classification using unpaired image and text samples on both ImageNet-ESC-19 and ImageNet-ESC-27 benchmarks when trained on top of CLIP ResNet-50 image and text encoders

### E.14 GAUSSIAN EXPERIMENTS

Here, we shift our attention to a more nuanced and intriguing question: can incorporating unpaired multimodal data actually improve the *reconstruction* quality of a single modality? At first glance, this seems unlikely—why would adding data from a different modality make $X$ reconstruction better than training with $X$? Moreover, we push this question further: can incorporating data from a different modality, *while keeping the total dataset size fixed*, still improve the reconstruction of $X$ compared to using the same number of samples $X$ dataset alone? This setup isolates the importance of multimodal information from mere data scaling, and surprisingly, our experiments show that this improvement is indeed possible.

To investigate this, we design a synthetic experiment inspired by our theoretical framework in Section 3.1. We generate data from two partially overlapping modalities, $X$ and $Y$, derived from a shared latent space $\theta_c$, while also containing unique components ($\theta_x$ and $\theta_y$). The observations follow the same linear structure as in our theory:

$$X_i = A_{c,i}\theta_c + A_{x,i}\theta_x + \epsilon_{X,i}$$
$$Y_j = B_{c,j}\theta_c + B_{y,j}\theta_y + \epsilon_{Y,j}$$

The training data are generated from Gaussian latents with dimensions $\dim(\theta_c) = 10$, $\dim(\theta_x) = 5$, and $\dim(\theta_y) = 5$. For $X$, only the first $10\%$ of the shared components in $\theta_c$ are retained at full strength, while the rest are downscaled by $0.05$; $Y$ observes all shared components at full strength. This asymmetry makes $Y$ informative about structure that is only weakly present in $X$. The validation set is constructed from the same projections but without attenuation, so both modalities fully observe $\theta_c$. Observations are 50-dimensional with Gaussian noise $\epsilon_X, \epsilon_Y \sim \mathcal{N}(0, 0.09I)$. *In the unimodal setting, training on $X$ alone uses $10{,}000$ samples from $X$. When training UML on unpaired $X$ and $Y$, we instead use $5{,}000$ samples from each modality to keep the total sample budget fixed, ensuring a fair comparison.*

Our architecture is a shared autoencoder. Each modality $X \in \mathbb{R}^{50}$, $Y \in \mathbb{R}^{50}$ is projected into a common space of dimension 128 through modality-specific linear layers. A shared encoder (two linear layers with ReLU) maps into a latent space of dimension 10, followed by a shared decoder (two linear layers) that expands back to dimension 128. Finally, modality-specific heads reconstruct the original inputs. The shared pathway enforces cross-modal alignment, while the separate adapters preserve modality fidelity.

As shown in Figure 37, the surprising outcome is that training on both modalities, even when they are unpaired, consistently improves the reconstruction of $X$ compared to training solely on $X$. More strikingly, this improvement holds even when the total number of training samples is fixed, with half the data coming from $X$ and half from $Y$; showing that the model is not just benefiting from increased data quantity but from the diversity and complementary information provided by the second modality.

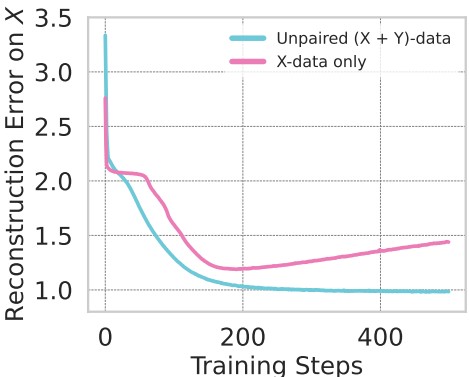

Figure 37: Training on $N/2$ samples from $X$ and $N/2$ unpaired samples from $Y$ improves test reconstruction on $X$, more than training on $N$ samples from $X$.

# F ANALYSIS OF THE LEARNED CLASSIFIER

## F.1 CHANGE IN DECISION BOUNDARIES WITH UNPAIRED DATA FROM ANOTHER MODALITY

Our decision boundary visualizations are constructed by projecting the high-dimensional embedding space of a given classifier to a 2D plane. Axis 1 is computed as the normalized difference between the classifier weights of the two selected classes, representing the primary decision direction. Axis

2 is chosen to be orthogonal to Axis 1, constructed from the difference between the class mean embeddings after removing the component parallel to Axis 1. This orthogonalization ensures that the two axes capture complementary aspects: Axis 1 reflects the primary model decision boundary, while Axis 2 captures the variation orthogonal to that decision. The final 2D projection matrix combines these two vectors as columns, and embedding vectors are then mapped to this plane using a simple dot product. Figure 38 , Figure 39 and Figure 40 show the change in decision boundary when adding unpaired textual information for 2-shot classification on top of frozen CLIP ResNet-50 features for Oxford Pets, DTD and Oxford Flowers datasets.

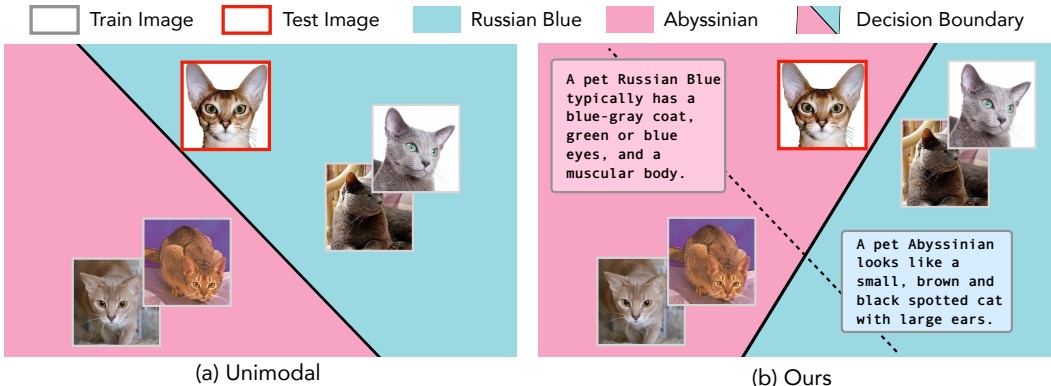

Figure 38: **Impact of unpaired text on decision boundaries (CLIP ResNet50.)** (Left) Visual features alone learn ambiguous class boundaries between Russian Blue and Abyssinian cats. (Right) Adding unpaired text sharpens the boundary, leveraging semantic cues to better distinguish similar categories.

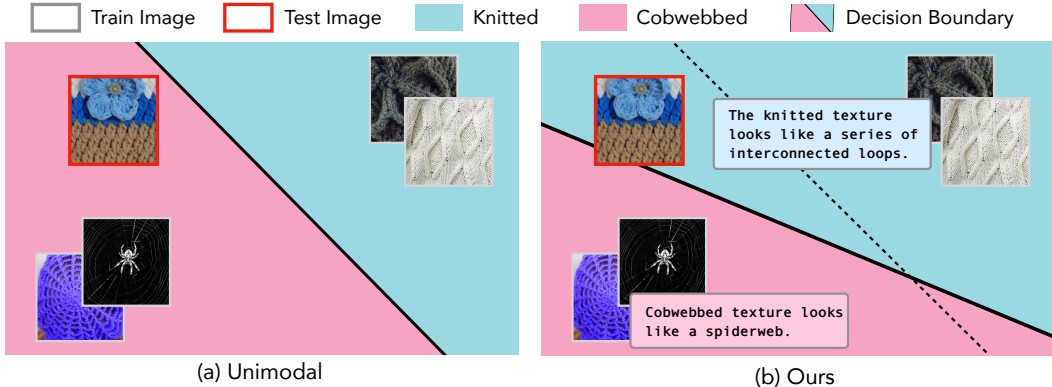

Figure 39: **Impact of unpaired text on decision boundaries (CLIP ResNet50).** (Left) Visual features alone learn ambiguous class boundaries between knitted and cobwebbed. (Right) Adding unpaired text sharpens the boundary, leveraging semantic cues to better distinguish similar categories

### F.2 WHAT DO MODELS LEARN FROM UNPAIRED DATA?

To understand what the model is truly learning and how its weights evolve, we develop and analyze three key metrics: functional margin, silhouette score, and class-prototype vectors. These metrics inform on how well the model distinguishes between classes and how text information influences the structure of feature-space

**Functional margin.** This quantifies how confidently a model separates a given sample from the decision boundary. For a sample $i$ belonging to class $y$, we calculate the margin relative to the next highest competing class. Specifically, we identify the second-highest logit among the incorrect

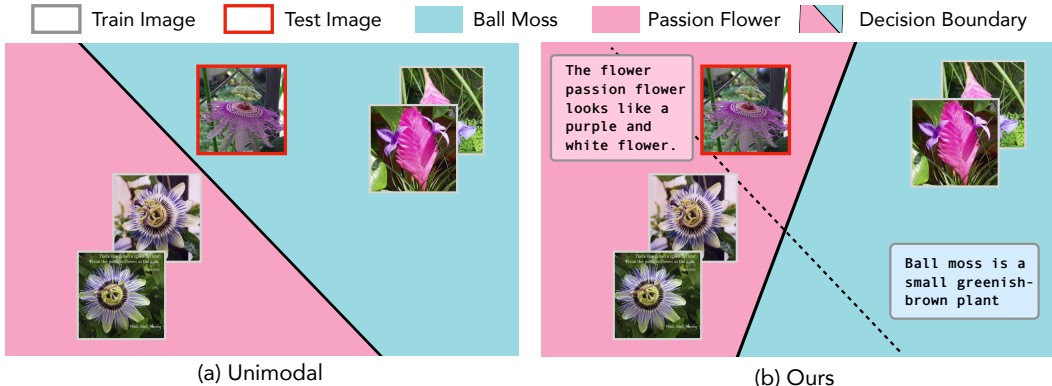

Figure 40: **Impact of unpaired text on decision boundaries (CLIP ResNet50).** (Left) Visual features alone learn ambiguous class boundaries between ball moss and passion flower. (Right) Adding unpaired text sharpens the boundary, leveraging semantic cues to better distinguish similar categories

classes, denoted as class $j^*$, and compute the functional margin as

$$\gamma_i = \frac{w_y^T x_i - w_{j^*}^T x_i}{\|w_y - w_{j^*}\|_2} \tag{8}$$

where $w_y^T x_i$ represents the logit for the true class, while $w_{j^*}^T x_i$ represents the highest logit among the competing classes. Larger margins indicate more confident and robust classification, while smaller margins imply that the sample lies closer to a misclassification boundary. As shown in Figure 41, both *Ours* and *Ours (init)* exhibit substantially larger classification margins than the unimodal baseline, demonstrating that augmenting primary-modality training with unpaired multimodal data improves confidence in predictions over the primary modality.

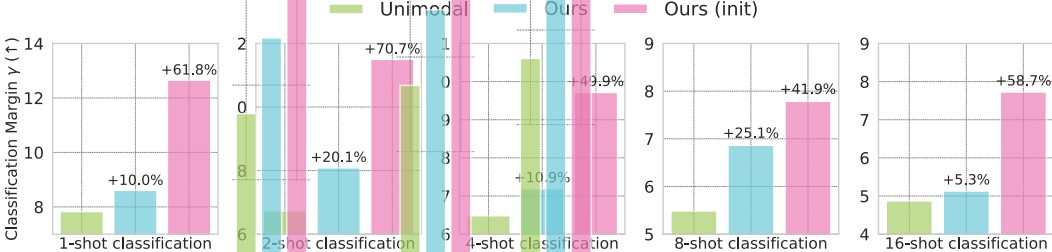

Figure 41: Functional margin of the linear head trained on SUN397 dataset for few-shot classification significantly increases when training with both UML and UML with linear head initialization.

**Silhouette Score and DB-Index.** The Silhouette Score indicates how well-separated the clusters are, while the DB-Index measures intra-class compactness versus inter-class separation. Higher silhouette and lower DB-Index values mean better-defined clusters, indicating that text helps tighten intra-class spread and widen inter-class gaps. As shown in Figure 42 and Figure 43, both *Ours* and *Ours (init)* exhibit reduced intra-class distances and increased inter-class separations, further confirming improved class separability.

**Class-Prototype Vectors.** These vectors are the rows of the final linear layer's weight matrix, representing the class centroids in the shared embedding space. We compute a heatmap of inner products between class prototypes and average text embeddings of the corresponding class to assess how well text features align with class centers. This helps reveal how the model organizes multimodal information. Figure 44 shows a pronounced diagonal structure, indicating that each class's text embedding aligns closely with the learned weights of the model.

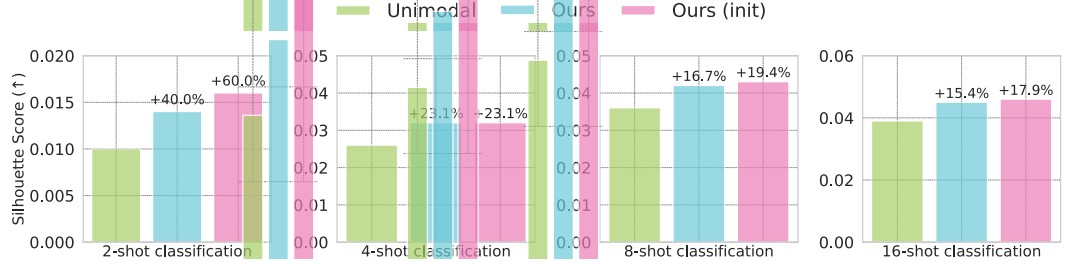

Figure 42: Silhouette Score of the linear head trained on SUN397 dataset for few-shot classification significantly increases when training with both UML and UML with linear head initialization.

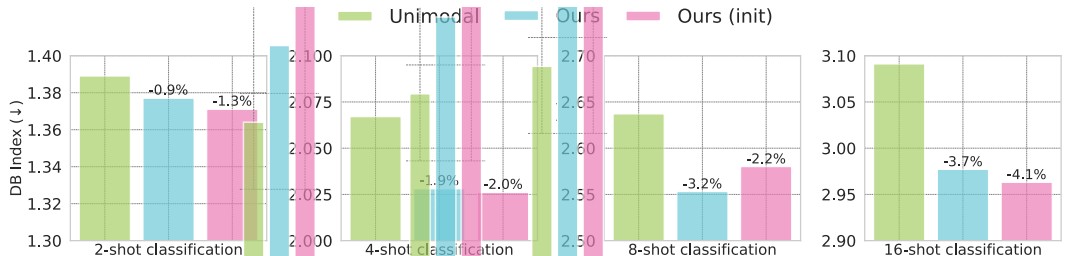

Figure 43: DB-Index of the linear head trained on SUN397 dataset for few-shot classification significantly improves when training with both UML and UML with linear head initialization.

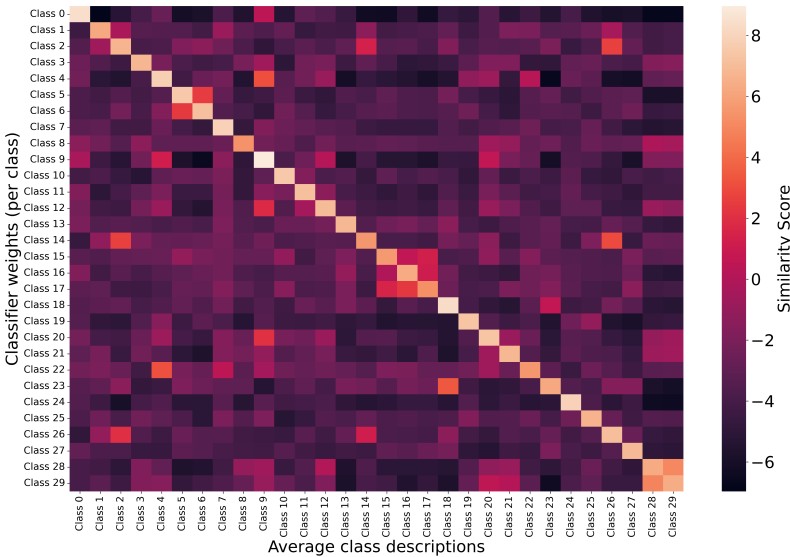

Figure 44: Inner products between each linear-head weight vector and its class's mean text embedding, demonstrating that text features align well with class prototypes.

