# OpenReview forum: "Better Together: Leveraging Unpaired Multimodal Data for Stronger Unimodal Models"
_ICLR.cc/2026/Conference — ICLR 2026 Poster_

### Official Review · Reviewer_mJFa · 2025-10-20

**Soundness:** 3
**Presentation:** 3
**Contribution:** 3
**Rating:** 4
**Confidence:** 4

**Summary:**

The authors propose the Unpaired Multimodal Learner (UML), which leverages auxiliary unpaired multimodal data to enhance representation learning in a target modality. Specifically, the authors present two frameworks, one for self-supervised learning and one for supervised learning, both of which utilize multiple encoders and a shared network. Theoretical and empirical results indicate that adding unpaired data of modality Y can lead to better reconstruction in modality X compared to adding additional data from X itself.

**Strengths:**

1. The topic addressed in the paper is interesting.
2. The paper is clearly written and easy to follow.
3. The theoretical analysis is sound, and the proposed method is simple yet effective.

**Weaknesses:**

1. The primary concern lies in the motivation: unpaired multimodal data is relatively uncommon, whereas missing-modality data is more frequently encountered. If the data is only unpaired, why not use existing models (e.g., CLIP) to align them and create paired data? Therefore, a more practical setting might be using images from one dataset alongside text from another dataset.
2. It would strengthen the paper to include experiments using various encoders and various shared networks to validate the general effectiveness of the method.

**Questions:**

1. The experiments do not include comparisons with related works. Do the authors think it is necessary to compare against other approaches mentioned in the Further Related Works section?

---

> ### Author Response · Authors · 2025-11-18
> **Response to Reviewer mJFa**
>
> We thank the reviewer for the positive assessment of our clarity, soundness, and empirical effectiveness, and for recognizing the interest of the problem. We address feedback below; please let us know there are further comments or concerns.
>
> > Unpaired multimodal data is relatively uncommon
>
> As noted in **Section 2 of our global response**, prior work does not address the setting of joint training on fully unpaired data. In our experiments we learn from unpaired multimodal data; this enables exploiting the vast reservoir of unpaired data on the internet (e.g., 9 trillion photos and videos in Google Photos and  hundreds of billions of web pages on Common Crawl).
>
> Nonetheless, we agree that a “grand challenge” of unpaired multimodal learning is to learn from combining single-modality datasets. Our results provide evidence, through our theory and experiments, that learning in this setting is possible.
>
> We provide the following stepping stones towards the missing-modality setting:
> - **Theory**: Our theoretical analysis only assumes shared latent parameters across modalities, not that every class or instance is observed in every modality, so missing-modality patterns are naturally subsumed by our framework.
> - **Experiments:**
>   - In our labeled datasets, many classes have images but no associated text samples. We then generate conceptually related, unpaired text samples (refer to Section 4.2).
>   - In Section 4.3 we initialize a ViT with weights from pretrained BERT (trained solely on text data) and show transfer performance on ImageNet (solely image data). This provides initial results on the fully missing-modality setting.
>
> We finally note that using CLIP to “align and create pairs” assumes access to a powerful cross-modal model trained on massive paired data, which may not be available for new domains and new modalities like audio–biosignals, etc.
>
> Our work shows that, even without such a cross-modal model, and without the presence of explicit correspondences, we can learn shared latent structures in unpaired multimodal data via shared weights. We agree that missing-modality data is a significant eventual goal and a natural extension of our work; our results provide evidence that this is potentially possible. **We have revised our limitations section (Section 5) to mention the missing-modality setting**.
>
> > Use various encoders and shared networks
>
> We agree that generality across architectures is important. Our results ablate the choice of encoders and shared networks in the following ways:
> - **Different encoders**:
>   - Across our experiments in Sections 4 and E, we use multiple encoders: DINOv2 (vision), CLIP (text/vision), OpenLLaMA-3B (text), BERT (text), and AudioCLIP (audio).
>   - We ablate the impact of different vision encoder sizes in Section E.5 (ViT-S, ViT-B, ViT-L) and the choice of text encoder in Section E.6 (BERT-Large, RoBERTa-Large, GPT2-Large, OpenLLaMA-3B)
> - **Different shared networks**:
>   - In self-supervised UML (Section 4.1) we use a transformer shared head.
>   - In supervised UML (Section 4.2) we use a shared classifier / linear head.
>   - In transfer experiments (Section 4.3) we reuse BERT transformer layers as a shared body for vision.
> - **Theoretical results**: Our theoretical argument is architecture-agnostic: Theorems 1–3 depend only on Fisher information of the linearized data-generating process, not on a specific neural architecture. Any shared-weight learner that accumulates gradients from $X$ and $Y$ is an empirical instantiation of that principle.
>
> > Comparisons with related work
>
> Each of our experiments compare to unimodal baselines. Since no prior work addresses the setting of unpaired multimodal learning, we include various ablations and comparisons to establish strong baselines for this new problem:
> - UML v. unimodal objectives under the same capacity and optimization (Tables 1-2)
> - Modality-aligned v. unaligned encoders (Sections E.1-E.2)
> - Backbone scale (Section E.1, E.5)
> - Backbone v. shared-head training (Sections E.1-E.2)
> - Choice of text encoder (Section E.6)
> - Coarse-grained v. fine-grained text captions (Section E.7)
> - Semantically related v. unrelated auxiliary modalities (Section E.9)
> - Ratio of batches across modalities (Section E.11)
> - Frozen v. unfrozen text encoder (Section E.12)
>
> The methods discussed in Related Works (Appendix A) either assume paired data or pre-aligned cross-modal features, and therefore do not operate in the fully unpaired setting we study. We partially bridge this gap by evaluating UML on top of pre-aligned encoders such as CLIP, and still observe consistent gains over the unimodal training regime. Since no prior work directly tackles unpaired multimodal learning in this form, the most meaningful and fair comparisons are against unimodal and CLIP-based baselines, complemented by ablations.

---

> ### Comment · Reviewer_mJFa · 2025-11-20
>
> Thank the authors for the detailed rebuttal, which largely addresses my concerns, especially regarding why cross-modal alignment models such as CLIP are not used. I am willing to raise my score. However, I still have two remaining questions:
>
> 1. About the shared network $h$.
> Beyond the Transformer/Linear shared head described in Section 4, what other architectures could serve as the shared network to handle different modality combinations, such as audio–biosignal or text–audio–video? In addition, does the training strategy for $h$ in this paper encounter or mitigate the modality conflict or imbalance issues raised by Reviewer 8Sv3?
>
> 2. About comparisons with related work.
> I revisited Tables 1–2 and Section E. The comparisons focus mainly on single-modality baselines and your ablation studies. Although existing related multimodal methods mentioned in Section A may not be designed specifically for unpaired settings, can they be applied to unpaired datasets to report their performance? Such results could further validate the effectiveness of your method.

---

> > ### Author Response · Authors · 2025-11-20
> > **Continued Response to Reviewer mJFa**
> >
> > > Regarding the shared network $h$
> >
> > In our framework, $h$ can be any neural network mapping modality-specific embeddings into a shared representation space. Beyond the Transformer / linear heads we use, this could be an MLP on pooled embeddings, a temporal model (e.g., 1D convolutions or transformers) for audio–biosignal streams, or a sequence model over patch / token / frame embeddings for modalities such as text, audio, or video. The modality-specific encoders are the ones that must adapt to raw data (e.g., spectrograms, video frames), while $h$ itself is architecturally flexible as long as it operates on a common embedding dimension.
> >
> > > On modality conflict / imbalance in our setting
> >
> > In our experiments, we do not observe negative effects from  modality conflicts: across all benchmarks, UML only matches or improves over the unimodal baselines. Thus, we did not introduce explicit conflict-mitigation mechanisms.
> >
> > As we note in our revised Related Works section, existing studies of modality conflict primarily consider joint training on *paired* data, often from *random initialization*.  Modality conflicts can be particularly important in paired settings, as the paired objective may encourage alignment between samples that are, in fact, unaligned. Our setting uses pretrained encoders and fully unpaired data, so the specific failure modes they target do not necessarily apply in our regime, and would require further study.
> >
> > > On comparisons with related multimodal methods
> >
> > Thank you for raising this. Some existing multimodal methods can indeed be adapted to the unpaired setting, even though they were not designed for it. We agree that such comparisons would add useful context. Due to compute and time constraints during the rebuttal period, we will include them in the camera-ready version.
> >
> > *If this response addresses your remaining questions, we would appreciate your consideration of an updated score.*

---

> > > ### Comment · Reviewer_mJFa · 2025-11-21
> > >
> > > Thanks for the response from the authors. Currently, I have no further questions, so I have raised my score to 6.

---

### Official Review · Reviewer_EcBF · 2025-10-27

**Soundness:** 4
**Presentation:** 3
**Contribution:** 3
**Rating:** 6
**Confidence:** 4

**Summary:**

This paper proposes a novel method, **UML** (**U**npaired **M**ultimodal **L**earner), to improve the performance of unimodal models by leveraging abundant, unpaired data from other modalities.
The core idea is that even without direct pairings (like an image and its specific caption), data from an auxiliary modality (e.g., text) can provide complementary information to enhance a model focused on a target modality (e.g., images).
The **UML** method works by having a single model with shared parameters (weight sharing) alternately process inputs from the different modalities. This design allows the model to capture shared underlying concepts and structures from both datasets, even though they are not explicitly linked.
Moreover, the paper theoretically demonstrates that this approach strictly increases the Fisher information under linear assumption of generating data, resulting in more informative and robust representations for the target modality than training on that modality alone.

**Strengths:**

- This paper presents strong theoretical approaches demonstrating how unpaired multimodal datasets, or datasets with missing modalities, can be effectively synergized with existing datasets. It shows that these datasets can be linearly combined, leading to an increase in Fisher Information.
- The paper provides extensive experiments on various benchmark datasets that support its theoretical framework, validating the proposed methods and concepts.
- The paper is well written, particularly in the Introduction and the section explaining the main concept.

**Weaknesses:**

**Major**

The paper’s effort to ground its empirical findings in theoretical analysis is commendable; however, the theoretical assumptions appear too restrictive to be fully explanatory. The authors provide valuable intuition for UML by presenting theorems (Sec. 3.1) derived under a linear data-generating process.
Nonetheless, this represents a significant simplification of the highly non-linear dynamics of the large Transformer-based models (e.g., DINOv2, OpenLLaMA) used in the experiments. While the idea that shared weights act as a practical analogue of *Fisher information linear combination* is intriguing, it remains more of an intuitive analogy than a formal justification.
In my view, this creates a potential gap between theory and practice. The empirical results are undeniably strong and well-presented, yet they seem to be motivated by the theoretical framework rather than explained by it.

**Minor**

- Although the theoretical motivation is clearly written, the derivations of the theorems are somewhat verbose and occasionally redundant, which reduces readability. A more concise and streamlined presentation could enhance the clarity of the theoretical section.
- While the experimental validation is extensive, its presentation in Section 4 and the Appendix seem overly dense. For example, much of the dataset and model information is relegated to the Appendix and only briefly mentioned in Section 4.1, where it hinders readability. Additionally, the color scheme used in Table 1 and Table 2 (blue and pink) is identical, even though the categories differ (Table 1: datasets; Table 2: settings).
As a suggestion (not a requirement), reorganizing the experimental sections might improve clarity. (e.g., focusing on the supervised setting in Section 4.1 and moving the self-supervised setting to Section 4.2, and removing the color scheme from Tables 1 and 2 to avoid confusion)

**Questions:**

- What is the main difference from prior works, and what constitutes the key novelty of this paper? In my view, as the authors mention in Section 3 and Appendix A.1, UML might seem slightly incremental, as it combines concepts from previous studies, such as shared model parameters (e.g., [1]), and the use of unpaired datasets (e.g.,[2]).
- What happens if the text or data (modality $\text{Y}$) are randomly generated? For instance, what if the text description is entirely unrelated to the image (e.g., the image depicts a dog, but the text describes playing sports)? Would such mismatched modalities disrupt the increase in Fisher Information?
- Yet the authors note in the limitations section that most experiments are conducted on classification tasks, I still raise a concern regarding the applicability of the proposed method to other tasks, such as image–text retrieval.


[1]  Chada, et al. "Momo: A shared encoder model for text, image and multi-modal representations." arXiv preprint 2023\
[2] Lee, Jae-Jun, and Sung Whan Yoon. "Can One Modality Model Synergize Training of Other Modality Models?." ICLR 2025

=======================================================

**Note**: I acknowledge that I may have partially misunderstood certain aspects of the paper. Therefore, I am willing to raise my rating score if these questions and concerns are adequately addressed.

---

> ### Author Response · Authors · 2025-11-18
> **Response to Reviewer EcBF**
>
> We thank the reviewer for the thoughtful assessment of our theory, experiments, and writing, and for explicitly noting openness to revising the score. We address the main concerns and questions below.
>
> >  Linear data-generating process as significant simplification… while the idea that shared weights act as a practical analogue of Fisher information linear combination is intriguing, it remains more of an intuitive analogy than a formal justification.
>
> We appreciate the reviewer’s observation. The linear modeling assumption is indeed a simplification, but is widely adopted in the literature [1,2,3], due to the difficulty of guaranteeing identifiability in more general settings. Prior works [1,2] develop sufficient conditions for identifying the joint distribution and shared causal graph under this linear framework. Our work uses the same framework to explore how joint unpaired multimodal training can benefit an individual modality. To the best of our knowledge, ours is the first to study this question in the unpaired setting. We view our theoretical results as providing principled intuition as to what conditions enable unpaired multimodal training to yield performance gains, within a well-established framework for such analysis.
>
> Nonetheless, we agree with the reviewer that our theoretical results do not fully explain our empirical results. **We have added Section C.4 to the Appendix to strengthen the theoretical connection between the weight sharing in UML and our Fisher information argument in Section 3.1.** Proposition 1 shows that for a general (including non-linear) likelihood model with modality-specific parameters and a shared parameter block updated by both unimodal losses, the Fisher information on the shared block is the sum of the Fisher information on the unimodal blocks. This formalizes the intuition that weight sharing accumulates information across modalities, extending the key mechanism of the linear analysis to more general models.
>
> >Main differences from prior work
>
> We agree with the reviewer that neither weight sharing nor using unpaired data from multiple modalities is, by itself, new; our contribution is what we prove and demonstrate with these ingredients. We summarize this contribution with respect to prior work in **Section 2 of our global response**; please let us know if further clarification is needed.
>
> >What if the auxiliary modality is random or unrelated?
>
> We thank the reviewer for this insightful question! If the auxiliary modality is random or unrelated, the shared latent parameters see no additional Fisher information from $Y$; the theory predicts no benefit in that case. We test this in Sec. E.9: when we replace semantically related text with unrelated text sampled from other datasets, UML does not improve over the unimodal baseline. This confirms that our gains are driven by semantic correlation.
>
> > Beyond classification (e.g., retrieval)
>
> We agree that extending UML to retrieval and generation is an important next step, and acknowledge this in Section 5 (Limitations). Due to space and time constraints, we focus on a broad classification suite (self-supervised and supervised, robustness, three-modality audio–vision–text, transfer from language to vision). **We have updated our draft to clarify that retrieval is a natural extension and explicit direction for follow-up work.**
>
> > Concise presentation of theorem derivations
>
> We thank the reviewer for these suggestions to improve the clarity of our theoretical section. **We have revised Section C.3 of the paper, particularly the proofs of Lemma 3, Theorem 1, and Theorem 2. We will further rewrite this section for the camera-ready version.**
>
> > Improving clarity of Section 4
>
> We thank the reviewer for the helpful comments on the presentation of our experiments. **We have removed the color scheme for Tables 1-2 and added further experimental details to Section 4.1.**
>
> [1] Sturma, Nils, et al. "Unpaired multi-domain causal representation learning."
>
> [2] Timilsina, Subash, Sagar Shrestha, and Xiao Fu. "Identifiable shared component analysis of unpaired multimodal mixtures."
>
> [3] Huang, Yu, et al. "What makes multi-modal learning better than single (provably)."

---

> ### Comment · Reviewer_EcBF · 2025-11-19
>
> I thank the authors for their response; the rebuttal has effectively addressed my primary concerns.
>
> I have one minor follow-up question regarding the treatment of implicitly correlated semantics. Specifically, how does the proposed method behave when modalities appear uncorrelated at a fine-grained (instance) level but share an underlying coarse-grained hypothesis space?
>
> For example, consider a paired sample such as **`[Bulldog Image, Golden Retriever bark]`** from a vision–audio dataset. Although the two modalities are not semantically identical at the instance level, they still belong to the same broader concept (“Dog”). Thus, while the fine-grained semantics differ, they arguably lie within the same region of the conceptual manifold.
>
> In such cases, does the method leverage this implicit structural relationship, or does the instance-level mismatch hinder performance? In other words, does the model tend to interpret this as useful high-level signal or as uncorrelated noise?
>
> Aside from this point, the remainder of my concerns has been resolved. Thanks for clarifying such concerns.

---

> > ### Author Response · Authors · 2025-11-20
> > **Continued response to Reviewer EcBF**
> >
> > Thank you for your insightful discussion and prompt response. In all our supervised experiments, the situation you describe is exactly the regime we operate in: the two modalities only agree at a coarse, class level, not at the instance level. As shown in Table 4 and Section B.3.3, the textual descriptions are class-level prompts (e.g., “a photo of a bulldog”, “A great pyrenees is a large, white, shaggy-coated dog.”) rather than captions of specific images. Moreover, Figures 31 and 32 show that as these class-level descriptions become more descriptive, UML’s gains increase monotonically. Thus, UML is already trained and evaluated in a setting where samples like [Bulldog image, Golden Retriever bark] share only the broader “dog” concept, not fine-grained semantics, and we still observe consistent gains.
> >
> > More importantly, because our setting is fully unpaired, there is no notion of “instance-level mismatch” in the first place: the samples are never assumed to correspond pairwise, and UML only relies on structure shared at the distribution level.
> >
> > For our self-supervised experiments, we do have fine-grained cross-modal data, and again, UML improves over the corresponding unimodal baselines. Taken together, these results indicate that UML can leverage both coarse shared structure and more fine-grained information when available.
> >
> > We are very glad our earlier rebuttal helped resolve your main concerns, and we hope this addresses the remaining conceptual point as well. *Since you mentioned you would be open to raising your score if your questions were clarified, we would appreciate it if you could consider updating your rating accordingly.*

---

> > > ### Comment · Reviewer_EcBF · 2025-11-20
> > >
> > > Thank you for the clarifications, and apologies for the earlier misunderstanding. I agree that most of my concerns have now been resolved. Accordingly, I will raise my score.

---

> > > > ### Author Response · Authors · 2025-11-20
> > > >
> > > > Thank you for the thoughtful discussion and for engaging so carefully with our work. Your feedback significantly helped strengthen the submission, and we appreciate your decision to raise the score.

---

### Official Review · Reviewer_8Sv3 · 2025-11-01

**Soundness:** 2
**Presentation:** 3
**Contribution:** 2
**Rating:** 2
**Confidence:** 4

**Summary:**

The authors propose to augment unimodal models with unpaired training data from other modalities. Their approach shares weights across all modalities while optimizing for the downstream task on a joint, unpaired, multimodal training dataset. They evaluate their approach on several supervised, self-supervised, and transfer learning settings.

**Strengths:**

1. The paper is very well written - ideas and motivations are expressed clearly and in easy-to-understand terms; experimental results are stated and discussed in a well-organized manner.

2. The experiment in Section 4.4 on marginal rate-of-substitution between modalities is quite interesting and the results are insightful.

3. The authors empirically show that in the scenarios considered, it is indeed possible to achieve some practical performance improvement through joint/pre-training with data from modalities other than the target.

**Weaknesses:**

1. The idea that data from multiple modalities can be used for unified pretraining without any special consideration for the nature of the modalities being integrated is a fundamentally flawed premise. It is well known in literature that modalities can often have conflicting information, and even multimodal models, which often operate on explicitly paired data and are trained with the objective of aligning the modalities, struggle with this integration. There are plenty of works dedicated to addressing specifically this problem [a, b]. At the low level, for instance, modalities can often have different convergence rates [c] and provide conflicting gradients to the model [d], none of which can be thought of as being helpful from a training perspective without special treatment [d]. In fact, [c] specifically established the general impossibility of what the authors propose in this work.

2. Although the experiments show performance some improvements, the generality with which the paper is presented is misleading. Attempting to perform joint multimodal training using the proposed approach in the settings and datasets used in [a, b, c, d], is likely to falsify the findings reported, since special adjustments to cope with the challenges of integrating multiple modalities is needed to deal with those settings.

3. No ablation or analytical studies have been reported for the proposed approach, which makes it difficult to evaluate the contribution of the various design choices, for instance, the proposed UML objectives, the contribution of unpaired samples from a different modality, convergence in the uni-modal vs multi-modal setting, etc.

4. Finally, neither the idea of sharing model weights, nor doing pretraining with data from multiple modalities can be regarded as novel, since they have been around in the multimodal learning community for a long time [e, f, g]. In fact, this is also the problem that works on domain generalization attempt to solve, albeit framed in a different manner [h].

Minor:\
Line 449: "Further results on" -> "For further results on"

References:\
[a] Zhang et al., "Robust Multimodal Large Language Models Against Modality Conflict", ICML 2025.\
[b] Ma et al., "Improving Multimodal Learning Balance and Sufficiency through Data Remixing", ICML 2025.\
[c] Wang et al., "What makes training multi-modal classification networks hard?", CVPR 2020.\
[d] Javaloy et al., "Mitigating Modality Collapse in Multimodal VAEs via Impartial Optimization", ICML 2022.\
[e] Ngiam et al., "Multimodal Deep Learning", ICML 2011.\
[f] Hu et al., "Towards Unsupervised Sketch-based Image Retrieval", BMVC 2022.\
[g] Rastegar et al., "MDL-CW: A Multimodal Deep Learning Framework with Cross Weights", CVPR 2016.\
[h] Gulrajani et al., "In Search of Lost Domain Generalization", ICLR 2021.

**Questions:**

Please refer to the Weaknesses section.

---

> ### Author Response · Authors · 2025-11-18
> **Response to  Reviewer 8Sv3 (part 1)**
>
> We appreciate the reviewer’s detailed feedback and positive remarks on the clarity of the paper and the empirical gains from joint/pre-training. However, we respectfully believe that several core objections rest on misunderstandings of our problem setting, the cited works [a-d], and our paper’s claims. We address these in turn.
>
> > “Fundamentally flawed premise”
>
> Our paper does not claim that arbitrary modalities can always be naively merged. We work in the regime where different modalities share information about a common latent reality, i.e. where the mutual information $I(X;Y)>0$. Under this assumption, auxiliary unpaired data can help the target modality; if $I(X;Y) \approx 0$, they should not.
>
> We empirically verify this: when we use semantically unrelated text as the auxiliary modality for images, performance gains disappear (Table 24, Section E.9). In contrast, when text is semantically related, we consistently see improvements across 15+ benchmarks.
>
> > “Modality conflicts” can’t be handled without special treatment
>
> The cited works [a–d] analyze modality conflict and gradient imbalance in *paired* multimodal training, showing that there are settings where auxiliary gradients can hinder optimization unless reweighted or controlled. We fully agree that such regimes exist; our work does **not** claim that joint training is always beneficial for every possible conflict pattern.
>
> In Section 4.1 we show a case in which joint training can be beneficial with local conflicts. We evaluate our model on the MUSTARD sarcasm-detection dataset, where visual features (facial expressions) and utterance text are often intentionally misaligned. Despite this, UML improves unimodal performance, indicating that even with local conflicts, the modalities still provide a useful shared signal about intent. Negative or partially contradictory correlations can provide complementary information, requiring the model to learn structured dependencies. This is consistent with our theoretical regime and demonstrates that naive weight sharing *can* yield positive results in such settings.
>
> > [c] specifically established the general impossibility of what the authors propose
>
> Paper [c] does not claim a general impossibility result. It studies **paired** multimodal classification networks and, for specific architectures and datasets, empirically observes that naive joint training can underperform unimodal baselines due to overfitting and mismatched convergence rates. The phenomenon is framed as an optimization/overfitting issue for particular models and classification tasks, not as a fundamental impossibility of joint multimodal learning. [c] also discusses successful multimodal systems and notes that their negative findings do not extend to all tasks or settings. They do not claim that joint multimodal training is impossible in general, and they do not analyze the unpaired setting we study.
>
> Likewise, our paper does not claim a fully general result. We do not assert that our results extend to every possible multimodal training setting. As noted below, we provide relevant caveats in our paper, and have further revised the language to emphasize the scope of our claims.
>
> In summary, [c] and our paper are not contradictory. [c] empirically finds cases where unimodal networks outperform poorly optimized paired multimodal networks.  We theoretically and empirically show conditions in which *unpaired* multimodal learning outperforms unimodal learning.
>
> > the generality with which the paper is presented is misleading
>
> We agree that we do not want readers to overgeneralize beyond the particular regime that we study. As noted above, our work does not claim to apply to every possible multimodal learning setting.
>
> Our paper emphasizes the relevant scope and acknowledges that our results are not fully general in the following sections:
> - Our theoretical results (Section 3.1) specify the crucial assumption of shared information between the two modalities.
> - In Sections 4.1-4.2 we note that the text samples are conceptually related to the image samples.
> - In Section E.9 we show that performance gains disappear when the auxiliary modality is semantically unrelated to the target modality (reinforcing the need for shared information).
> - In Section 5 (Limitations) we note that our results are mainly on classification, and acknowledge other evaluation tasks that offer rich ground for future work.
>
> **We have also revised our paper to further improve the clarity of our claims:**
> - Added clarification to our Introduction (Section 1) and Limitations (Section 5) that our claim does not extend to every multimodal setting.
> - Added discussion on the limitations of multimodal learning to Section 2 including the works cited by the reviewer.

---

> ### Author Response · Authors · 2025-11-18
> **Response to Reviewer 8Sv3 (part 2)**
>
> > No ablation or analytical studies
>
> Below, we point to the ablations in our paper that address the design choices the reviewer mentions:
> - **UML vs unimodal objectives.** In both supervised and self-supervised settings, we keep the encoders and architecture identical and only switch the loss from unimodal training to the UML shared-head objective (Tables 1–2). This isolates the effect of the UML objective and weight sharing itself.
> - **Contribution of unpaired data from another modality.** We compare X-only training to X+unpaired-Y training under the same capacity and optimization, and additionally show that when Y is semantically unrelated to X, the gains disappear (Section E.9). This demonstrates that improvements come from informative unpaired data, not from architecture.
> - **Which weights to train (backbone vs shared head).** We contrast unimodal, UML, and UML(init) variants, differing in whether the backbone, shared head, or both are updated with multimodal data (Section E.1 and E.2 in the Appendix). Performance gaps between these settings quantify the impact of each training choice.
> - **Ratio of batches across modalities.** We vary the text:image batch ratio and find UML robust across a wide range, showing that the effect is not a fragile artifact of a particular scheduling heuristic (Section E.11).
> - **Unimodal v. multimodal convergence**: We empirically analyze convergence in our Gaussian experiments (Section 3, Figure 3). In a linear-Gaussian setting with two modalities (X, Y) derived from a ground-truth latent variable, we compare training using a fixed sample budget with only X versus splitting samples between X+Y. The model trained with X+Y reaches a lower reconstruction error on X, faster than training on X only.
>
> Beyond these, we also include ablations on the level of encoder alignment (Section E.1 and E.2), backbone scale (Section E.1, E.5), freezing vs unfreezing the text encoder (Section E.12), choice of text encoder (E.6), adding a third modality (audio–vision–text) (Section E.10), and synthetic Gaussian experiments (Section E.14) that match the theory. These ablations show how each design choice affects performance.
>
> > No novelty in weight sharing / multimodal pretraining
>
> We agree with the reviewer that neither weight sharing nor using unpaired data from multiple modalities is, by itself, new; our contribution is what we prove and demonstrate with these ingredients. We summarize this contribution with respect to prior work in **Section 2 of our global response**; please let us know if further clarification is needed.

---

> > ### Comment · Reviewer_8Sv3 · 2025-11-19
> >
> > 1. I thank the authors for taking the time to respond to my comments. Although their rebuttal addresses my concerns to some extent, I am still not entirely convinced that the authors successfully circumvent the well known limitations of jointly training over different data distributions / modalities. The authors claim that so long as the modalities share information / are conceptually related / share the same causal structure, it is possible to leverage them for joint training. However, [c] illustrates that differing convergence and generalization rates as an ubiquitous challenge when dealing with multiple modalities. Although not an impossibility result in the strictest sense, the general issue of conflict among modalities is still fairly fundamental and is observed in later works such as [a, b, d] as well. Therefore, given the well-known challenging nature of integrating modalities (even for tasks that actually need all modalities, i.e., multi-modal tasks), the claim of using one modality as pre-training / augmentation for another without any precise characterization of the preconditions, seems too strong. Merely mentioning that the modalities should have non-zero mutual information / share causal structure, etc., is too loose, since works [a-d], that tackle the conflicting nature among modalities explicitly, also use datasets where modalities share information, but where multimodal learning suffers from the said limitations of conflict nonetheless. My present view is, therefore, that more work needs to be done to precisely characterize and state the conditions when the proposed joint training would be beneficial and those under which it would not work. Without such an understanding, the work still remains somewhat misleading and incomplete.
> >
> > 2. The authors mention that "[c] also discusses successful multimodal systems and notes that their negative findings do not extend to all tasks or settings"; however, I was unable to find any such discussions in [c]. If the authors could point to the specifics, that would be helpful.
> >
> > 3. Reviewer mJFa raised a concern about the applicability of the proposed method in the missing modality setting. The authors responded that they have revised their limitations section to include this setting, but I could not find any discussion on missing modalities in the updated paper.

---

> ### Author Response · Authors · 2025-11-20
> **Continued response to Reviewer 8Sv3**
>
> > Further clarification regarding conflicting signals across modalities
>
> Thank you for the careful follow-up and for engaging so seriously with the paper. We fully agree that precisely characterizing all regimes in which joint multimodal training helps or hurts, including optimization pathologies and modality conflict as in [a–d],  is an important and largely open problem.  To clarify our scope, we have revised the Related Works section (Appendix A) to include a more extensive discussion of modality conflicts. We emphasize again that key concerns raised in [c] are only observed when joint training with **paired data** and training **from scratch**, neither of which applies to our setting.
>
> At the same time, we wish to clarify the scope of our contribution.
>
> (1) **Existence proof for unpaired multimodal learning**: Our results identify a new regime where unpaired multimodal learning is provably possible. To our knowledge, no prior work (theoretical or empirical) has demonstrated that fully unpaired multimodal data can improve the performance of a unimodal learner. In this sense, our work serves as **proof of existence**, demonstrating that there are settings where unpaired multimodal training is beneficial.
>
> (2) **Information gain characterization**: We characterize this regime through a shared information gain perspective and empirically validate it (e.g., the unrelated-text experiments where gains vanish).
>
> A full taxonomy of all multimodal conflict modes is beyond the scope of a single paper.  Demonstrating a clear regime where a method works, while acknowledging open theoretical gaps, is essential for scientific progress.  In the same spirit, we identify and validate a concrete regime where unpaired multimodal learning is effective, while noting that a full characterization of all multimodal conflicts is a critical avenue for future work.
>
> > Clarification about [c]
>
> [c] states in the final paragraph of its introduction:
>
> “Multi-modal networks have successfully been trained jointly on tasks including sound localization, image–audio alignment, and audiovisual synchronization. However, these tasks cannot be performed with a single modality, so there is no unimodal baseline, and the performance drop found in this paper does not apply.”
>
> Here, [c] acknowledges that their results do not necessarily apply to cases of joint training that fall outside of the regime that they study. They provide specific examples based on the presence of a unimodal baseline; while our work has a unimodal baseline, it likewise falls in a very different regime based on two key differences:
>
> 1. [c] studies multimodal training from scratch; however, our experiments, both supervised and self-supervised, use pretrained encoders or preprocessed features, placing us in a different optimization regime.
>
> 2. [c] analyzes paired multimodal learning, while our setting is fully unpaired. Whether the specific failure modes they identify arise in unpaired multimodal learning remains unknown and is itself an interesting future direction.
>
> > Revision for missing modality setting
>
> In our previous revision, we addressed this in the Limitations section by noting: “Other learning setups also offer natural extensions, such as combining separate unimodal datasets to form an unpaired multimodal dataset.” In our newest revision we have made this reference more explicit (Section 5, lines 501-503).

---

> > ### Comment · Reviewer_8Sv3 · 2025-11-21
> > **Response to the Authors (Part 1/2)**
> >
> > 1. I thank the authors for their prompt response to my comments. With the ongoing discussion, I should acknowledge that the scope of this paper is becoming increasingly clear to me. However, the presentation of the paper still remains generally misleading. There is no clear statement of why their method may or may not be immune to the hurdles of joint training with multiple modalities as identified in prior works [a-d] (or the fact that the authors did not explore this aspect), and more generally, the limitations of the proposed form of multimodal learning due to the inherent distribution shift present in the training data, which is a well-known open problem [i], in the introductory sections (such as the abstract and introduction) of the paper. In the revised versions, it is mentioned only in passing, often rather obscurely. I fully agree with the claim that this paper is a general proof of existence for the idea that unpaired data from other modalities can be used to enhance the abilities of unimodal models in certain specific scenarios. However, due to the existence of literature [a-d] (as well as [i] in the general case) that specifically illustrate the non-triviality of the problem, it becomes necessary to acknowledge the scope of the propositions and the limitations more clearly and in explicit terms throughout the paper, but more so in the introductory sections, based on which readers would build an initial impression of what exactly is being claimed.
> >
> > I appreciate the authors' admission of the fact that their results do not extend to every possible multimodal training setting (initial response). I also agree with the authors on their point that "Demonstrating a clear regime where a method works, while acknowledging open theoretical gaps, is essential for scientific progress". However, in the present form, neither have I been able to recognize a "clear regime" where their method works nor seen a complete acknowledgement of the "open theoretical gaps". The authors describe their paradigm as "unpaired" multimodal learning and one where the modalities share a non-zero mutual information / causal structure. Dealing with unpaired multimodal data makes their setting even more general relative to the paired ones dealt with in [a-d], which only increases (or maintains, in the best case) the possibilities of encountering the failure modes enunciated in such works. What keeps me hesitant about this paper is not so much the need for rigorous characterization of its various potential failure modes and the radical nature of the counterexamples to existing results that it purports to provide, but a lack of acknowledgement of such limitations.

---

> > > ### Comment · Reviewer_8Sv3 · 2025-11-21
> > > **Response to the Authors (Part 2/2)**
> > >
> > > 2. The point in [c] does not mean what is being claimed by the authors. When [c] states that "the performance drop found in this paper does not apply", what it means is that it is just not possible to benchmark such settings due to the impossibility of acquiring unimodal models. I suspect such a statement was written with more of an intent of disambiguation (rather than a disclaimer of generality, as I suppose is interpreted by the authors) to prevent readers from conflating its scope with the said scenario, which, at first glance might appear to be similar to the one considered. Furthermore, the authors claim that the findings of [c] do not apply to their work because they (i) use pretraining, and (ii) operate in an unpaired regime. For (i), it is not immediately obvious why pretraining would prevent differing optimization dynamics and generalization behaviour for different modalities. For (ii), as argued in the previous point, operating in the unpaired setting subsumes the paired scenario, and this increased generality can only be conceived of as one that is likely to increase (or in the best case, retain) the possibilities of encountering such failure modes.
> > >
> > > 3. The discussion about the limitations of the proposed method in the missing modality setting needs to be more extensive. Existing works show that the phenomenon of modality collapse [d] suffers particularly severe consequences in the case of missing modalities [j-m]. Moreover, in Figure 6 (as well as in the global response), the authors claim that the performance of their model improves as more modalities are added. However, [l] shows that the opposite phenomenon occurs under modality collapse. Therefore, it becomes important to acknowledge the possibility of modality collapse and the consequences thereof in the context of this work.
> > >
> > > To summarize, since the claims made in this paper are fairly radical and appear to contradict well-established lineages of prior work [a-d; i-m], stating the limitations / gaps / lack of understanding of the regimes considered in this work merely in passing and in the limitations section is not sufficient. They ought to have a more visible position in the paper, and any claims (theoretical or empirical) made that appear to go against the established literature, but whose reasons for being so is not understood / explored by the authors, should explicitly acknowledge such limitations.
> > >
> > > References:
> > >
> > > [i] Yang et al., "Change is Hard: A Closer Look at Subpopulation Shift", ICML 2023.\
> > > [j] Wu et al., "Multimodal Patient Representation Learning with Missing Modalities and Labels", ICLR 2024.\
> > > [k] Kim et al., "Missing Modality Prediction for Unpaired Multimodal Learning via Joint Embedding of Unimodal Models", ECCV 2024.\
> > > [l] Chaudhuri et al., "A Closer Look at Multimodal Representation Collapse", ICML 2025.\
> > > [m] Dai et al., "Unbiased Missing-modality Multimodal Learning", ICCV 2025.

---

> ### Author Response · Authors · 2025-11-22
> **Continued Response to Reviewer 8Sv3**
>
> > On acknowledging limitations in earlier sections of the work
>
> We thank the reviewer for their continued and careful engagement with our paper. Based on the reviewer’s suggestion, we have substantially strengthened the discussion of modality conflict, optimization instability, and missing-modality regimes throughout the paper, including the introduction section. **Specifically, we have added explicit clarifications in the Introduction (Sec. 1, lines 84–89, 97–98), Sec. 2 (lines 134–138), Sec. 3 (lines 176-177), Sec. 3.2 (lines 260-262, 266–290), Sec. 4 (lines 323–328), the Limitations section (Sec. 5, lines 516–519; 521–526), and Related Work (Appendix A, lines 1010–1029).** We believe these revisions now clearly communicate both the scope and limitations of our setting, ensuring that readers walk away with the key message of the paper while remaining cognizant of its open challenges.
>
> > Regarding subpopulation shifts
>
> We agree that subpopulation shift is an important problem, but it is orthogonal to the setting studied in our work. The paper cited by the reviewer [i], studies a different regime: training on one subpopulation and evaluating on a held-out subpopulation,  rather than cross-modal subpopulation shifts that the reviewer points out. Our work neither targets nor is designed to address that problem. Instead, we do evaluate robustness to distribution shifts in our joint training setup under covariate shifts, and observe consistent improvements over unimodal baselines (Fig. 5).
>
> More broadly, across multiple types of distribution shift, empirical risk minimization (ERM), where data from different domains is pooled and trained jointly, without explicit domain labels, has been shown to outperform many specialized methods designed to address distribution shifts [1]. This observation is, in spirit, aligned with our approach, where we jointly train a single model on data from multiple modalities (effectively multiple domains).
> We therefore view subpopulation shift as a largely separate line of work and outside the intended scope of this paper.
>
> [1] Gulrajani, Ishaan, and David Lopez-Paz. "In search of lost domain generalization." arXiv preprint arXiv:2007.01434 (2020).
>
> > Operating in the unpaired setting subsumes the paired scenario, and this increased generality can only be conceived of as one that is likely to increase (or in the best case, retain) the possibilities of encountering such failure modes.
>
> We respectfully disagree with the reviewer’s claim that moving from paired to unpaired multimodal data “only increases (or maintains) the possibilities of encountering” the failure modes studied in [a–d]. At the level of data, yes, an unpaired setting is more general than a paired one. However, the failure modes in these works are observed in architectures that exploit *paired* supervision: the model is explicitly trained to align or co-explain two modalities at the *sample level*, so any spurious or contradictory cross-modal correlations are directly reinforced by the loss.
>
> In contrast, our unpaired setting only sees **one** data point at a time from a single modality and aggregates gradients across modalities only through shared parameters. This setup does not force pointwise alignment between conflicting views and therefore does not instantiate the same mechanisms that give rise to the paired failure modes in [a–d].
>
> To be clear, we do not claim that unpaired training is free of multimodal conflicts or optimization challenges. However, the reviewer’s assertion that it *necessarily* worsens the risk of these specific failure modes does not account for the fundamentally different optimization landscape induced by unpaired training.
>
> >  For (i), it is not immediately obvious why pretraining would prevent differing optimization dynamics and generalization behaviour for different modalities.
>
> Our intuition is guided by empirical findings such as the Platonic Representation Hypothesis [2], which demonstrates that independently trained, pretrained models across modalities converge towards a shared geometric structure in representation space, including aligned kernels. This suggests that optimization operating in pretrained feature space can be more stable than optimization over raw inputs, which can exhibit substantial distribution shift. We stress that this is an intuition, not a claim: without concrete evidence, we neither assert nor rule out whether the paired-modality failure modes in [c] transfer to our unpaired setting, and therefore form an interesting base for future work (as also discussed in the latest Limitations section (Sec. 5)).
>
> [2] Huh, Minyoung, et al. "The platonic representation hypothesis." arXiv preprint arXiv:2405.07987 (2024).
>
> *We thank the reviewer for the constructive discussion, which has helped improve the clarity of the paper. If these responses resolve the remaining concerns, we would appreciate consideration of an updated score.*

---

> > ### Comment · Reviewer_8Sv3 · 2025-11-24
> >
> > I thank the authors for their diligent follow-ups on my comments. Although my concerns have been partly addressed, some key gaps still remain, which I state below:
> >
> > 1. It is important to acknowledge the limitations in the abstract as well. Since such limitations are fundamental to the joint training with multiple modalities, reading the abstract, one should not have the incorrect impression that they are resolved in this paper, something that a reader is likely to get from the abstract in its present form.
> >
> > 2. The authors' interpretation of subpopulation shift, as described in [i] (and extensively studied in later literature), is incorrect. Quoting directly from [i], it refers to the "changes in the proportion of some subpopulations between training and deployment (Koh et al., 2021)". It does not refer to "training on one subpopulation and evaluating on a held-out subpopulation", as misinterpreted by the authors. Now, each training modality can be thought of as a subpopulation. Therefore any of the subpopulation shifts, i.e., spurious correlations, attribute imbalance, class imbalance, and attribute generalization, as described in [i], are likely in the setting considered by the authors, depending on the choice of the source and the target modalities. To the best of my knowledge, since these are one of the most general categorizations of potential challenges in learning under distribution shifts (which should subsume some of the challenges specific to multimodal training as studied in [a-d]), I made the associated remark in my previous comment.
> >
> > 3. The comments about unpaired training and the impact of pretraining address my concerns. However, these are quite central to the premise of the paper and are not immediately obvious. Therefore, they need to be included in appropriate places around section 3. Also, the comment that the unpaired "setup does not force pointwise alignment between conflicting views and therefore does not instantiate the same mechanisms that give rise to the paired failure modes in [a–d]" needs to be made more theoretically rigorous, as it sounds rather hand-wavy at the moment.

---

> ### Author Response · Authors · 2025-11-27
> **Continued Response to Reviewer 8Sv3**
>
> > It is important to acknowledge the limitations in the abstract as well
>
> We thank the reviewer for their diligent engagement with the paper and our rebuttal. **To ensure that our abstract is not overly general, we have edited it to include the main limitation that we empirically observe:** That it is necessary for the unpaired data to share underlying semantic information.
>
> However, we have opted not to explicitly foreground “modality conflict/imbalance” in the abstract for two key reasons:
>
> 1. The conflict/collapse phenomena cited by the reviewer are established primarily for paired multimodal training, and there is no evidence that the same mechanisms necessarily arise in fully unpaired joint training. Moreover, unpaired training does not impose any sample-wise alignment objective, which is a key driver of conflict in paired settings.
>
> 2. We do not observe such degradations in our experiments across ~20 benchmarks, and multiple vision/text/audio encoders. Given this, adding “modality conflict” language to the abstract would be speculative and risks reporting a failure mode that we did not observe.
>
> We do agree that it is important to clearly acknowledge what we do not address. Accordingly, we have added extensive discussion on modality conflict in our Introduction, Related Works, Methods, and Limitations sections. We believe that this is sufficient for a reader to understand the limitations of our experimental setting.
>
> > Regarding subpopulation shifts
>
> We agree that distribution-shift frameworks are broad, and that many multimodal failure modes in [a–d] can be viewed as instances of learning under heterogeneous distributions. Accordingly, we fully concur that such suboptimalities could also arise in our setting, and our manuscript discusses these conflicts throughout the paper (specifically in the Related Work section).
>
> > Including unpaired training and the impact of pretraining in Section 3
>
> We have added remarks regarding unpaired v. paired training and the impact of pretraining to the end of section 3.2 (lines 314-319).
>
> > The comment that the unpaired "setup does not force pointwise alignment between conflicting views and therefore does not instantiate the same mechanisms that give rise to the paired failure modes in [a–d]" needs to be made more theoretically rigorous, as it sounds rather hand-wavy at the moment.
>
>  In paired settings (e.g., [c]), the loss typically includes a sample-wise alignment term:
>
> $L_{\text{paired}} = E_{(x,y) \sim P_{XY}} [ \| f(x) - g(y) \|^2 ]$
>
> This objective explicitly penalizes the model whenever $f(x)$ deviates from $g(y)$. If a specific pair $(x_i, y_i)$ contains conflicting information (e.g., an image $x_i$ and a sarcastic caption $y_i$ that implies the opposite sentiment), minimizing $L_{\text{paired}}$ forces the model to suppress the features of one modality to match the other, driving the optimization instability observed in prior work.
>
> In contrast, the unpaired UML objective factorizes over the marginals:
>
> $L_{UML} = E_{x \sim P_X} [L_X(x; \theta)] + E_{y \sim P_Y} [L_Y(y; \theta)]$
>
> Here, gradients from modality $Y$ update the shared parameters $\theta$ to better represent the distribution $P(Y)$, but there is no term that forces $f(x_i)$ to align with any specific $g(y_j)$. Consequently, the specific mechanism of pointwise alignment conflict—where the model is penalized for local discrepancies between mismatched modalities—is structurally absent in our objective. While gradient interference on shared parameters $\theta$ is still possible, the optimization landscape is fundamentally distinct from the paired setting.

---

### Author Response · Authors · 2025-11-18
**Global Response**

We thank all of the reviewers for their insightful and helpful feedback. We are glad that they found:
- The paper is well-written and clear [8Sv3, EcBF, mJFa]
- The experiments are extensive [EcBF]
- The theoretical results are strong and sound [EcBF, mJFa]

Below we summarize key revisions and clarify our paper’s contribution (and answer individual questions in reviewer-specific responses).

**1. UPDATES TO THE PAPER**

In response to comments from the reviewers, we have made the following changes to our paper. **Updates are purple in our revised PDF**.

- Revised Sections 1, 2, and 5 to clarify the extent and scope of our claims, and discuss limitations of multimodal learning found in the literature.
- Added further theoretical results in Section C.4 to strengthen the connection between weight-sharing in UML and our Fisher information argument in Section 3.1.
- Revised the presentation of our experiments in Section 4.1 and our theoretical results in Section C.3 to improve clarity.
- Revised our limitations (Section 5) to comment further on natural extensions of our work.

**2. CONTRIBUTION WITH RESPECT TO PRIOR WORK**

Existing work typically falls into one of the following categories:

- **Paired multimodal learning improving $X$ using paired $(X,Y)$**: Most multimodal methods rely on paired $(x,y)$ supervision and cross-modal losses where improvements on $X$ are driven by explicit pairing and alignment objectives [1,2].
- **Unpaired data only in combination with paired or pre-aligned signals**: “Unpaired” methods usually still use some paired data [3], pseudo-alignments [4,5,6], or pre-aligned feature spaces (e.g., CLIP) to transfer signal across modalities [8].
- **Unpaired multimodal data confined to intra-domain or heavily engineered alignment**: Existing unpaired approaches often stay within a single modality family (e.g., vision) [7] or require explicit pseudo-labeling / cross-domain matching that hard-codes inductive biases into the representation [4,5,6].


In contrast, our work shows that one can train on fully unpaired audio, vision, and text datasets using separate encoders with a single shared head. Our results show that this:
-  Improves unimodal performance on the tested benchmarks,
- Scales monotonically as more modalities are added,
- Learns multimodal structure (neurons, prototype alignment) without ever seeing paired data, and
- Is backed by a theoretical analysis motivating when/why this works.


[1] Zhai, Xiaohua, et al. "Lit: Zero-shot transfer with locked-image text tuning." Proceedings of the IEEE/CVF conference on computer vision and pattern recognition. 2022.

[2] Cijo Jose, et al. “A unified framework for image-and pixel-level vision-language alignment.” arXiv preprint arXiv:2412.16334 (2024).

[3] Geng, Xinyang, et al. "Multimodal masked autoencoders learn transferable representations." arXiv preprint arXiv:2205.14204 (2022).

[4] Xi, Johnny, et al. "Propensity score alignment of unpaired multimodal data." Advances in Neural Information Processing Systems 37 (2024): 141103-141128.

[5] Demetci, Pinar, et al. "SCOT: single-cell multi-omics alignment with optimal transport." Journal of computational biology 29.1 (2022): 3-18.

[6] Ryu, Jayoung, et al. "Cross-modality matching and prediction of perturbation responses with labeled Gromov-Wasserstein optimal transport." arXiv preprint arXiv:2405.00838 (2024).

[7] Girdhar, Rohit, et al. "Omnimae: Single model masked pretraining on images and videos." Proceedings of the IEEE/CVF conference on computer vision and pattern recognition. 2023.

[8] Gao, Peng, et al. "Clip-adapter: Better vision-language models with feature adapters." International Journal of Computer Vision 132.2 (2024): 581-595.

---

### Meta-Review · Area_Chair_WAXe · 2026-01-05

**Summary:**

The paper introduces UML (Unpaired Multimodal Learner), a framework for leveraging unpaired multimodal data to enhance representation learning in a target modality. The idea is to share parameters across modalities. Section 3 provides theorems for a linear model stating that adding an unpaired auxiliary modality reduces estimator variance. Experiments show that auxiliary modalities boost target representations, and the benefits compound with more modalities. The experimental results evaluate self-supervised, supervised, and transfer learning settings.

The reviewers generally acknowledge that the theoretical and empirical results are interesting.

**Reviewer Concerns:**

Reviewer mJFa:
- **Why not use existing models (e.g., CLIP) to align unpaired data**: The authors note that this approach is not applicable when a cross-modal model trained on massive paired data does not exist for a particular pair of modalities and argue their work shows that even without paired data learning is possible.
- **Experiments using various encoders and shared networks**: The authors note that their ablations include various encoders for different modalities and shared networks and further ablations are provided in appendix E.
- **Comparison to related works**: The authors note that they mainly compare to unimodal baselines since no prior work addresses the setting of unpaired multimodal learning. They acknowledge that some existing multimodal methods can be adapted to the unpaired setting but they were unable to include it during the rebuttal.

Reviewer EcBF:
- **Theoretical assumptions are too restrictive**: The authors agree and provide new analysis in Appendix C.4 including a proposition for non-linear cases.
- **Verbose derivations of theorems**: The authors have improved the writing Appendix C.4 and proofs.
- **Verbose experimental section**: The authors made revisions to Section 4.
- **Novelty and difference from prior works?** The authors summarize contributions and revise section 2.
- **Most experiments are conducted on classification tasks, what about image-text retrieval?**: The authors noted that they have covered a broad set of classification tasks and retrieval is a natural extension.

Reviewer 8Sv3:
- **Flawed premise**: The reviewer referred to multiple prior works that observed negative results in training models on multiple modalities that led to “modality conflicts” and hindered optimization. The authors engaged extensively with the reviewer and emphasized that their paper should be considered as a positive result in a particular setting not considered in prior works. Specifically, they clarify that they consider unpaired multimodal learning and show that it outperforms unimodal learning and that they operate in a regime where different modalities share information about a common latent reality and empirically verify that is needed. The authors made edits throughout the paper to clarify their setting.
- **Missing ablations**: The authors pointed to ablations in the paper.
- **Novelty**: The authors note that their contribution is to prove and demonstrate a positive result that did not exist before.

**Reviewer Scores:**

Initially reviewers gave a score of 4 (marginally below acceptance threshold) and two scores of 6 (marginally above acceptance threshold). Reviewer mJFa and EcBF agreed to raise their scores to 6 and 8 (from 4 and 6) after the discussion with the authors as their concerns were resolved through changes to the paper and clarifications by the authors.

Reviewer 8Sv3 had a major concern regarding the positioning of the paper in relation to other negative results. The reviewer engaged with authors extensively. As a result, authors made various changes throughout the paper to clarify their position. The AC recommends that the new text on lines 84-98 be further highlighted as a separate paragraph and potentially include references to related assumptions in Section 3.1 and the fact that gains disappear when modalities are unrelated as shown in Section E.9.

The authors should consider adding comparison with existing multimodal methods that can be adapted to their setting as noted by reviewer mJFa.

---

### Decision · Program_Chairs · 2026-01-26

Accept (Poster)